# The dynamic conformational landscape of the protein methyltransferase SETD8

Shi Chen[1,2†], Rafal P Wiewiora[1,3†], Fanwang Meng[4], Nicolas Babault[5,6,7,8], Anqi Ma[5,6,7,8], Wenyu Yu[9], Kun Qian[10], Hao Hu[10], Hua Zou[11], Junyi Wang[2], Shijie Fan[4,12], Gil Blum[2], Fabio Pittella-Silva[2], Kyle A Beauchamp[3], Wolfram Tempel[9], Hualiang Jiang[4,12], Kaixian Chen[4,12], Robert J Skene[11], Yujun George Zheng[10], Peter J Brown[9], Jian Jin[5,6,7,8], Cheng Luo[4,12]*, John D Chodera[3]*, Minkui Luo[2,13]*

[1]Tri-Institutional PhD Program in Chemical Biology, Memorial Sloan Kettering Cancer Center, New York, United States; [2]Chemical Biology Program, Memorial Sloan Kettering Cancer Center, New York, United States; [3]Computational and Systems Biology Program, Memorial Sloan Kettering Cancer Center, New York, United States; [4]Drug Discovery and Design Center, CAS Key Laboratory of Receptor Research, State Key Laboratory of Drug Research, Shanghai Institute of Materia Medica, Chinese Academy of Sciences, Shanghai, China; [5]Mount Sinai Center for Therapeutics Discovery, Icahn School of Medicine at Mount Sinai, New York, United States; [6]Department of Pharmacological Sciences, Icahn School of Medicine at Mount Sinai, New York, United States; [7]Department of Oncological Sciences, Icahn School of Medicine at Mount Sinai, New York, United States; [8]Tisch Cancer Institute, Icahn School of Medicine at Mount Sinai, New York, United States; [9]Structural Genomics Consortium, University of Toronto, Toronto, Canada; [10]Department of Pharmaceutical and Biomedical Sciences, University of Georgia, Athens, United States; [11]Takeda California, Science Center Drive, San Diego, United States; [12]University of Chinese Academy of Sciences, Beijing, China; [13]Program of Pharmacology, Weill Cornell Medical College of Cornell University, New York, United States

*For correspondence:
cluo@simm.ac.cn (CL);
john.chodera@choderalab.org (JDC);
luom@mskcc.org (ML)

[†]These authors contributed equally to this work

**Abstract** Elucidating the conformational heterogeneity of proteins is essential for understanding protein function and developing exogenous ligands. With the rapid development of experimental and computational methods, it is of great interest to integrate these approaches to illuminate the conformational landscapes of target proteins. SETD8 is a protein lysine methyltransferase (PKMT), which functions in vivo via the methylation of histone and nonhistone targets. Utilizing covalent inhibitors and depleting native ligands to trap hidden conformational states, we obtained diverse X-ray structures of SETD8. These structures were used to seed distributed atomistic molecular dynamics simulations that generated a total of six milliseconds of trajectory data. Markov state models, built via an automated machine learning approach and corroborated experimentally, reveal how slow conformational motions and conformational states are relevant to catalysis. These findings provide molecular insight on enzymatic catalysis and allosteric mechanisms of a PKMT via its detailed conformational landscape.

DOI: https://doi.org/10.7554/eLife.45403.001

**eLife digest** Our cells contain thousands of proteins that perform many different tasks. Such tasks often involve significant changes in the shape of a protein that allow it to interact with other proteins or ligands. Understanding these shape changes can be an essential step for predicting and manipulating how proteins work or designing new drugs. Some changes in protein shape happen quickly, whereas others take longer. Existing experimental approaches generally only capture some, but not all, of the different shapes an individual protein adopts.

A family of proteins known as protein lysine methyltransferases (PKMTs) help to regulate the activities of other proteins by adding small tags called methyl groups to specific positions on their target proteins. PKMTs play important roles in many life processes including in activating genes, maintaining stem cells and controlling how organs develop.

It is important for cells to properly control the activity of PKMTs because too much, or too little, activity can promote cancers and neurological diseases. For example, genetic mutations that increase the levels of a PKMT known as SETD8 appear to promote the progression of some breast cancers and childhood leukemia. There is a pressing need to develop new drugs that can inhibit SETD8 and other PKMTs in human patients. However, these efforts are hindered by the lack of understanding of exactly how the shape of PKMT proteins change as they operate in cells.

Chen, Wiewiora et al. used a technique called X-ray crystallography to generate structural models of the human SETD8 protein in the presence or absence of native or foreign ligands. These models were used to develop computer simulations of how the shape of SETD8 changes as it operates. Further computational analysis and laboratory experiments revealed how slow changes in the shape of SETD8 contribute to the ability of the protein to attach methyl groups to other proteins.

This work is a significant stepping-stone to developing a complete model of how the SETD8 protein works, as well as understanding how genetic mutations may affect the protein's role in the body. The next step is to refine the model by integrating data from other approaches including biophysical models and mathematical calculations of the energy associated with the shape changes, with a long-term goal to better understand and then manipulate the function of SETD8.
DOI: https://doi.org/10.7554/eLife.45403.002

## Introduction

Proteins are not static, but exist as an ensemble of conformations in dynamic equilibrium (*Wei et al., 2016*). Characterization of conformational heterogeneity can be an essential step towards interpreting function, understanding pathogenicity, and exploiting pharmacological perturbation of target proteins (*Ferguson and Gray, 2018*; *Latorraca et al., 2017*; *Lu et al., 2016*). Biophysical techniques such as X-ray crystallography (*Shi, 2014*), nuclear magnetic resonance (NMR) (*Huang and Kalodimos, 2017*), and cryo-electron microscopy (*Fernandez-Leiro and Scheres, 2016*) mainly provide static snapshots of highly-populated conformational states. While complementary techniques such as relaxation-dispersion NMR can resolve a limited number of low-population states, they are incapable of providing detailed structural information (*van den Bedem and Fraser, 2015*). By contrast, molecular simulations provide atomistic detail—a prerequisite to structure-guided rational ligand design—and insight into relevant conformational transitions (*Wei et al., 2016*). The emergence of Markov state models (MSMs) has shown the power of distributed molecular simulations in resolving complex kinetic landscapes of proteins (*Husic and Pande, 2018*; *Plattner et al., 2017*). By integrating simulation datasets with MSMs, functionally relevant conformational dynamics as well as atomistic details can be extracted (*Plattner et al., 2017*). Recently, MSMs have been used to identify key intermediates for enzyme activation (*Shukla et al., 2014*; *Sultan et al., 2017*) and allosteric modulation (*Bowman et al., 2015*). However, these approaches are limited by the number of seed structures and timescales accessible by molecular simulations (generally microseconds for one structure) relative to the reality of complicated conformational transitions (up to milliseconds for multiple structures) (*Klepeis et al., 2009*). To overcome the limitations of individual techniques, efforts have been made to combine simulation with experiment to characterize and experimentally validate conformational landscape models of proteins that provide insight into functions (*Hart et al., 2016*; *Knoverek et al., 2019*; *Latallo et al., 2017*; *Zimmerman et al., 2017*).

Protein lysine methyltransferases (PKMTs) comprise a subfamily of posttranslational modifying enzymes that transfer a methyl group from the cofactor S-adenosyl-L-methionine (SAM) (*Luo, 2018*). PKMTs play epigenetic roles in gene transcription, cellular pluripotency, and organ development (*Allis and Jenuwein, 2016*; *Murn and Shi, 2017*). Their dysregulation has been implicated in neurological disorders and cancers (*Dawson, 2017*; *Flavahan et al., 2017*). SETD8 (SET8/Pr-SET7/KMT5A) is the sole PKMT annotated for monomethylation of histone H4 lysine 20 (H4K20me) (*Fang et al., 2002*; *Nishioka et al., 2002*) and many nonhistone targets such as the tumor suppressor p53 and the p53-stabilizing factor Numb (*Dhami et al., 2013*; *Shi et al., 2007*). Disruption of endogenous SETD8 leads to cell cycle arrest and chromatin decondensation, consistent with essential roles for SETD8 in transcriptional regulation and DNA damage response (*Beck et al., 2012*; *Liu et al., 2010*; *Veschi et al., 2017*). SETD8 has also been implicated in cancer invasiveness and metastasis (*Yang et al., 2012*). High expression of SETD8 is associated with pediatric leukemia and its overall low survival rate (*Hashemi et al., 2014*). While there is enormous interest in elucidating functional roles of SETD8 in disease, it has been challenging to develop potent, selective, and cellularly active SETD8 inhibitors (*Blum et al., 2014*; *Milite et al., 2016a*; *Milite et al., 2016b*).

Given the essential roles of conformational dynamics in enzymatic catalysis (*Schramm, 2011*; *Wei et al., 2016*) and our current limited knowledge of conformational landscapes of PKMTs, we envisioned characterizing the dynamic conformational landscapes of SETD8 and its cancer-associated mutants with atomic resolution. To access previously-unseen, less-populated conformational states of SETD8 to seed parallel distributed molecular dynamics (MD) simulations, we envisioned trapping these conformations with small-molecule ligands. Here we solved four distinct crystal structures of SETD8 in alternative ligand-binding states with covalent SETD8 inhibitors and native ligands. With the aid of these new structures, we generated an aggregate of six milliseconds of unbiased explicit solvent MD simulation data for apo- and SAM-bound SETD8. Using a machine learning approach to select features and hyperparameters for MSMs via extensive cross-validation, we clustered apo-SETD8 conformers into 24 kinetically distinct, likely functionally relevant metastable conformational states and annotated how the conformational landscape is remodeled upon SAM binding. We then explored these conformational landscape models experimentally with stopped-flow kinetics and isothermal titration calorimetry by examining SAM binding, characterizing rationally-designed SETD8 variants with increased catalytic efficiency, and resolving multiple timescales associated with transitions among these conformers. The resulting model furnishes key insights into how these dynamic conformations play a role in catalysis of SETD8 and how cancer-associated SETD8 mutants alter this process allosterically through reshaping the conformational landscape rather than directly affecting the catalytic site. These findings suggest the importance of referencing conformational landscapes for elucidating enzymatic catalysis and allosteric regulation of SETD8 and likely other PKMTs.

## Results

### Crystal structures of SETD8 associated with hidden conformations

To identify hidden high-energy conformational states of SETD8, we envisioned a strategy of trapping the associated conformers with small-molecule ligands. The development of high-affinity SETD8 inhibitors with canonical target-engagement modes is challenging (*Milite et al., 2016b*), and led us to exploit covalent inhibitors (*Blum et al., 2014*; *Butler et al., 2016*). These compounds can overcome the high energy penalties associated with hidden conformers through the irreversible formation of energetically-favored inhibitor–SETD8 adducts. Our prior efforts led to the development of covalent inhibitors containing 2,4-diaminoquinazoline arylamide and multi-substituted quinone scaffolds by targeting Cys311 (*Blum et al., 2014*; *Butler et al., 2016*). Upon further optimization of these scaffolds, we identified MS4138 (**Inh1**) and SGSS05NS (**Inh2**) (*Luo et al., 2015*), two structurally distinct covalent inhibitors with the desired potency against SETD8 (*Figure 1a*, *Figure 1—figure supplement 1*). X-ray crystal structures of SETD8 were then solved in complex with **Inh1** and **Inh2**, respectively (*Figure 1b,c*, *Figure 1—figure supplements 2* and *3*, *Table 1*). Notably, despite the overall structural similarity of the pre-SET, SET, and SET-I motifs, the **Inh1**- and **Inh2**-SETD8 binary complexes (**BC-Inh1** and **BC-Inh2**) differ from the SETD8-SAH-H4 ternary complex (**TC**) (*Couture et al., 2005*; *Couture et al., 2008*; *Xiao et al., 2005*) by the distinct conformations of their

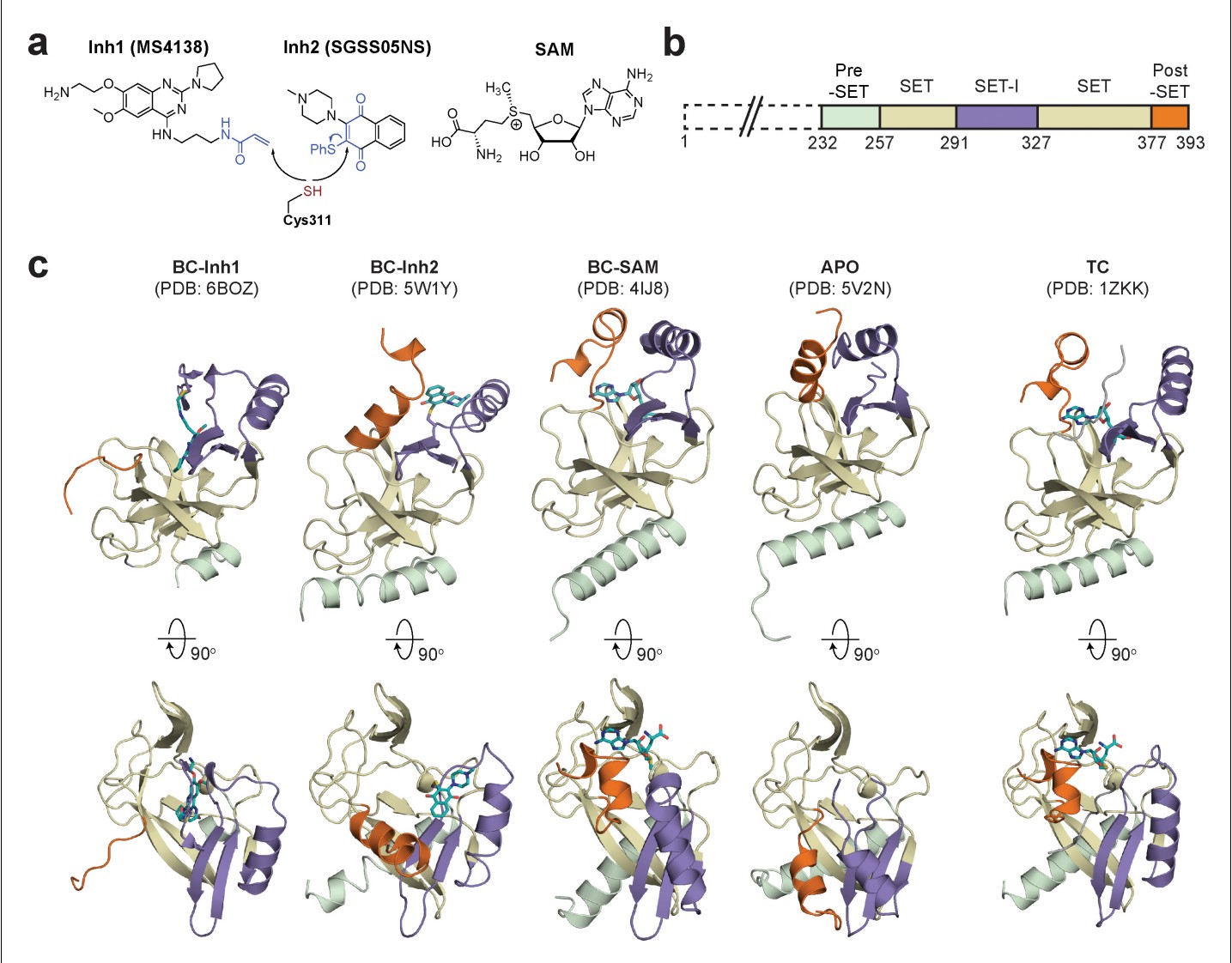

**Figure 1.** Diverse SETD8 conformations captured in altered ligand-binding states. (**a**) Structures of SETD8 ligands involved in this work. Two covalent inhibitors targeting Cys311 (MS4138 as **Inh1** and SGSS05NS as **Inh2**) and the cofactor SAM were used as ligands to trap neo-conformations of SETD8. (**b**) Domain topology of SETD8 (Uniprot: Q9NQR1-1). Four functional motifs at SETD8's catalytic domain are colored: pre-SET (light green), SET (dark yellow), SET-I (purple), and post-SET (orange). (**c**) Cartoon representations of four neo-structures of SETD8 (**BC-Inh1**, **BC-Inh2**, **BC-SAM**, and **APO**) and a structure of a SETD8-SAH-H4 ternary complex (**TC**). These structures are shown in two orthogonal views with ligands, pre-SET, SET, SET-I, and post-SET colored in cyan, light green, dark yellow, purple, and orange, respectively.

DOI: https://doi.org/10.7554/eLife.45403.003

The following figure supplements are available for figure 1:

**Figure supplement 1.** Synthesis and characterization of two covalent inhibitors targeting SETD8.
DOI: https://doi.org/10.7554/eLife.45403.004
**Figure supplement 2.** Crystal structure of SETD8 in complex with Inh1 (BC-Inh1).
DOI: https://doi.org/10.7554/eLife.45403.005
**Figure supplement 3.** Crystal structure of SETD8 in complex with Inh2 (BC-Inh2).
DOI: https://doi.org/10.7554/eLife.45403.006
**Figure supplement 4.** Crystal structure of SETD8 in complex with the cofactor SAM (BC-SAM).
DOI: https://doi.org/10.7554/eLife.45403.007
**Figure supplement 5.** Crystal structure of apo SETD8 (APO).
DOI: https://doi.org/10.7554/eLife.45403.008

**Table 1.** Data collection and refinement statistics of crystallography.

| | BC-Inh1 | BC-Inh2 | BC-SAM | APO |
|---|---|---|---|---|
| PDB Code | 6BOZ | 5W1Y | 4IJ8 | 5V2N |
| **Data collection** | | | | |
| Wavelength (Å) | 0.98 | 0.98 | 0.98 | 0.98 |
| Space group | $P2_12_12_1$ | $P2_12_12_1$ | $P6_122$ | $P4_32_12$ |
| **Cell dimensions** | | | | |
| a, b, c (Å) | 31.56, 68.06, 125.90 | 58.35, 39.79, 131.90 | 101.44, 101.44,140.80 | 60.6, 60.6, 80.7 |
| α, β, γ (°) | 90, 90, 90 | 90, 90, 90 | 90, 90, 120 | 90, 90, 90 |
| Resolution (Å) | 62.95–2.40 (2.49–2.40) | 43.70–1.70 (1.73–1.70) | 47.72–2.00 (2.11–2.00) | 50.00–2.00 (2.03–2.00) |
| Unique reflections | 11,209 (1550) | 34,422 (1769) | 29,619 (4231) | 10,736 (918) |
| Redundancy | 3.6 (3.0) | 3.8 (3.6) | 21.6 (22.0) | 14.5 |
| Completeness (%) | 99.5 (97.8) | 99.4 (96.3) | 100.0 (100.0) | 99.8 (97.0) |
| I/σ(I) | 8.2 (3.4) | 15.0 (1.8 | 19.7 (4.0) | 20.0 (1.1) |
| $R_{sym}$[a] | 0.110 (0.361) | 0.064 (0.657) | 0.112 (0.942) | 0.13 (0.460) |
| $R_{pim}$ | 0.065 (0.129) | 0.036 (0.386) | 0.025 (0.205) | 0.040 (0.4) |
| **Refinement** | | | | |
| No. protein molecules/ASU | 2 | 2 | 2 | 1 |
| Resolution (Å) | 62.95–2.40 | 35.00–1.70 | 43.96–2.00 | 48.47–2.00 |
| Reflections used or used/free | 11,165/1065 | 32,998/1373 | 28,045/1516 | 10,153/513 |
| Rwork | 0.179 | 0.201 | 0.176 | 0.183 |
| Rfree | 0.242 | 0.237 | 0.199 | 0.249 |
| Average B value (Å$^2$) | 38.9 | 20.9 | 37.8 | 41.6 |
| *Protein* | 39.8 | 20.7 | 37.9 | 40.7 |
| *Compound* | 20.0 | 16.4 | 24.5 | n/a |
| *Other* | 38.5 | 20.7 | 44.7 | 50.9 |
| *Water* | 30.4 | 24.3 | 36.9 | 49.4 |
| Number of Atoms | 2299 | 2835 | 2675 | 1404 |
| *Protein* | 2161 | 2553 | 2416 | 1267 |
| *Compound* | 60 | 38 | 54 | 0 |
| *Other* | 4 | 36 | 72 | 12 |
| *Water* | 74 | 208 | 133 | 125 |
| RMS Bonds (Å) | 0.007 | 0.014 | 0.015 | 0.010 |
| RMS Angles (°) | 0.9 | 1.6 | 1.5 | 1.4 |
| Wilson B value (Å$^2$) | 30.0 | 18.7 | 32.6 | 35.3 |
| **Ramachandran plot** | | | | |
| Most favored (%) | 94.8 | 96.9 | 98.4 | 92.3 |
| Additional allowed (%) | 5.2 | 3.1 | 1.6 | 7.0 |
| Generously allowed (%) | 0.0 | 0.0 | 0.0 | 0.7 |
| Outliers (%) | 0.0 | 0.0 | 0.0 | 0.0 |

DOI: https://doi.org/10.7554/eLife.45403.010

post-SET motifs. The post-SET motif of **TC** was characterized by its U-shaped topology with a double-kinked loop-helix-helix architecture, which appears to be optimally oriented for binding both SAM and a peptide substrate (*Figures 1c* and *2*) (*Couture et al., 2005*; *Couture et al., 2008*; *Xiao et al., 2005*). In comparison, **BC-Inh1** and **BC-Inh2** rotate their post-SET motifs by 140° and 60°, respectively (*Figure 2*). Moreover, the post-SET motifs of **BC-Inh1** and **BC-Inh2** adopt more

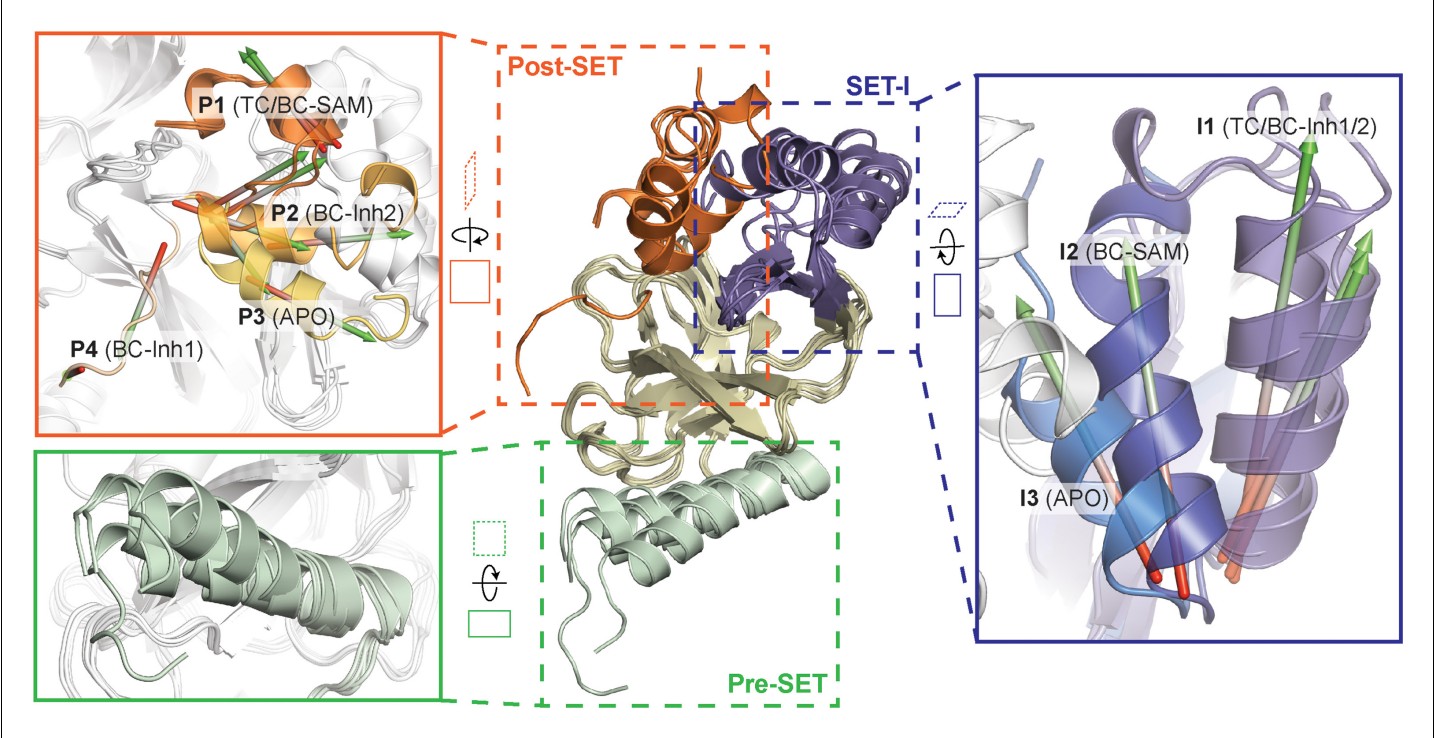

**Figure 2.** Superposition of five crystal structures highlighted with detailed views of post-SET, SET-I, and pre-SET motifs. The five X-ray structures reveal four distinct conformational states of the post-SET motif (**P1-4**) and three distinct conformational states of the SET-I motif (**I1-3**).
DOI: https://doi.org/10.7554/eLife.45403.009

extended configurations with a less structured loop and a singly-kinked helix, respectively (*Figures 1c* and *2*). Whereas multiple factors may influence the overall conformations, the formation of Cys311 adducts likely made the key contribution to the discovery of these hidden post-SET motif conformers.

To reveal additional hidden conformers that are structurally distinct from **TC**, we also solved crystal structures of SETD8 upon depleting native ligands and obtained structures of the SAM-SETD8 binary complex (**BC-SAM**) and apo-SETD8 (**APO**) (*Figure 1c*, *Figure 1—figure supplements 4* and *5*, *Table 1*). Strikingly, **BC-SAM** and **APO** differ from **TC** by their distinct SET-I motifs in the context of the otherwise similar SET-domain (*Figure 2*). Furthermore, the post-SET motif of **APO** structurally resembles an intermediate state between **BC-Inh1** and **BC-Inh2** but is distinct from those of **BC-SAM** and **TC** (*Figure 2*). In contrast to the structurally diverse SET-I (**I1-3**) and post-SET motifs (**P1-4**) in these structures, their pre-SET motifs show only slightly altered configuration (*Figure 2*). The differences between these structures highlight the conformational plasticity of the SET-I and post-SET motifs. Collectively, these observations provide strong structural rationale for the existence of a dynamic conformational landscape of SETD8.

## Hidden conformations of apo-SETD8 revealed by structural chimeras

The **BC-SAM**, **BC-Inh1**, **BC-Inh2**, **APO**, and **TC** structures can be readily classified into three distinct SET-I configurations (**I1-3**) and four distinct post-SET configurations (**P1-4**) (*Figure 2*). Given the relative spatial separation between the SET-I and post-SET motifs, we envisioned additional combinations of discrete motifs might be accessible to yet-unobserved conformations of SETD8. We thus constructed putative 'structural chimeras' of apo-SETD8 containing orthogonal **I1-3** and **P1-4** in a combinatorial (3 × 4) manner (*Figure 3a*, *Figure 3—figure supplement 1*). Among the twelve structural chimeras as potential seeds for MD simulations, five were crystallographically-determined conformers (**BC-Inh1**, **BC-Inh2**, **BC-SAM**, **TC** with ligands removed, and **APO**), four were new structurally-chimeric conformers, and three were excluded because of obvious steric clashes (*Figure 3a*, *Figure 3—figure supplement 2*). The four structurally-chimeric conformers were

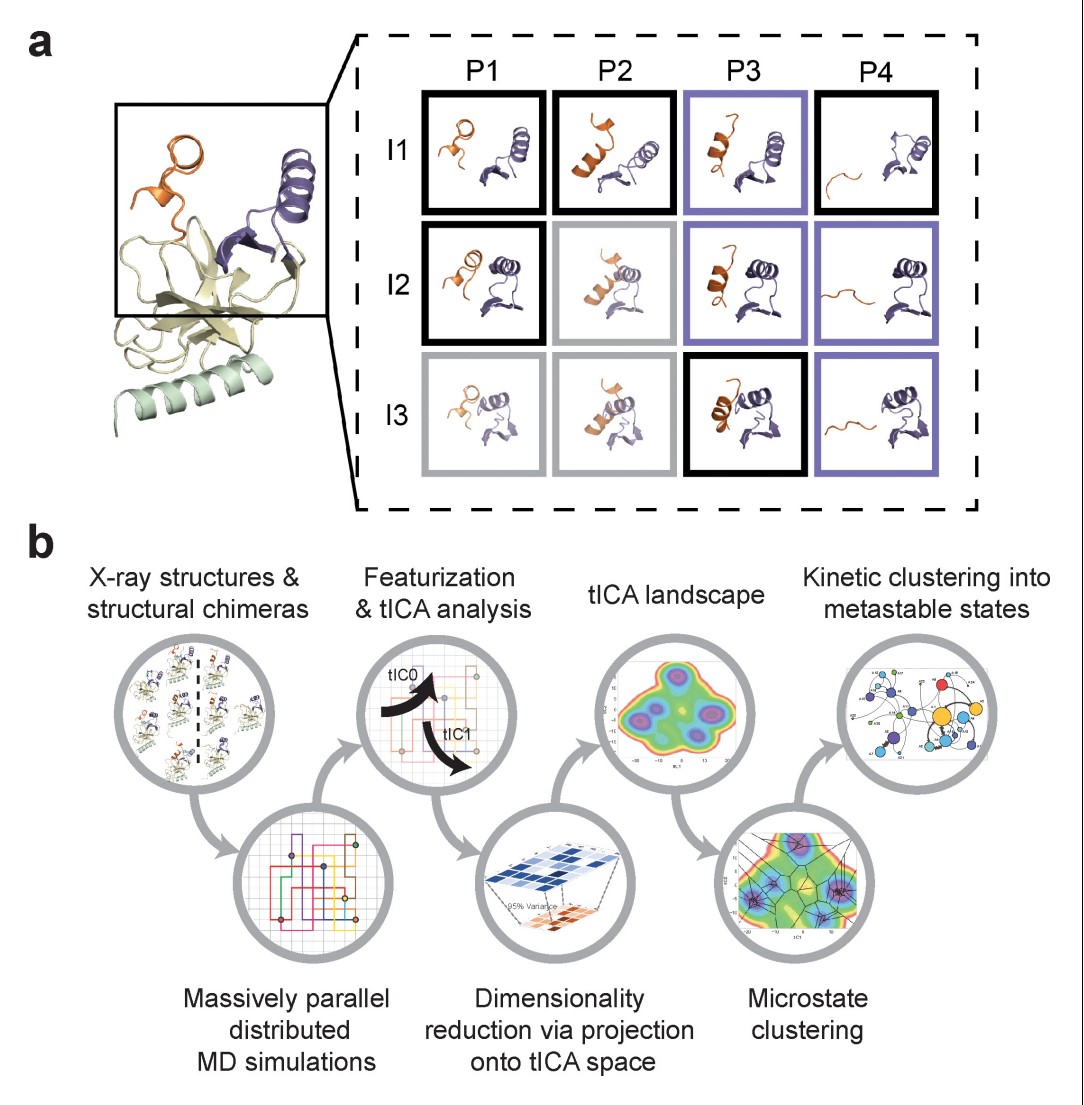

**Figure 3.** Construction of conformational landscapes of apo- and SAM-bound SETD8 through diversely seeded, parallel molecular dynamics simulations and Markov state models. (a) Combinatorial construction of putative structural domain chimeras using crystallographically-derived post-SET and SET-I conformations. Each conformer is boxed and color-coded with black for five X-ray-derived structures, blue for four putative structural chimeras included as seed structures for MD simulations, and gray for three structural chimeras excluded from MD simulations because of obvious steric clashes. (b) Schematic workflow to construct dynamic conformational landscapes via MSM. The five X-ray structures and the four structural chimeras were used to seed parallel MD simulations on Folding@home (see Materials and method). Markov state models were constructed from these MD simulation results to reveal the conformational landscape.

DOI: https://doi.org/10.7554/eLife.45403.011

The following figure supplements are available for figure 3:

**Figure supplement 1.** Workflow of MD simulations and MSM analysis.

DOI: https://doi.org/10.7554/eLife.45403.012

**Figure supplement 2.** Cartoon representations of the four structural chimeras.

DOI: https://doi.org/10.7554/eLife.45403.013

included to seed MD simulation with the intention to uncover the conformational landscape more effectively, although this operation proved to be redundant for the discovery of new conformations in the validation process (see details below).

## Dynamic conformational landscape of apo-SETD8 via Markov state modeling from 5 ms MD simulation dataset

With seed conformations prepared as above, we envisioned illuminating the conformational landscape with distributed long-timescale MD simulations and resolving its kinetic features with Markov state models (MSMs) (*Figure 3b*, *Figure 3—figure supplement 1*). Because there is no prior report of the conformational landscapes of PKMTs that can be used as the reference of SETD8, we leveraged extensive computational power for MD simulation with the intention to not only uncover the conformational landscape of SETD8 in an unbiased manner but also cross-validate the completeness of the dataset. Here we conducted approximately $500 \times 1$ μs explicit-solvent MD simulations from each seed and accumulated 5 milliseconds of aggregate data in 10 million conformational snapshots for apo-SETD8 (*Appendix 1—figure 1*, *Supplementary file 1a*). To identify functionally relevant conformational states and their transitions, we built MSMs using a pipeline that employs machine learning and extensive hyperparameter optimization to identify slow degrees of freedom and structural and kinetic criteria to cluster conformational snapshots into discrete conformational states (*Appendix 1—figures 2–9*, *Supplementary file 1b, 1c*) (*Husic et al., 2016*). This approach identified 24 kinetically metastable conformations (macrostates) from an optimized, cross-validated set of 100 microstates (*Figure 4a*, *Figure 5—figure supplement 1*, *Supplementary file 1d, 1e*). These macrostates are remarkably diverse, spanning up to 10.5 Å Cα RMSD from **APO**. To visualize the kinetic relationships between functionally important conformations, dimensionality reduction was used to project the landscape into 2D while preserving log inverse fluxes between states (*Figure 4b*). The relative populations of these macrostates were also calculated, resolving rare conformational states up to 6 kT in free energy (*Figures 4b* and *5a*).

## The dynamic conformational landscape of SAM-bound SETD8

Given the success in constructing the dynamic conformational landscape of apo-SETD8, we applied the same strategy to SAM-bound SETD8. With the two crystal structures of SETD8 in complex with SAM (**BC-SAM** and **TC**) as the seed conformations, we conducted ~$500 \times 1$ μs explicit solvent MD simulations from each structure and accumulated 1 millisecond of aggregate data (2M snapshots) (*Appendix 1—figure 10*). The MSM of the conformational landscape of SAM-bound SETD8 was constructed using the same degrees of freedom as that of apo-SETD8 to facilitate direct comparison of the models (*Appendix 1—figures 11-13*). The resulting MSM for SAM-bound SETD8 contained 10 kinetically metastable macrostates arising from 67 microstates (*Figure 5—figure supplement 2*, *Supplementary file 1f, 1g*). Similar to those of apo-SETD8, the relative macrostate populations of SAM-bound SETD8 and their flux kinetics were computed and embedded into 3D/2D scatter plots and a chord diagram (*Figures 4a, b* and *5b*). The smaller number of metastable states identified for SAM-bound SETD8 is anticipated given that specific conformations are required for optimal interaction between SAM and SETD8's post-SET motif (*Couture et al., 2005*; *Couture et al., 2008*; *Xiao et al., 2005*). We also compared the timescale structure of the apo- and SAM-bound SETD8 MSMs, as well as an MSM constructed from the subset of apo-SETD8 trajectories originating from the same conformations as the SAM-bound trajectories (*Appendix 1—figure 14*). We found a large decrease in the number of slow processes seen in the SAM-bound model compared to the other two (respectively for the apo, SAM-bound, and subset of apo MSMs there are 14, 4, and 9 processes slower than 1 μs). SAM binding thus restricts overall conformational accessibility of SETD8.

## Experimental corroboration of the conformational landscapes of SETD8

Upon uncovering the dynamic conformational landscapes of apo- and SAM-bound SETD8 for the first time of the PKMT family of enzymes, we were able to extract new structural information and designed experiments to further examine this model (*Figure 6*). Comparison of the conformational ensembles between apo- and SAM-bound SETD8 revealed that SAM binding dramatically alters the environment of Trp390 (*Figure 6a*, blue sticks), the sole tryptophan residue in the catalytic domain of SETD8. This residue is flexible and mainly solvent-exposed in apo-SETD8 conformational ensembles but restricted in a hydrophobic environment through SAM-mediated pi-pi stacking in SAM-bound SETD8 conformational ensembles (*Figure 6a*). Such environmental changes upon SAM binding are expected to quench fluorescence of Trp390 (*Royer, 2006*). To verify this prediction, we designed rapid-mixing stopped-flow kinetic experiments with 5 ms dead time and 0.1 ms resolution

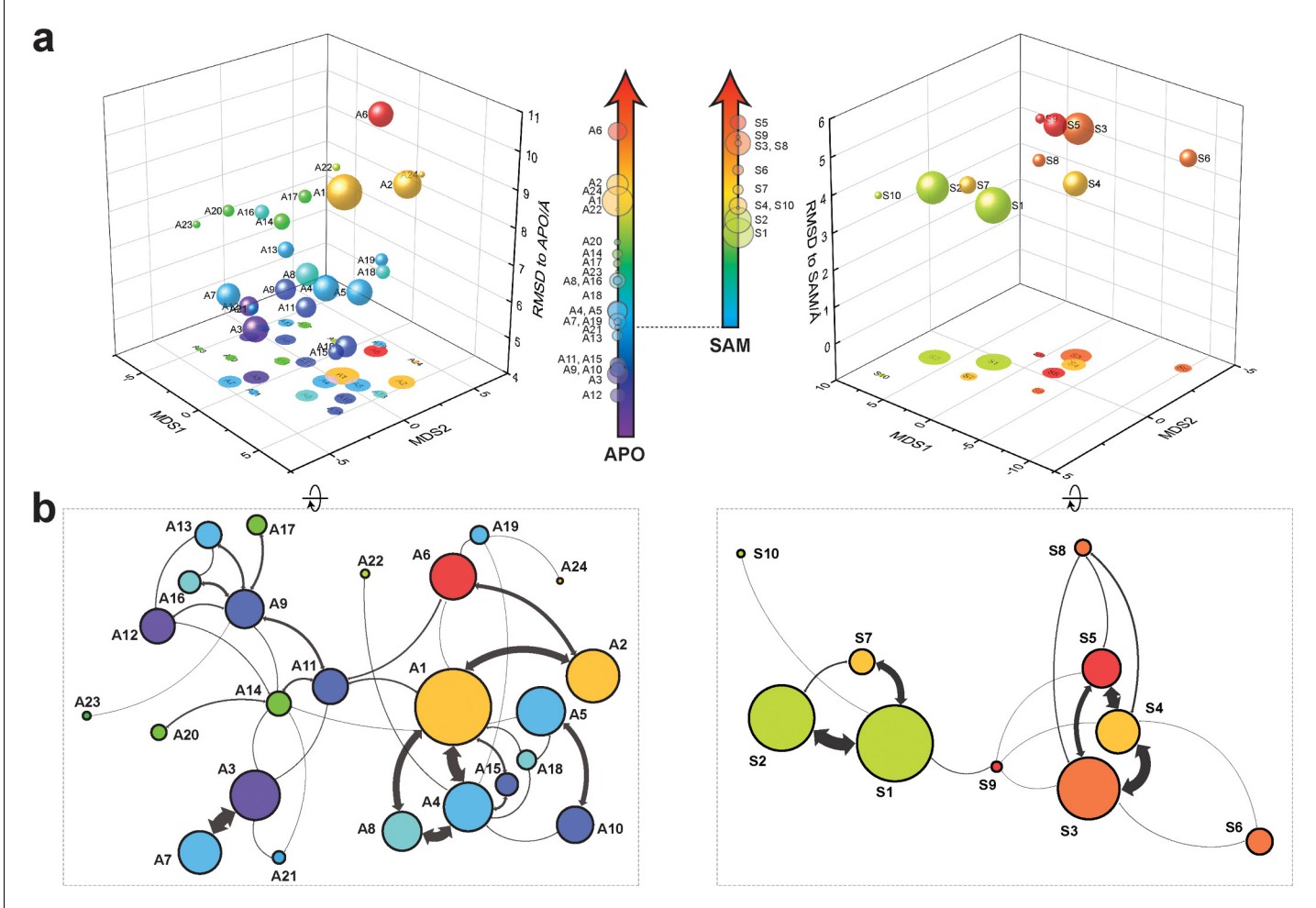

**Figure 4.** Markov state models and conformational landscapes of apo- and SAM-bound SETD8. Kinetically metastable conformations (macrostates) obtained from kinetically coupled microstates via Hidden Markov Model (HMM) analysis. The revealed dynamic conformational landscapes consist of 24 macrostates for apo-SETD8 (left panel) and 10 macrostates for SAM-bound SETD8 (right panel). (a) Kinetic and structural separation of macrostates in a 3D scatterplot. The MDS1/MDS2 axes are the two top vectors used in multidimensional scaling (MDS), a dimensionality reduction method, for separation of macrostates via log-inverse flux kinetic embedding (see Materials and methods). The Z axis reports root-mean-square deviations (RMSDs) of each macrostate to **APO** (left) or **BC-SAM** (right). The relative population of each macrostate of apo- or SAM-bound SETD8 ensembles is proportional to the volume of each representative sphere. (b) Cartoon depiction of macrostates in a 2D scatterplot. The diameter of the corresponding circle in the 2D scatterplot is proportional to the diameter of the respective sphere in the 3D scatterplot above. Equilibrium kinetic fluxes larger than $7.14 \times 10^2$ s$^{-1}$ for apo- and $1.39 \times 10^3$ s$^{-1}$ for SAM-bound SETD8 are shown for interconversion kinetics with thickness of the connections proportional to fluxes between two macrostates.

DOI: https://doi.org/10.7554/eLife.45403.014

to track the fluorescence change of Trp390 upon SAM binding (*Figure 6b*). We observed SAM-dependent biphasic kinetics of the fluorescence decrease within 1 s with >80% of the change occurring in the fast phase (0–0.1 s) (*Figure 7a*). In the context of the conformational landscape of apo-SETD8, we interpreted the major decrease in fluorescence intensity (fast-phase kinetics) as a consequence of the collective changes of Trp390 from the solvent-exposed hydrophilic environment in apo conformations to the hydrophobic environment in SAM-bound conformations (*Figure 6a*). In contrast, the minor decrease in fluorescence intensity (slow-phase kinetics) reflects the slow conformational changes of Trp390 in the SAM-bound SETD8 conformational ensembles (*Figure 7a*). With unsupervised global fitting to this two-step model, we obtained forward and reverse rate constants for the fast- and slow-phase kinetics, which are in agreement with conventional fitting to double exponential kinetics (*Johnson, 1992*) (*Figures 7a, b* and *8a*, *Figure 7—figure supplement 1*,

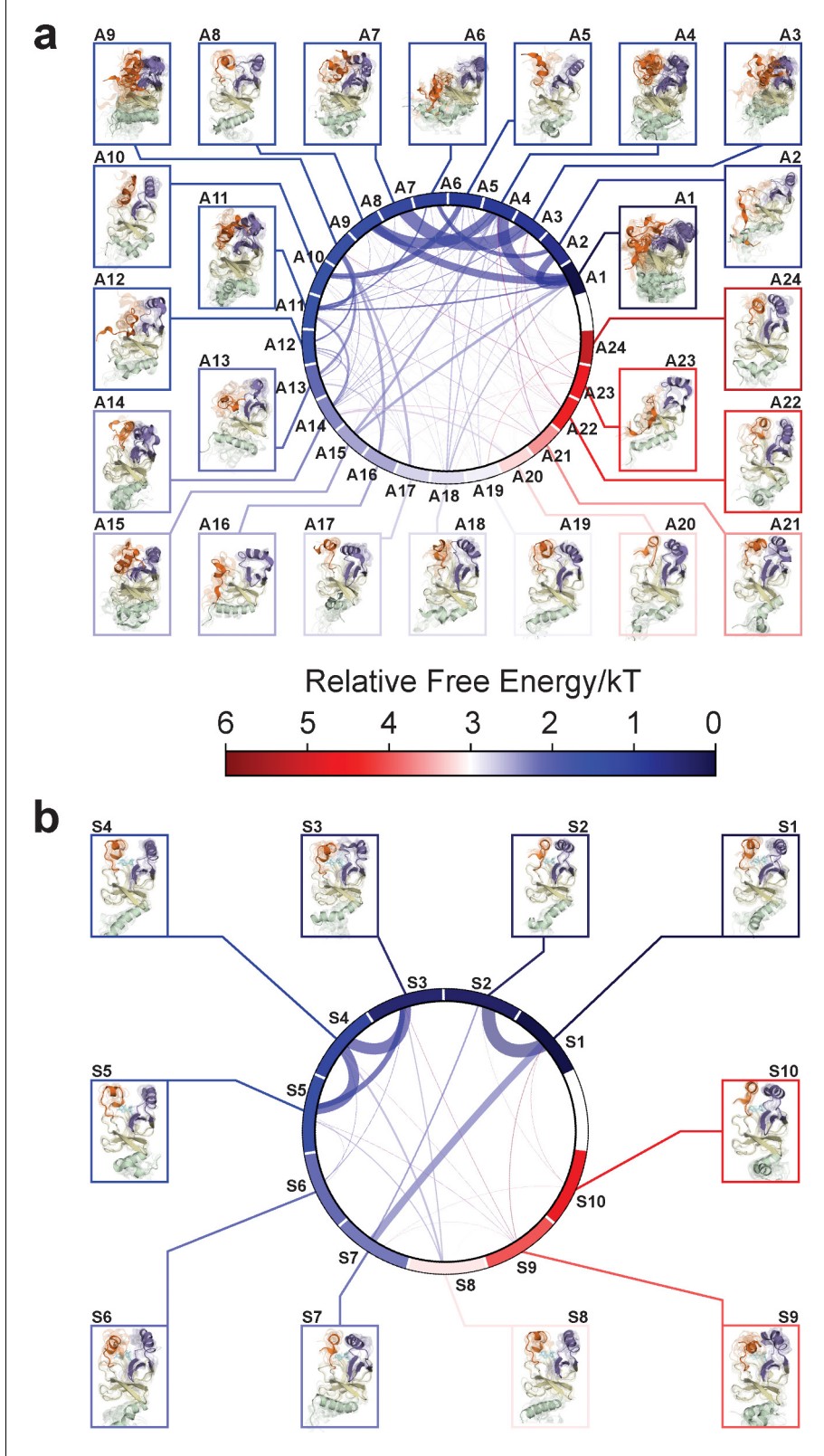

**Figure 5.** Chord diagrams and representative conformers of macrostates. The colors represent the free energy of each macrostate relative to the lowest free energy macrostate. The equilibrium flux between two macrostates is proportional to thickness of connecting arcs.

DOI: https://doi.org/10.7554/eLife.45403.015

*Figure 5 continued on next page*

*Figure 5 continued*

The following figure supplements are available for figure 5:

**Figure supplement 1.** Representative conformations of macrostates in the conformational landscape of apo-SETD8.

DOI: https://doi.org/10.7554/eLife.45403.016

**Figure supplement 2.** Representative conformations of macrostates in the conformational landscape of SAM-bound SETD8.

DOI: https://doi.org/10.7554/eLife.45403.017

*Supplementary file 1h*). The $k_{-1}$ value was also confirmed independently by rapid-mixing stopped-

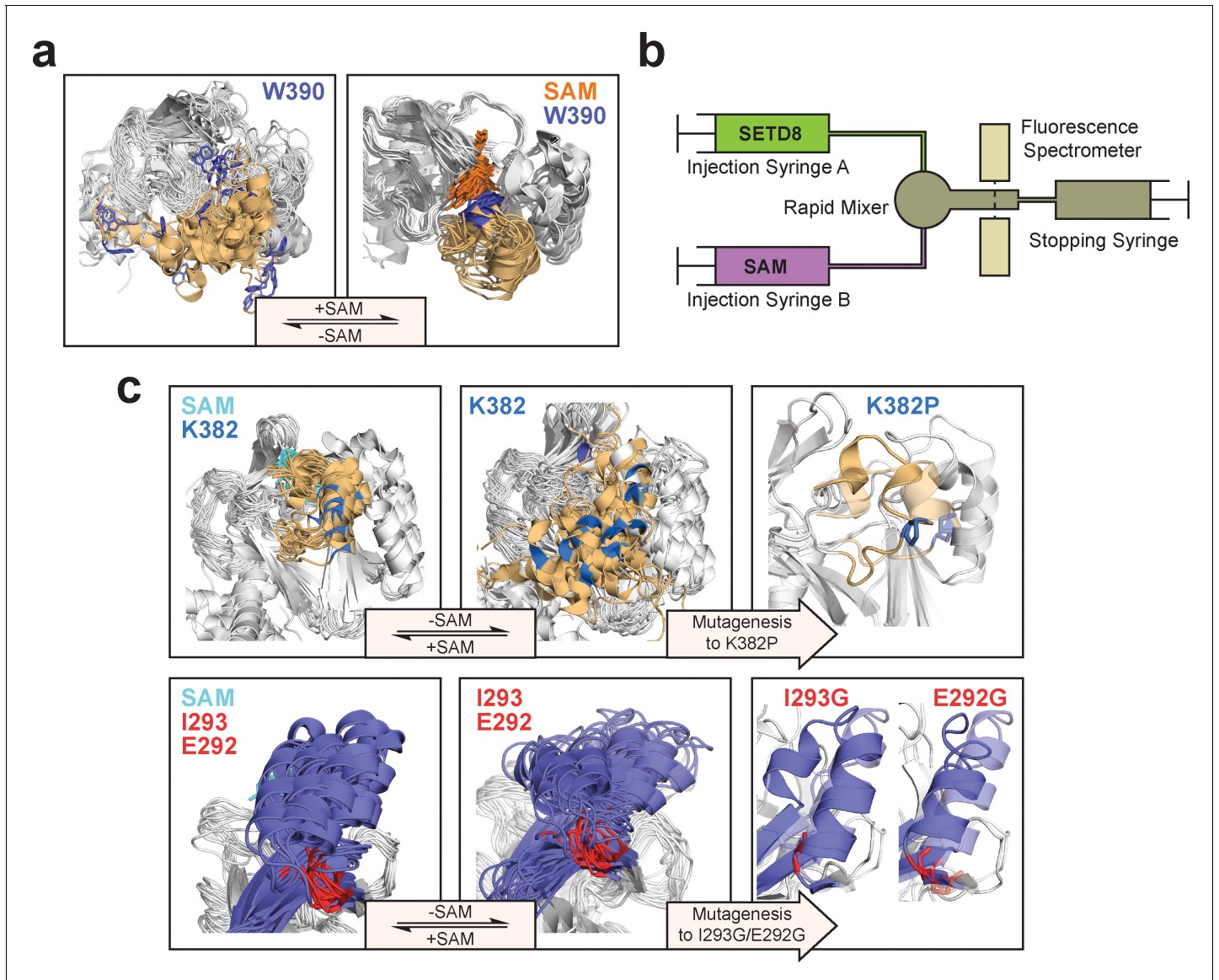

**Figure 6.** Experimental design to probe the conformational landscape of SETD8. (**a**) Comparison of binding environments of Trp390 (blue) between apo and SAM-bound (orange) SETD8 in the context of their dynamic conformational landscapes. (**b**) Illustration of rapid-quenching stopped-flow experiments. These experiments were conducted to trace fluorescence changes of Trp390 upon SAM binding. (**c**) Comparison of the conformations of post-SET kink and SET-I helix between apo- and SAM-bound SETD8 in the context of their dynamic conformational landscapes. Analysis of key structural motifs indicated K382P (blue in the upper panel), I293G and E292G (red in the lower panel) as potential gain-of-function variants.

DOI: https://doi.org/10.7554/eLife.45403.018

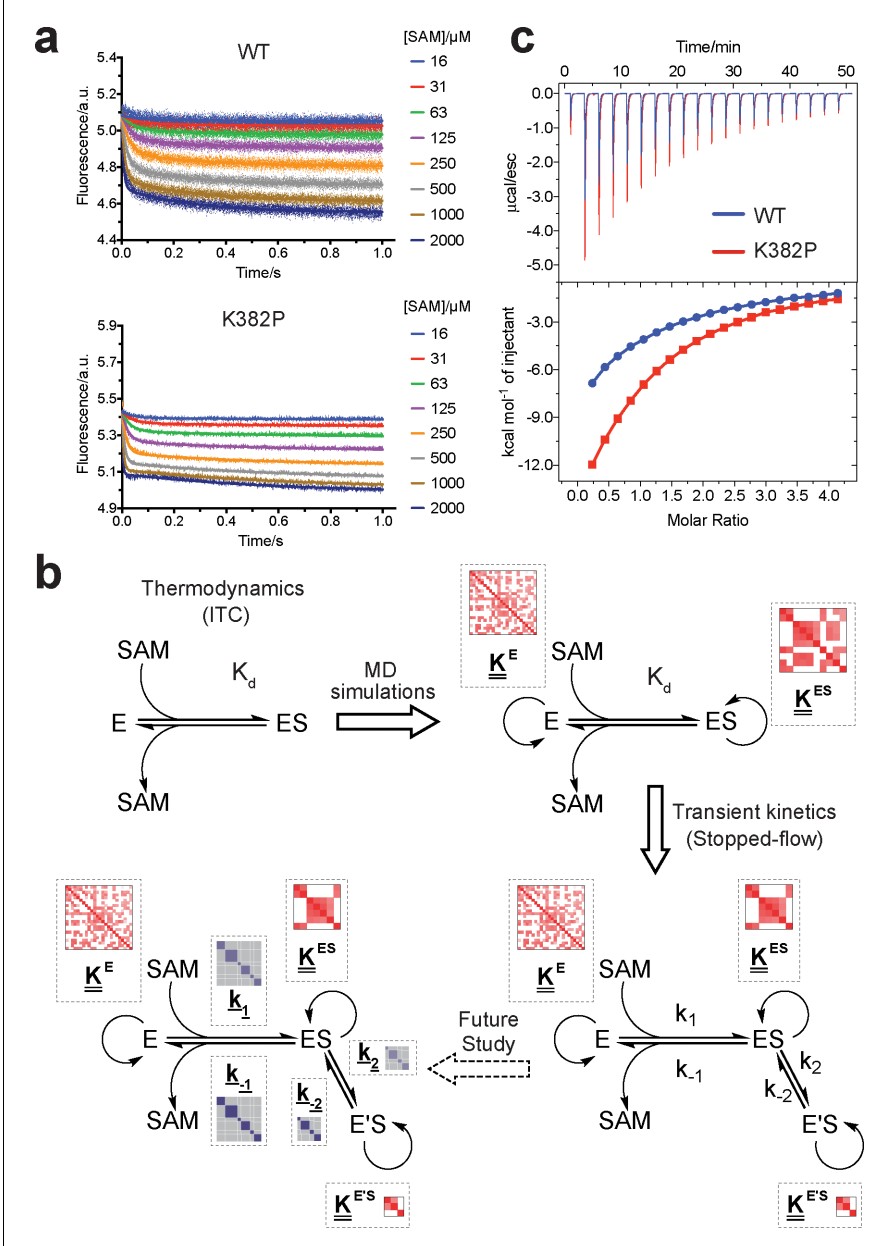

**Figure 7.** Biochemical characterization of gain-of-function mutations revealed by conformational landscapes of SETD8. (a) Fluorescence changes of wild-type and K382P SETD8 traced with a rapid-quenching stopped-flow instrument within 1 s upon SAM binding. (b) Stepwise SAM-binding of SETD8 in the integrative context of biochemical, biophysical, structural, and simulation data. ITC determines the thermodynamic constant of SAM binding by SETD8. MD simulations and MSM uncover metastable conformations and interconversion rates of apo- and SAM-bound SETD8 ($\underline{K}^{apo}$ and $\underline{K}^{SAM}$). Stopped-flow experiments revealed that SETD8 binds SAM via biphasic kinetics. Rate constants uncovered by stopped-flow experiments ($k_1$, $k_{-1}$, $k_2$, $k_{-2}$) represent macroscopic rates of SAM binding by SETD8 with multiple metastable conformations. The microscopic behavior of individual metastable states and corresponding rates ($\underline{k}_1$, $\underline{k}_{-1}$, $\underline{k}_2$, $\underline{k}_{-2}$) have not been resolved. Transition probability matrices (red) and microscopic rate constant matrices (blue) are shown as colored grids. A rigorous mathematical derivation of this scheme is shown in *Figure 7—figure supplement 3*. (c) ITC enthalpogram for the titration of SAM into wild-type and K382P SETD8.

DOI: https://doi.org/10.7554/eLife.45403.019

The following figure supplements are available for figure 7:

*Figure 7 continued on next page*

*Figure 7 continued*

**Figure supplement 1.** Rapid-mixing stopped-flow experiments of SAM-binding and double-exponential conventional fitting analysis.
DOI: https://doi.org/10.7554/eLife.45403.020
**Figure supplement 2.** Isothermal Titration Calorimetry (ITC) of wild-type SETD8 and its mutants in complex with SAM.
DOI: https://doi.org/10.7554/eLife.45403.021
**Figure supplement 3.** Rigorous derivation of stepwise, microscopic resolution of SETD8 SAM-binding kinetics.
DOI: https://doi.org/10.7554/eLife.45403.022

flow dilution of SAM-bound SETD8 (*Agafonov et al., 2014*) ('ES +E'S', *Appendix 1—figure 15*, *Supplementary file 1h*). Here the $k_{-1}/k_1$ ratio (*Figure 7b*) of $309 \pm 6$ µM corresponds to the average SAM dissociation constant $K_{d1}$ of apo-SETD8 conformers, which is consistent with independently determined ITC $K_d$ of $251 \pm 16$ µM (*Figure 7b and c*, *Figure 7—figure supplement 2*). In contrast, the large $k_{-2}/k_2$ ratio (*Figure 7b*) of $30 \pm 11$ suggests that the second phase corresponds to a slow equilibrium between ES and E'S with minimal contribution of E'S to the overall SAM dissociation constant $K_d$ (*Figure 7c*). The conformational ensembles we identified for apo- and SAM-bound SETD8 demonstrate the statistical nature of its SAM-binding process. Therefore, the observed fluorescence changes and herein determined macroscopic kinetic constants represent an ensemble-weighted average of microscopic behaviors of all species that exist in the solution. A rigorous mathematical description of microscopic kinetics of SAM binding was thus obtained under the consideration of interconversion of the metastable conformational states of apo- and SAM-bound SETD8 (*Figure 7—figure supplement 3*).

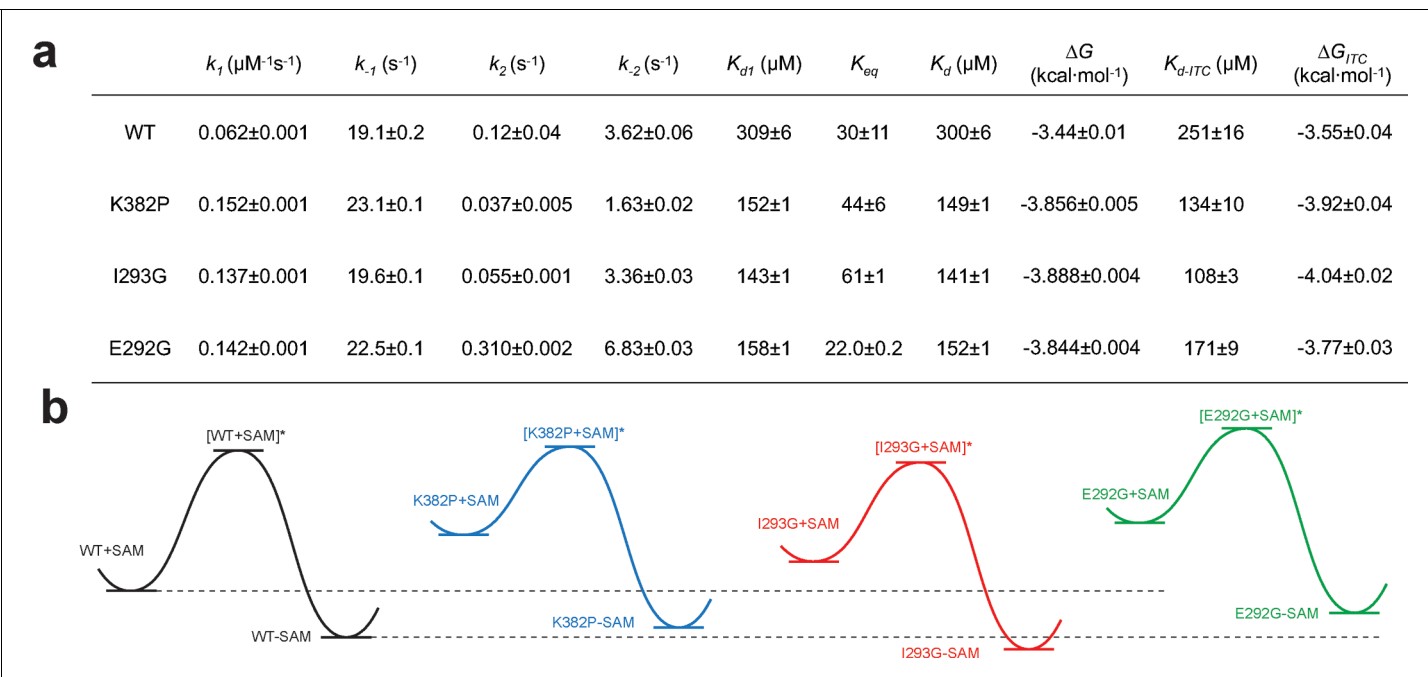

| | $k_1$ (µM⁻¹s⁻¹) | $k_{-1}$ (s⁻¹) | $k_2$ (s⁻¹) | $k_{-2}$ (s⁻¹) | $K_{d1}$ (µM) | $K_{eq}$ | $K_d$ (µM) | $\Delta G$ (kcal·mol⁻¹) | $K_{d\text{-}ITC}$ (µM) | $\Delta G_{ITC}$ (kcal·mol⁻¹) |
|---|---|---|---|---|---|---|---|---|---|---|
| WT | 0.062±0.001 | 19.1±0.2 | 0.12±0.04 | 3.62±0.06 | 309±6 | 30±11 | 300±6 | -3.44±0.01 | 251±16 | -3.55±0.04 |
| K382P | 0.152±0.001 | 23.1±0.1 | 0.037±0.005 | 1.63±0.02 | 152±1 | 44±6 | 149±1 | -3.856±0.005 | 134±10 | -3.92±0.04 |
| I293G | 0.137±0.001 | 19.6±0.1 | 0.055±0.001 | 3.36±0.03 | 143±1 | 61±1 | 141±1 | -3.888±0.004 | 108±3 | -4.04±0.02 |
| E292G | 0.142±0.001 | 22.5±0.1 | 0.310±0.002 | 6.83±0.03 | 158±1 | 22.0±0.2 | 152±1 | -3.844±0.004 | 171±9 | -3.77±0.03 |

**Figure 8.** Kinetic and thermodynamic constants of wild-type SETD8 and its rationally designed mutants. For $k_1$, $k_{-1}$, $k_2$, $k_{-2}$ in *Figure 7*, data are best fitting values ± standard error (s.e.) from KinTek. For $K_{d\text{-}ITC}$, data are mean ±s.e. of at least three replicates. $K_{d1}$, $K_{eq}$, and $K_d$ are calculated based on equations in Methods. Uncertainties of $K_{d1}$, $K_{eq}$, $K_d$, and $\Delta G$ are s.e. calculated by the propagation of uncertainties from individual rate constants and dissociation constants, respectively. h, Relative energy landscapes of apo- and SAM-bound SETD8 and its gain-of-function mutants. The relative energy of apo- and SAM-bound (wildtype and mutated) SETD8 as well as their transition states were determined on the basis of their $k_1$, $k_{-1}$, and $K_d$ values. The relative position of each energy landscape was then set on the basis of the rough counts of mutation-associated loss or gain of favorable interactions in contrast to apo- or SAM-bound wild-type SETD8. All SETD8 variants except SAM-bound I293G disrupt the favorable interactions to various degrees.
DOI: https://doi.org/10.7554/eLife.45403.023

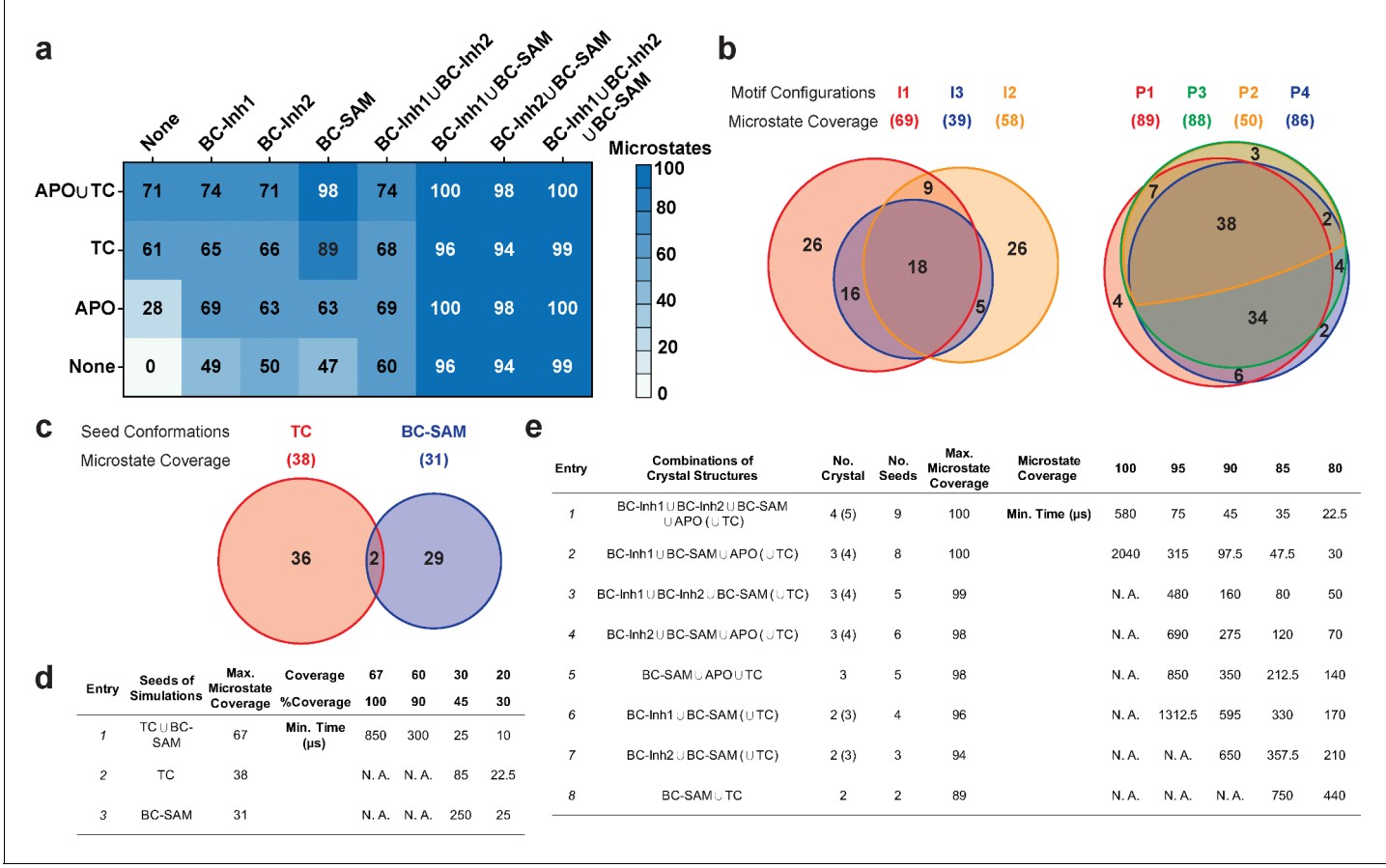

**Figure 9.** Evaluation of key simulation parameters of molecular simulations. (**a**−**b**) Assessments of simulations of apo-SETD8: (**a**) Heat map for the coverage of the 100 microstates with all combinations of the crystal structures (BC-Inh1, BC-Inh2, BC-SAM, APO, and TC) as seed conformations; (**b**) Venn diagrams of the coverage of the 100 microstates with all conformational combinations of SET-I and post-SET motifs (I1-3 and P1-4) as seed structures for MD simulations. (**c**−**d**) Robustness of simulations of SAM-bound SETD8: (**c**) Venn diagram of the coverage of the 67 microstates with TC, BC-SAM or both as seed structures for MD simulation; (**d**) Minimal time required by MD simulations to reach certain coverage of the 67 microstates of SAM-bound SETD8 with representative combinations of seed structures. (**e**) Minimal time required by MD simulations to reach certain coverage of the 100 microstates of apo-SETD8 with representative combinations of seed structures.

DOI: https://doi.org/10.7554/eLife.45403.024

We then proposed to confirm our understanding of functionally-relevant conformations and their thermodynamics by identifying SETD8 variants with increased affinity for SAM. We uncovered a collection of characteristic kink motifs around Lys382 in the post-SET motif of SAM-bound SETD8 conformational ensembles (*Figure 6c*), while this region is less structured in apo-SETD8 conformational ensembles. We hypothesized that a proline mutation (K382P) could better stabilize the conformational ensembles of SAM-bound SETD8 than apo-SETD8 (*Figures 6c* and *8b*). We also considered the characteristic α-helix in the SET-I motif, which adopts flexible and diverse configurations in the apo ensembles but becomes constrained and elongated in SAM-bound ensembles (*Figure 6c*). We proposed that the replacement of I293 or E292 adjacent to the α-helix with a flexible glycine should relax this distortion to better stabilize SAM-bound ensembles (*Figures 6c* and *8b*). We therefore characterized the SAM-binding kinetics and affinities of K382P, I293G, and E292G variants of SETD8 with stopped-flow kinetics and ITC (*Figures 6c*, *7a, b and c*, *Figure 7—figure supplements 1* and *2*, *Appendix 1—figure 15*). While exhibiting biphasic kinetics similar to that of wild-type SETD8, the stopped-flow mixing experiment revealed the three variants showed a significant two-fold decrease of $K_{d,SAM}$ (*Figures 7a, c* and *8a*). The stopped-flow data further revealed that the two-fold change of $K_{d,SAM}$ mainly arises from increased SAM-binding rates $k_1$ with relatively unchanged $k_{-1}$ (*Figure 8a*). These results are consistent with independently determined $K_d$ and $k_{-1}$ from ITC and stopped-flow

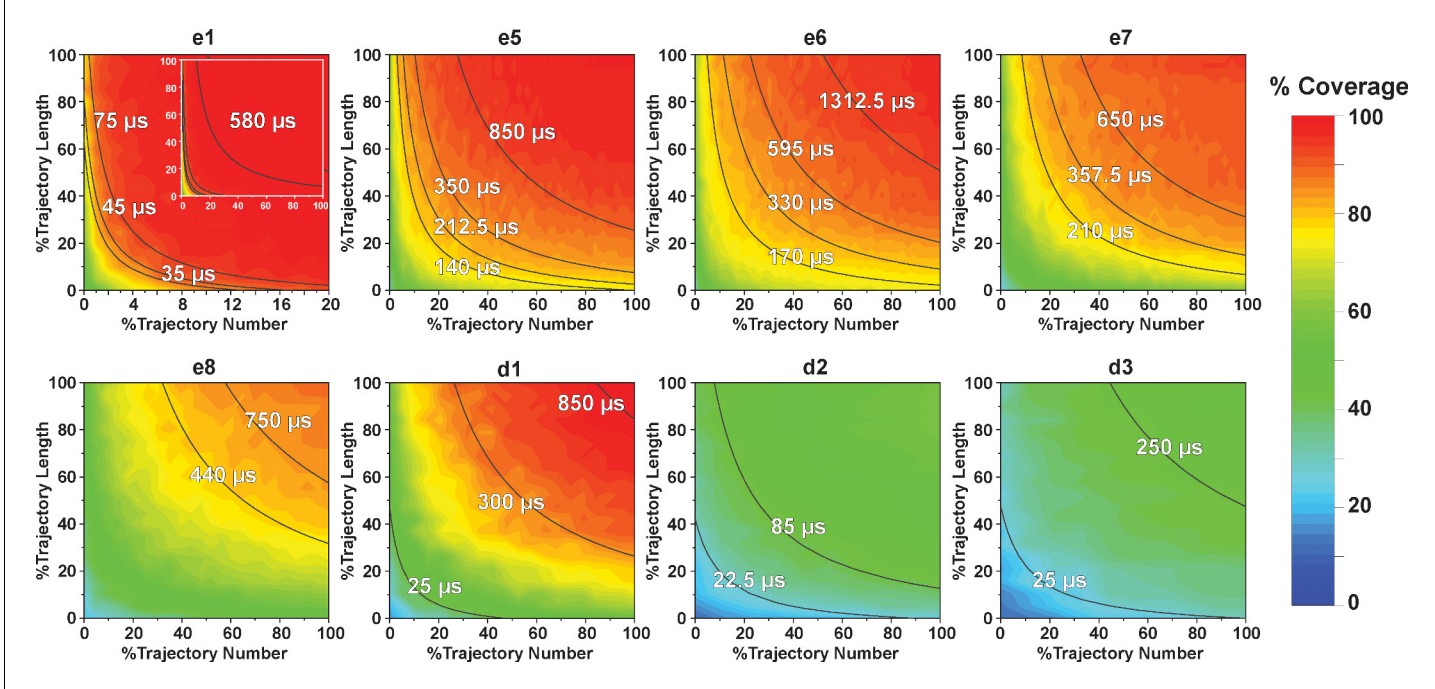

**Figure 10.** Contour map of microstate coverage at various combinations of trajectory lengths and numbers as percentage of the maximal trajectory length and number of MD simulations. The seed structures of each panel are listed as the simulation entries e1, e5−8 for apo-SETD8, and d1−3 for SAM-bound SETD8 in *Figure 9d and e*. Each curve corresponds to the aggregation of specific simulation time.

DOI: https://doi.org/10.7554/eLife.45403.025

The following figure supplement is available for figure 10:

**Figure supplement 1.** Contour maps presenting microstate coverage at various trajectory lengths and numbers versus the maximal possible trajectory length or number at different combinations of starting conformations.

DOI: https://doi.org/10.7554/eLife.45403.026

dilution, respectively (*Figure 7b and c*, *Figure 7—figure supplement 2*, *Appendix 1—figure 15*, *Supplementary file 1h*). Collectively, these observations confirm the robustness of our conformational landscape model for apo- and SAM-bound SETD8.

## Effects of key simulation parameters on construction of conformational landscapes

We systematically investigated how the choices of seed structures and simulation time—key computational parameters—influence microstate discovery and quality of conformational landscapes of SETD8 (*Figures 9* and *10*). The simulations of apo-SETD8 initiated from any single X-ray structure (**BC-Inh1**, **BC-Inh2**, **BC-SAM**, **APO**, or **TC** in *Figure 1c*) only reveal a partial conformational landscape (28–61% microstate coverage, *Figure 9a*, *Supplementary file 1i*). To achieve >90% microstate coverage, at least two crystal structures—**BC-SAM** in combination with either **BC-Inh1** or **BC-Inh2**—must be included (*Figure 9a*). If three crystal structures are included, **BC-SAM** in combination with **TC** and **APO** can provide >90% coverage (*Figure 9a*). In terms of the structural motifs (**I1-3** or **P1-4**, *Figures 2* and *3a*), simulations originating from the SET-I motif **I1**, **I2**, or **I3** alone led to the discovery of 69, 58, or 39 of the 100 microstates, respectively (*Figure 9b*, *Supplementary file 1j*). The combination of **I1** and **I2** is sufficient to cover all 100 microstates, arguing for the redundant character of **I3**. For the post-SET motif, any combination of two post-SET configurations except **P2**∪**P3** leads to >90 microstate coverage (*Figure 9b*, *Supplementary file 1j*). These findings are in agreement with the key requirement of structural motif conformations **I1** (equivalent to **BC-Inh1**, **BC-Inh2**, or **TC**), **I2** (equivalent to **BC-SAM**), and any two of **P1−4** except **P2**∪**P3** (*e.g.* **P1**∪**P3** is equivalent to the combination of **APO** with **BC-SAM** or **TC**) to achieve >90% microstate coverage. For SAM-bound SETD8, the seed conformations derived from **BC-SAM** and **TC** structures contribute 31 and 38 of 67

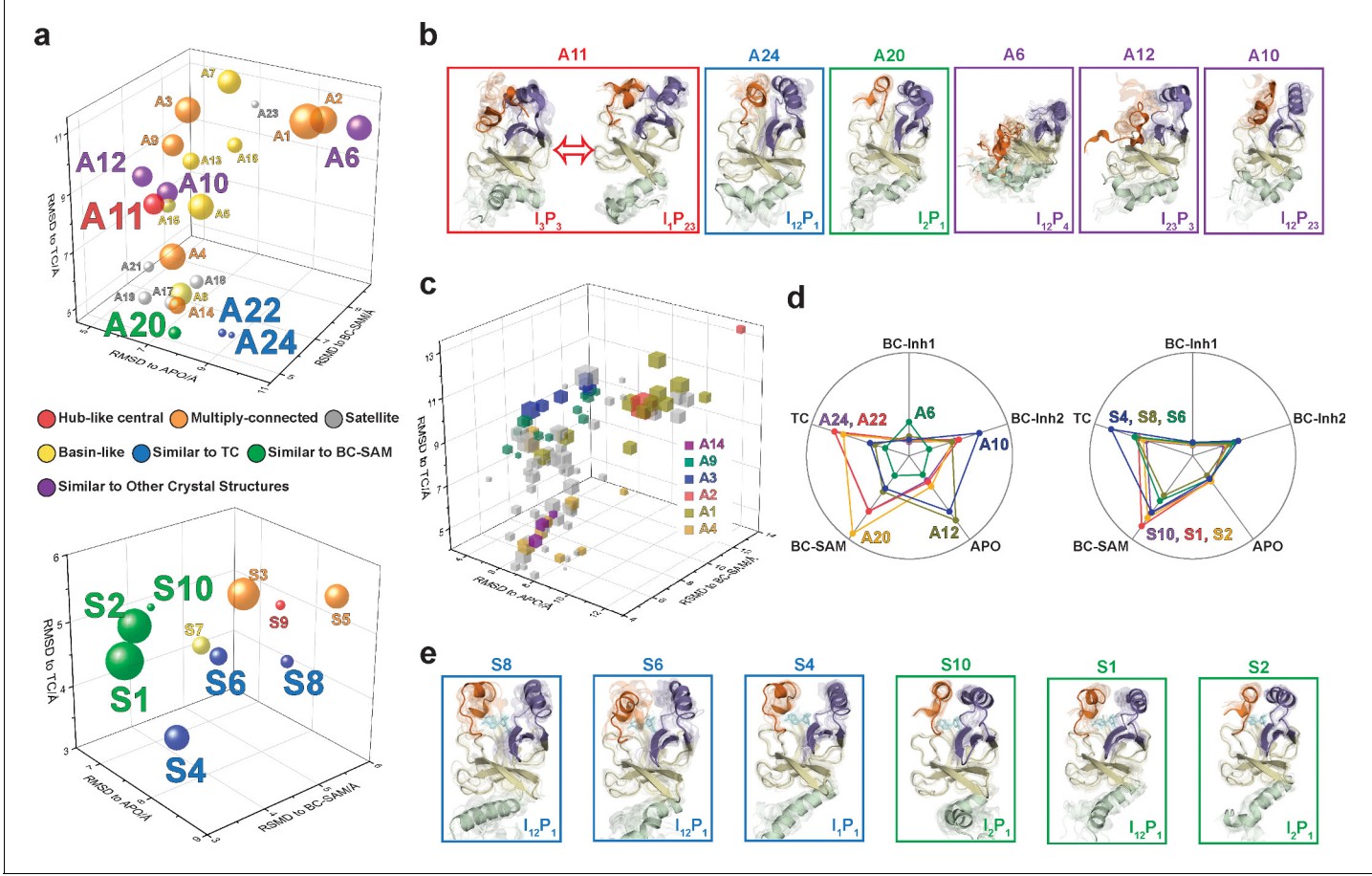

**Figure 11.** Functional annotation of the dynamic conformational landscapes of SETD8. (**a**) 3D scatterplots of the 24 macrostates of apo-SETD8 landscape and 10 macrostates of SAM-bound SETD8 landscape in the coordinates of RMSDs relative to **APO**, **BC-SAM**, and **TC**. Volume of each sphere is proportional to the relative population of the corresponding macrostate in the context of the 24 macrostates for apo-SETD8 or the 10 macrostates for SAM-bound SETD8. The RMSD of each macrostate is the average of its microstates weighted with their intra-macrostate populations. The RMSD of each microstate is the average of the top 10 frames most closely related to the clustering center of the microstate. The feature of each macrostate is annotated in color. (**b**) Cartoons of representative conformations of key macrostates in the apo-SETD8 landscape. Structural annotations are shown in bottom right of each conformation. (**c**) 3D scattering plot of 100 microstates of the apo landscape in the coordinates of RMSDs to **APO**, **BC-SAM**, and **TC**. Volume of each cube is proportional to the relative population of the corresponding microstate in the context of the 100 microstates. Microstates clustered in intermediate-like macrostates are highlighted in colors. Structural diversity of microstates within individual macrostates indicates that each intermediate-like state contains multiple structurally distinct but readily interconvertible microstates. (**d**) Radar chart of representative macrostates of apo (left) and SAM-bound (right) landscapes in reference to the five crystal structures. Distances between dots and cycle centers are proportional to the reciprocal values of RMSDs of macrostates relative to the crystal structures. (**e**) Cartoons of representative conformations of key macrostates in the SAM-bound SETD8 landscape. Structural annotations are shown in bottom right of each conformation.

DOI: https://doi.org/10.7554/eLife.45403.027

microstates (*Figure 9c and d*, *Supplementary file 1k*). These findings argue for the importance of using multiple structures to construct the landscape within achievable computer time. The seed conformations prepared from ligand-trapped SETD8 structures are essential to discovering the complete conformational landscapes of SETD8.

For simulation time, we observed that the fewer seed conformations of apo-SETD8 were employed, the more computing power (the product between the number of simulation trajectories and the time length per trajectory) was required to reach a comparable level of microstate coverage (*Figure 9e*, *Supplementary file 1l, 1m*). When computing power is fixed, comparable microstate coverages of apo- and SAM-bound SETD8 can be obtained by running either multiple short trajectories or few long trajectories (*Figure 10*, *Figure 10—figure supplement 1*). While the current aggregate simulation time (5 ms for apo-SETD8 and 1 ms for SAM-bound SETD8) appears sufficient to

discover essentially all relevant conformations in the landscapes of SETD8 and estimate their relative populations and corresponding uncertainties, more data would yet be needed to improve estimates of inter-macrostate kinetics in order to develop a fully kinetically accurate model (*Supplementary file 1h, 1l*). Collective contributions of the number of seed structures and the over-all simulation time determined the efficiency of uncovering conformational landscapes of SETD8. The conformational landscape of apo-SETD8 can be revealed upon implementing a minimum of two seed structures (**TC** and **BC-SAM**) or 10% of the current simulation time. With the two seed struc-tures (**TC** and **BC-SAM**) and sufficient simulation time, apo-SETD8 sampled 22 more microstates than SAM-bound SETD8 (89 states with 750 μs simulation versus 67 states with 850 μs simulation, *Figure 9d,e*), consistent with the conformational restriction of SETD8 upon SAM binding. We also noted that it is redundant to include the four structurally-chimeric conformers because this operation contributes less than 10% of microstate coverage and the comparable conformational landscape of apo-SETD8 can be generated with the subsets of seeds solely prepared from the X-ray structures (*Supplementary file 1l*).

## Functionally relevant conformations in the dynamic landscapes of apo- and SAM-bound SETD8

After experimentally corroborating the conformational landscapes of apo- and SAM-bound SETD8, we explored the dynamic details of these landscapes with the focus on the connectivity and equilib-rium fluxes between kinetically metastable macrostates (henceforth referred to as the 'network'). When projected into two dimensions, the conformational landscape of apo-SETD8 takes the form of a dumbbell-like shape containing two lobes, each composed of about 12 macrostates primarily con-nected via a single hub-like central macrostate A11 (*Figures 4b* and *11*, *Supplementary file 1e*). The conformational landscape also consists of other multiply-connected macrostates, including A1−A4, A9, and A14, as characterized by their rapid kinetic interconversion with multiple other mac-rostates (*Figures 4b* and *5a*). Most low-populated macrostates (A17−A24) appear as satellite macro-states in the periphery of the network with few high-flux channels of interconversion to other macrostates (*Figures 4b* and *5a*). The remaining states were classified as basin-like macrostates including {A5, A10}, A7, A8, {A12, A13, A16} and A15, because these macrostates are highly popu-lated and either are relatively isolated or appear in tightly interconnected but globally isolated groups.

The hub-like macrostate A11 consists of two structurally distinct microstates with comparable populations (*Figures 4b* and *11a*). One microstate structurally resembles the conformation of **APO** ($I_3P_3$), while the other microstate represents a conformer with the $I_1P_{23}$ feature for its SET-I and post-SET motifs (*Figure 11b*, *Supplementary file 1d*). Rapid conformational interconversions within A11 are consistent with its hub-like character, centered between the two lobes of the dumbbell-like net-work. Interestingly, macrostates kinetically adjacent to A11 have structurally similar SET-I motifs within each lobe but distinct SET-I motifs between the two lobes ({**I2** ~**3**} for the left and {**I1** ~**2**} for the right) (*Figures 4b* and *11b*). Therefore, A11 is a transition-type state essential for the conforma-tional fluxes of the macrostates between the two lobes, involved in a key step of conformational changes of the SET-I motif between {**I1** ~**2**} and {**I2** ~**3**}.

The intermediate-like macrostates A1−A4, A9, and A14 each contains multiple structurally dis-tinct but kinetically associated microstates (*Figures 4b*, *11a and b*). The satellite macrostates A17−A24 are less populated and more structurally homogeneous (*Figures 4b*, *11a and b*). Con-formers in the macrostates A22, A24, and A20 are structurally similar to **TC** and **BC-SAM** with slightly different but well-defined SAM-binding pockets, suggesting minimal conformational reorganization of A22, A24, and A20 is required to accommodate the cofactor (*Figure 11a,b,c*). Interestingly, A22 and A24, whose overall structures are similar to each other (**TC**-like), rarely interconvert in the apo landscape (*Figure 4b*). In contrast, the basin-like macrostates {A5, A10}, A7, A8, {A12, A13, A16} and A15 do not contain a well-defined SAM-binding pocket (*Figure 11a,b,c*). Here the conformers in macrostate A12 are similar to **APO**, the conformers in the macrostate A6 are similar to **BC-Inh1**, and the conformers in the macrostate A10 are similar to **BC-Inh2** (*Figure 11d*). The structural similar-ity between the simulated conformers and **BC-Inh1/2** suggests that the two covalent inhibitors suc-cessfully trapped key hidden conformers of apo-SETD8.

Similar to that of apo-SETD8, the interconversion network of the macrostates of SAM-bound SETD8 also displays a dumbbell-like shape with S9 as the hub-like state connecting the two lobes of

the network (*Figures 4b* and *11a*). The macrostates S1 and S3–S5 are multi-connected states; S6, S8, and S10 are satellite-like states; S2 and S7 are basin-like states (*Figure 11a,b*). Notably, the complexity of the overall conformational landscape of SAM-bound SETD8 is significantly reduced in comparison with those of apo-SETD8 (*Figures 4b* and *11a*). The conformers in S1, S2, and S10 are structurally similar to those of A20, as well as **BC-SAM**; the conformers in S4, S6, and S8 are structurally similar to those in A22 and A24, as well as **TC** (*Figure 11d,e*). The structural similarities between these apo and SAM-bound macrostates suggest possible pathways for connecting the two conformational landscapes upon SAM binding.

## Characterization of cancer-associated SETD8 mutants

Sequences from tumor samples retrieved from cBioPortal (*Cerami et al., 2012*; *Cheng et al., 2015*; *Gao et al., 2013*) contain two dozen point mutations in the catalytic domain of SETD8 (*Figure 12a and b*, *Supplementary file 1n*). We expect that some of these mutations perturb SETD8 function. Because of conformational heterogeneity, it has historically been challenging for in silico approaches to annotate how mutations—in particular those structurally remote from functional sites—allosterically affect a target protein on the basis of its static structure(s) (*Campbell et al., 2016*; *Klinman and Kohen, 2014*; *Stefl et al., 2013*). Here, we envisioned addressing this challenge with reference to the conformational ensemble of wild-type SETD8. To characterize mutations remote from catalytic sites (20 out of 24 known mutations), 40 independent microsecond-long MD simulations for each of the cancer-associated apo-SETD8 mutants were conducted with seed structures prepared from one ternary complex (**TC**) conformer—a structure resembling the enzymatic transition state and thus essential for SETD8-catalyzed methylation reaction (*Linscott et al., 2016*). We then constructed a differential residue-contact map for each variant (*Figure 12c,d*) and extracted snapshots representing the largest conformational deviations from the wild-type conformational ensembles (*Figure 12e*). Even with modest simulation time, 8 of the 20 examined cancer-associated mutants displayed neo-conformations that were not observed in the 5 ms wild-type dataset and cannot be predicted from static X-ray crystal structures. Interestingly, all of the neo-conformations display distinct reorganizations at the SET-I motif (*Figure 12e*). For instance, a single point mutation A296T, ~16 Å remote from the active site, yields five distinct neo-conformations (*Figure 12e,f*). In addition, relative to wild-type apo-SETD8, this mutant populates several conformations with a structurally relaxed α-helix at the SET-I motif (*Figure 12e*). C324del, ~20 Å from the SET-I motif, is associated with three neo-conformations and displays the largest changes in the differential contact map (*Figure 12d*, panel 13). The remote H340D mutation is associated with one neo-conformation as well as more populated conformations containing spatially compressed active sites (*Figure 12d*, panel 7; *Figure 12e*). Using in vitro radiometric assays, the A296T and H340D mutants were characterized by loss of the methyltransferase activity on H4K20 peptide substrate (*Figure 12g*). The failure to purify recombinant C324del also supports the impact of this deletion on SETD8 function. H388Q, which mutates a histidine involved in substrate binding, is also associated with neo-conformations as well as loss of the methyltransferase activity (*Figure 12e,g*). These observations provide potential molecular rationale for how remote mutations can alter the active sites and the SET-I motif—and hence catalysis allosterically—via modulating the overall conformational landscape rather than directly affecting specific residues at the catalytic site. Exceptions are T274I, R279W, R279Q, and A368V, which yielded neo-conformations but showed activity comparable to wild-type SETD8 (*Figure 12e,g*), suggesting that certain neo-conformations must either still be catalytically competent or their population may not significantly alter the ability to populate conformations relevant for catalysis. The exceptions suggest that a more complete picture of the conformational ensembles might be necessary to uncover quantitative correlations with the relative methyltransferase activities of these SETD8 mutants.

The differential residue-contact maps further revealed that 8 out of the 20 remote mutations alter conformational landscapes by changing populations of pre-existing conformations (*Figure 12c,d*). For instance, E257K, G280S, A301V, T309M, E330Q, D352Y mutations populate conformations containing spatially compressed active sites (*Figure 12—figure supplement 1*); E372D populates conformations containing a constrained post-SET motif; R333C populates conformations with reorganized SET motifs adjacent to the peptide binding pocket. All of these mutations showed partial loss of methyltransferase activity (*Figure 12g*). Notably, these structural alterations are often remote from the corresponding mutation sites (*Figure 12b*). In contrast, R244S and V356I (2 out of

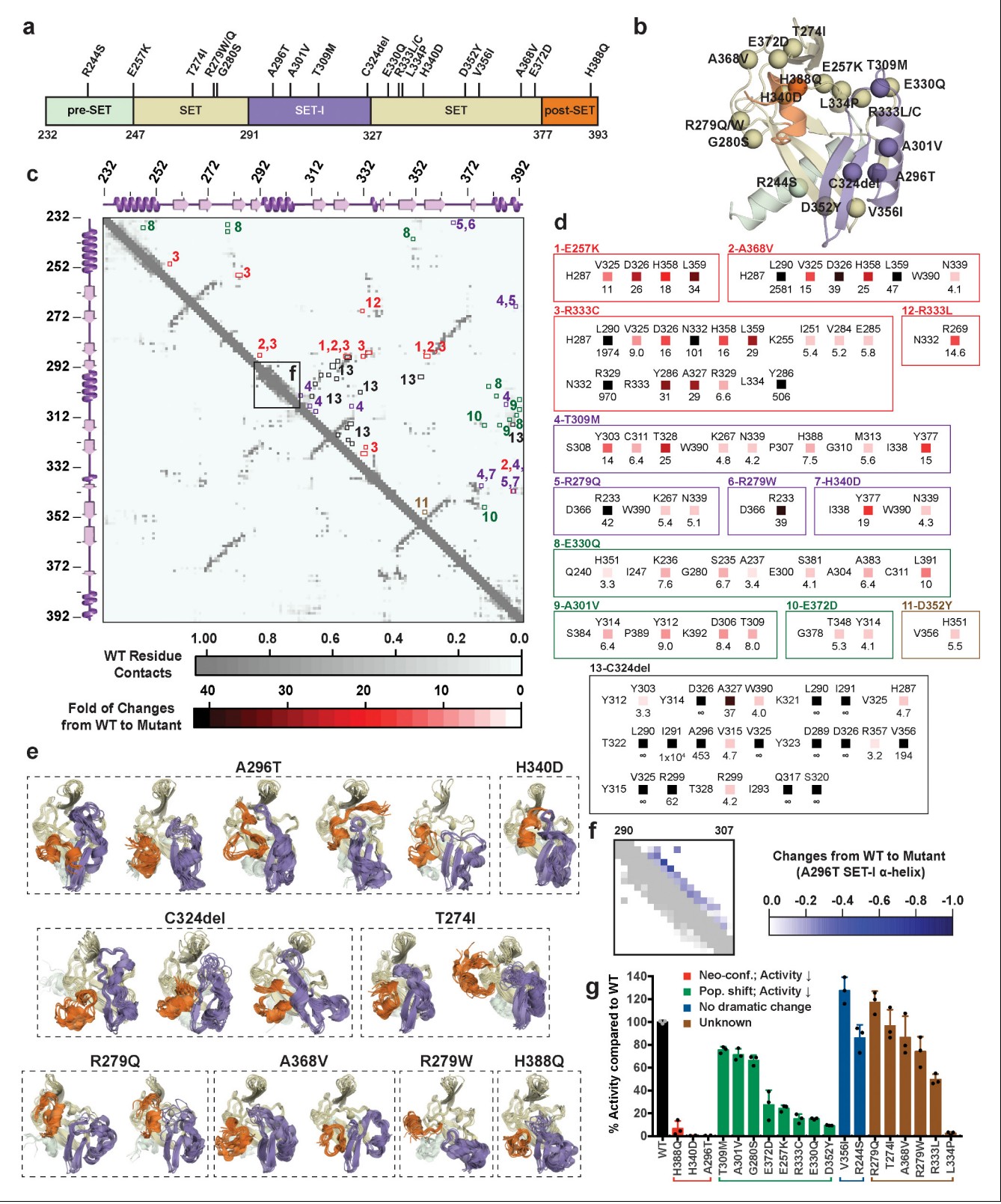

**Figure 12.** Computational and experimental characterization of cancer-associated SETD8 mutants. (a) Cancer-associated mutations in the catalytic domain of SETD8 examined in this work. (b) Cartoon representations of TC with cancer-associated SETD8 mutations highlighted. (c) Differential residue-contact maps of cancer-associated SETD8 mutants in reference to wild-type apo-SETD8 (gray). Residue-residue contact map of wild-type apo-SETD8 is presented as a 162 × 162 matrix. The vertical and horizontal axes show the residue numbers of SETD8's catalytic domain. The contact of a

*Figure 12 continued on next page*

*Figure 12 continued*

pair of residues is scored as '1' if their distance is shorter than 4.0 Å; '0' if the distance is equal or above 4.0 Å. For the 60 1ZKK(chain A)-seeded MD trajectory frames of wild-type apo-SETD8, the average contact fraction of each residue pair is presented in a square shape and depicted with a gray gradient at the corresponding vertical and horizontal coordinates. The contact fraction of cancer-associated SETD8 mutants were obtained in a similar manner. The vertical and horizontal coordinates of representative positive changes of the contact scores from wild-type to mutated SETD8 (newly acquired interactions) are highlighted in red-gradient squares with details expanded in the next panel. (**d**) Representative contacts in the differential residue-contact maps of cancer-associated SETD8 mutants. The contacts of SETD8 mutants with >3 fold gain of contact fraction relative to wild-type SETD8 are listed. Increased magnitude of the contact fraction is depicted in red gradient as described in the previous panel. Only positive changes (newly acquired interactions) are presented with the two residues involved labeled in left and top; the fold of the increase of their contact score labeled in bottom. (**e**) Cartoon representations of neo-conformations revealed by simulations of SETD8 mutants. Large conformational changes are observed in the SET-I (purple) and post-SET (orange) motifs. (**f**) Differential residue-contact maps of the structurally relaxed α-helix at the SET-I motif of SETD8 A296T mutant. Decrease of contact fraction relative to wild-type SETD8 is depicted in blue gradient. (**g**) Enzymatic activities of wild-type and mutated SETD8 determined by an in vitro radiometric assay with H4K20 peptide substrate. Here SETD8 mutants are categorized as the following: red, uncovered neo-conformations (Neo-conf.) with >90% loss of methyltransferase activity; green, populated inactive conformations (Pop. shift) with partially abolished methyltransferase activity; blue, no large change of differential contact maps with comparable methyltransferase activity with wild-type SETD8; brown, unknown relationship between differential contact maps and methyltransferase activities. Data are mean ±standard deviation (s.d.) of 3 replicates.

DOI: https://doi.org/10.7554/eLife.45403.028

The following figure supplements are available for figure 12:

**Figure supplement 1.** Cartoon representations of cancer-associated SETD8 variants with more populated inactive conformations.

DOI: https://doi.org/10.7554/eLife.45403.029

**Figure supplement 2.** Cancer-associated mutations in the SET-I region of PKMTs reported in cBioPortal.

DOI: https://doi.org/10.7554/eLife.45403.030

---

20) showed no significant conformational change on the basis of their differential contact maps, consistent with their comparable methyltransferase activity to wild-type SETD8 (*Figure 12g*). Likely due to insufficient simulation time (40 × 1 µs/mutant), R333L and L334P variants, characterized by partial-to-complete loss of the methyltransferase activity (*Figure 12g*), showed similar conformational landscapes to that of wild-type apo-SETD8. These exceptions, though only a small portion of all mutants studied, point to the necessity of a more extensive exploration of the conformational ensembles to obtain quantitative correlations of the atomistic structure with activities of this collection of SETD8 mutants. Exploring these conformational landscapes is thus an effective strategy to reveal structural alterations associated with majority of remote-site mutations of SETD8 for qualitative functional annotation. More importantly, this change provides a mechanistic rationale of the allosteric effect of remote residues of SETD8 with reference to its conformational landscape.

## Discussion

Here we have demonstrated that tight integration of structural determination—using covalent probes and multiple ligand-binding states to trap hidden conformations (*Figure 1*)—with distributed molecular simulations and the powerful framework of Markov state models (*Figure 3b*) can provide insights into the detailed conformational dynamics of an enzyme. The current work demonstrates the merit of an approach that leverages multiple X-ray structures with distinct diverse conformations of a PKMT for MD simulations and machine-learning-based MSM construction to elucidate complex conformational dynamics, and corroborates the resulting model experimentally with testable biophysical predictions (*Figures 6–8*). Previously, individual components of our integrative strategy have been employed to study the dynamics of transcriptional activators (*Wang et al., 2013*), kinases (*Shukla et al., 2014*; *Sultan et al., 2017*), and allosteric regulation (*Bowman et al., 2015*). Several efforts have also been made to combine experimental and computational approaches to explore conformational landscapes of proteins and their utilities (*Hart et al., 2016*; *Knoverek et al., 2019*; *Latallo et al., 2017*; *Zimmerman et al., 2017*). However, it is the first time that these diverse approaches are consolidated explicitly with the goal of illuminating conformational dynamics of a PKMT in a comprehensive and feasible manner. Assessment of key computational parameters concluded that we have utilized sufficient or even redundant seed structures and simulation time for essentially complete microstate discovery (*Figures 9* and *10*). This implementation is essential for the current work because of the lack of the conformational landscapes of PKMTs as reference or for

validation. Notably, we relied on a unique computational resource—Folding@home—to collect six-millisecond of aggregate simulation data (see Materials and methods). Without access to Folding@-home, contemporaneous progress on developing *adaptive* Markov state model construction algorithms—where iterative model building guides the collection of additional simulation data (*Hruska et al., 2018*; *Shamsi et al., 2017*; *Zimmerman et al., 2018*)—will still allow research groups to achieve this feat on local GPU clusters or cloud resources in the near future. Furthermore, the concept of adaptive model construction can be extended to identify which new structural or biophysical data would be valuable in reducing uncertainty (*Dixit and Dill, 2018*; *Matsunaga and Sugita, 2018*; *Olsson et al., 2017*) and producing refined MSMs. Utilizing the slow collective variables identified here, advanced sampling methods such as metadynamics (*Saladino and Gervasio, 2012*) or umbrella sampling (*Meng and Roux, 2014*) can be applied to more efficiently compute the free energy landscape for SETD8 and its mutants. With a transfer learning approach (*Sultan et al., 2017*), it is also possible to adapt these collective variables to other members of the PKMT protein family.

This work represents the first time that conformational dynamics of a protein methyltransferase have been definitively characterized with atomic details. SETD8 adopts extremely diverse dynamic conformations in apo and SAM-bound states (24 and 10 kinetically metastable macrostates, respectively, *Figure 4*). Interconversions between metastable conformers cover a broad spatio-temporal scale in particular associated with motions of SETD8's SET-I and post-SET motifs (*Figures 1*, *2* and *11*). In the apo landscape, the general structural features of the X-ray structures of **BC-Inh1**, **BC-Inh2**, **APO**, **BC-SAM** and **TC** (*Figure 1*) are recapitulated by a subset of macrostates (*e.g.* A6 for **BC-Inh1**; A10 for **BC-Inh2**; A12 for **APO**; A20 for **BC-SAM**; A22, A24 for **TC**, 6 of 24 macrostates, *Figure 11*). Such observation indicates that these X-ray structures trapped in the different ligand-binding states are not ligand-induced artifacts but indeed relevant snapshots of hidden conformations of apo-SETD8. Similarly, a few macrostates in the SAM-bound landscape also recapitulate major structural features of the two cofactor-bound X-ray structures (*e.g.* S1, S2, S10 for BC-SAM, S4, S6, S8 for TC, 6 of 10 macrostates, *Figure 11*). Meanwhile, our results also demonstrate that X-ray crystallography alone is insufficient to capture all metastable conformations of SETD8. In addition, there is no correlation of overall structural similarity and interconversion rates between metastable conformers. As observed previously in other studies of protein dynamics (*Bowman and Pande, 2010*), in addition to fast transitions between structurally similar conformers and slow transitions between structurally distinct conformers (*e.g.* microstates within individual satellite macrostates A17−A24 of apo–SETD8; S6, S8, and S10 of SAM-bound SETD8, *Figure 11*), we frequently observed fast kinetics of transitions between structurally distinct microstates (*e.g.* microstates within hub-like macrostates A11 and S8; multi-connected states A1−A4, A9, A14, S1 and S3−S5) and *vice versa* (*e.g.* macrostates A22 and A24) (*Figures 4* and *11*). It is thus interesting to examine how other factors such as specific residue contacts and cooperative long-range motions of certain structural motifs play roles in interconversion kinetics. Meanwhile, utilizing the power of Markov state models to stitch together multiple short (microseconds long) trajectories and generate synthetic trajectories orders of magnitude longer (milliseconds), we visualized the MSMs of apo- and SAM-bound SETD8 via 2 ms long (enough to visit all macrostates) movies (*Videos 1* and *2*).

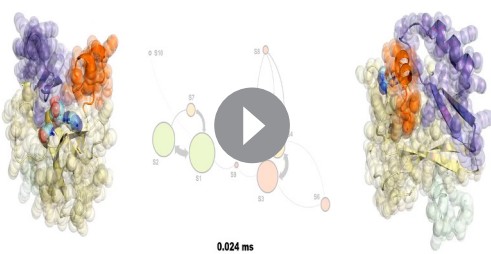

**Video 1.** A 2 ms molecular dynamics trajectory simulated from the HMM of apo-SETD8.
DOI: https://doi.org/10.7554/eLife.45403.031

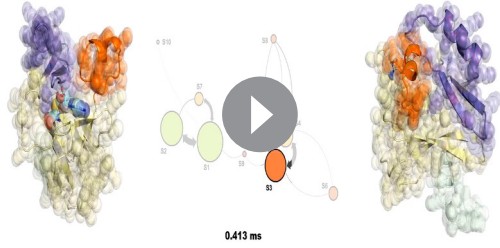

**Video 2.** A 2 ms molecular dynamics trajectory simulated from the HMM of SAM-bound SETD8.
DOI: https://doi.org/10.7554/eLife.45403.032

Functional annotation of the landscapes revealed that the SET-I motif adopts diverse conformations (*Figures 4* and *5*), and its overall configuration is a key feature that differentiates the lobes of the dumbbell-like conformational landscape of SETD8. The conformational dynamics within the hub-like macrostate A11 primarily involve motions of the SET-I motif, secondarily coupling a shift of the post-SET motif. Two gain-of-function I293G and E292G variants of SETD8 were designed for relaxing constrained elongate helix configurations of the SET-I motif upon SAM binding (*Figure 6*). These findings argue the functional essentiality of the intrinsically dynamic motions of SET-I motif for SETD8 SAM binding and catalysis. Importance of dynamic conformational modulation of the SET-I motif has also been shown for other SET-domain PKMTs. For instance, the SET domains of MLLs and EZH1/2 alone are catalytically inert but active in the presence of binding partners WDR5-RbBP5-Ash2L-Dpy30 (referred as MLL-WRAD) and EED-Suz12 (referred as PRC2), respectively (Luo, M., 2018). Recent structural evidence implicated that the formation of these complexes regulates the conformational dynamics of the SET-I motif, which is essential for catalysis (*Justin et al., 2016*; *Li et al., 2016*). Interestingly, this region has also been exploited by cancer-associated mutants of PKMTs. For instance, NSD2's E1099 is located in its SET-I motif and its E1099K mutant was characterized as a hot-spot cancer mutation with the gain-of-activity of H3K36 methylation (*Oyer et al., 2014*). Additionally, many mutations of PKMTs have been mapped in their SET-I motifs, implicating their potential roles in alternation of function (*Figure 12—figure supplement 2*, *Supplementary file 1o*). In contrast to static X-ray structures, this analysis greatly facilitated the characterization of cancer-associated SETD8 mutants (*Figure 12*). Among the 20 examined SETD8 mutations, eight deplete the pre-existing conformations of **TC** and showed the partial loss of activity in comparison with wild-type SETD8 (8 out of 8); eight have neo-conformations with four characterized with the partial loss of methyltransferase activity (4 out of 8); four do not affect the conformational landscape with two characterized for no loss of methyltransferase activity (2 out of 4). Collectively, comparing the conformational landscapes between SETD8 mutations and wild-type **TC** allows us to predict the methyltransferase activity with 70% accuracy (14 out of 20). However, we could not quantitatively correlate the amounts of the neo- or altered conformations of these SETD8 mutants with their methyltransferase activities. We reason that certain nonnative conformations can still be catalytically active. A significant portion of cancer-associated, loss-of-function SETD8 mutations, though remote from active sites, were revealed to perturb the SET-I motif and thus catalysis allosterically via altering the conformational landscape, which is relevant to the formation of the ternary complex and likely the transition state of native SETD8 (*Figure 12*). We also discovered significant changes in the connective networks and a large decrease in conformational heterogeneity of SETD8 upon SAM binding (*Figures 4* and *5*). This finding highlights how SETD8-SAM interactions reshape conformational landscapes. The conformational landscapes of SETD8 thus provide a platform for virtual screening of ligand candidates as inhibitors via exploring different modes of interaction (SAM-competitive, substrate-competitive, covalent or allosteric). Uncovering hidden conformations can thus be essential for developing potent and selective SETD8 inhibitors by targeting these conformations.

Furthermore, it seems feasible that additional simulation effort—if appropriately allocated among poorly-sampled transitions—can produce a statistically precise kinetic model of the conformational dynamics of apo- and SAM-bound SETD8, and that these landscapes could be used to seed simulations for the construction of atomistic models of the rest of the catalytic cycle. Furthermore, the structural information in the resulting models and the kinetic experimental observables could be reconciled using the dynamical fingerprints framework (*Noé et al., 2011*). This approach can also be used to design new experiments by proposing locations of site-specific labels for optimal experimental probing of the molecular relaxation processes of interest. Future work could therefore furnish a quantitative atomistic explanation of the experimentally observed kinetics.

Additionally, these metastable states could be paired with alchemical free energy calculations (*Gapsys et al., 2016*) to rapidly assess the impact of point mutations on the populations of each metastable state in each stage of the catalytic cycle to aid the annotation of the functional impact of these mutations. A prerequisite of our approach was the determination of conformationally diverse structures as seeds for molecular simulations. Here, this was achieved with Cys-covalent inhibitors and native ligand depletion because of the lack of conventional structural probes of SETD8. Given the significant interest in exploring PKMT catalysis and developing selective inhibitors to study functions (*Luo, 2018*), we envision applying similar strategies to other native or disease-associated PKMTs (*Nacev et al., 2019*).

# Materials and methods

## Key resources table

| Reagent type (species) or resource | Designation | Source or reference | Identifiers | Additional information |
|---|---|---|---|---|
| Gene (Homo sapiens) | Human SETD8 catalytic domain | *Blum et al., 2014* | Uniprot: Q9NQR1-1 (positions 232–393) | with an N-t 6 × His tag for in vitro crystallogra |
| Gene (Homo sapiens) | Human SETD8 catalytic domain | *Ma et al., 2014* | Uniprot: Q9NQR1-1 (positions 232–393) | with an N-t 6 × His tag for crystallo BC-Inh2 an |
| Gene (Homo sapiens) | Human SETD8 catalytic domain | Addgene | Plasmid #51327 | with an N-t 6 × His tag for crystallo of BC-Inh1 |
| Strain, strain background (*E. coli*) | Rosetta 2(DE3) | Novagen | #71400 | |
| Strain, strain background (*E. coli*) | BL21-CodonPlus(DE3)-RIL | Stratagene | #230245 | |
| Strain, strain background (*E. coli*) | BL21 (DE3) V2R-pRARE | SGC | | |
| Sequence-based reagent | Forward Primer for K382P | IDT | | 5'-CTATGC CGGCTTC |
| Sequence-based reagent | Forward Primer for I293G | IDT | | 5'-CGGGG GCACCGA |
| Sequence-based reagent | Forward Primer for E292G | IDT | | 5'-CGGGG GCATCAC |
| Peptide, recombinant protein | H4K20 peptide (10-30) | The Rockefeller University Proteomics Resource Center | | $NH_2$-LGKG KVLRDNIQ |
| Chemical compound | SAM | Sigma Aldrich | #A2408 | |
| Chemical compound | [$^3$H-Me]-SAM | PerkinElmer Life Sciences | #NET155001MC | |
| Commercial assay or kit | UltimaGold | PerkinElmer Life Sciences | #6013327 | |
| Software, algorithm | Anaconda Python | *Oliphant, 2007*; *Millman and Aivazis, 2011* | | |
| Software, algorithm | ARP/wARP | *Perrakis et al., 1997*; *Murshudov et al., 2011* | | |
| Software, algorithm | AUTOBUSTER | *Emsley et al., 2010* | | |
| Software, algorithm | CCP4 suite | *Collaborative Computational Project, Number 4, 1994* | | |
| Software, algorithm | COOT | *Emsley and Cowtan, 2004* | | |
| Software, algorithm | Ensembler 1.0.5 | *Parton et al., 2016* | | |
| Software, algorithm | Folding@home | *Shirts and Pande, 2000* | | |
| Software, algorithm | GRADE | *Bruno et al., 2004* | | |
| Software, algorithm | HKL2000 | PMID: 27799103 | | |
| Software, algorithm | IPython | *Perez and Granger, 2007* | | |
| Software, algorithm | Jupyter Notebook | DOI: 10.3233/978-1-61499-649-1-87 | | |
| Software, algorithm | KinTek Explorer | *Johnson et al., 2009* | | |
| Software, algorithm | matplotlib 2.2.2 | *Hunter, 2007* | | |
| Software, algorithm | MDTraj | *McGibbon et al., 2015a* | | |
| Software, algorithm | MOGUL | *Langer et al., 2008* | | |

*Continued on next page*

*Continued*

| Reagent type (species) or resource | Designation | Source or reference | Identifiers | Additional information |
|---|---|---|---|---|
| Software, algorithm | MolProbity | PMID: 20057044 | | |
| Software, algorithm | MOLREP | PMID: 20057045 | | |
| Software, algorithm | MSMBuilder | *Harrigan et al., 2017* | | |
| Software, algorithm | MSMExplorer 1.1 | *Harrigan et al., 2017* | | |
| Software, algorithm | NumPy | https://www.numpy.org | | |
| Software, algorithm | OpenMM 6.6.1 | *Eastman et al., 2013* | | |
| Software, algorithm | Origin 7.0 | OriginLab | | |
| Software, algorithm | OriginPro 2018 | OriginLab | | |
| Software, algorithm | pandas | https://conference.scipy.org/proceedings/scipy2010/pdfs/mckinney.pdf | | |
| Software, algorithm | PDBFixer 1.3 | https://github.com/pandegroup/pdbfixer | | |
| Software, algorithm | PHASER | *McCoy, 2007* | | |
| Software, algorithm | phenix.refine | *Adams et al., 2010*; *Afonine et al., 2012* | | |
| Software, algorithm | POINTLESS/ AIMLESS | *Evans and Murshudov, 2013* | | |
| Software, algorithm | PRODRG | *Schüttelkopf and van Aalten, 2004* | | |
| Software, algorithm | PyEMMA | *Scherer et al., 2015* | | |
| Software, algorithm | PyMOL 1.8.4 | Schrödinger, LLC | | |
| Software, algorithm | REFMAC | *Murshudov et al., 1997* | | |
| Software, algorithm | seaborn 0.8.1 | DOI: 10.5281/zenodo.883859 | | |
| Software, algorithm | MODELLER 9.16 | *Sali and Blundell, 1993* | | |
| Software, algorithm | XDS | *Kabsch, 2010* | | |
| Software, algorithm | XtalView | *McRee, 1999* | | |

## Synthesis of MS4138 (Inh1)

### General procedure for synthesis of MS4138 (Inh1)

HPLC spectra for all compounds were acquired using an Agilent 1200 Series system with DAD detector. Chromatography was performed on a 2.1 × 150 mm Zorbax 300 SB-C18 5 µm column with water containing 0.1% formic acid as solvent A and acetonitrile containing 0.1% formic acid as solvent B at a flow rate of 0.4 mL/min. The gradient program was as follows: 1% B (0–1 min), 1–99% B (1–4 min), and 99% B (4–8 min). High resolution mass spectra (HRMS) data were acquired in positive ion mode using an Agilent G1969A API-TOF with an electrospray ionization (ESI) source. Nuclear Magnetic Resonance (NMR) spectra were acquired on a Bruker DRX-600 spectrometer with 600 MHz for proton ($^1$H-NMR) and 150 MHz for carbon ($^{13}$C-NMR); chemical shifts are reported in ppm (δ). Preparative HPLC was performed on Agilent Prep 1200 series with UV detector set to 254 nm. Samples were injected onto a Phenomenex Luna 75 × 30 mm, 5 µm, C18 column at room temperature. The flow rate was 30 mL/min. A linear gradient was used with 10% (or 50%) of MeOH (A) in $H_2O$ (with 0.1% TFA) (B) to 100% of MeOH (A). HPLC was used to establish the purity of target compounds. All final compounds had >95% purity using the HPLC methods described above.

### *N*-(3-((7-hydroxy-6-methoxy-2-(pyrrolidin-1-yl)quinazolin-4-yl)amino)propyl)-acrylamide (3)

The precursor *N*-(7-(benzyloxy)−6-methoxy-2-(pyrrolidin-1-yl)quinazolin-4-yl)propane-1,3-diamine 2 was prepared from 7-(benzyloxy)−2,4-dichloro-6-methoxyquinazoline1 as previously published

(*Butler et al., 2016*) and dissolved in methanol. Into the solution were added Pd/C and ammonium formate, and stirred for 1 hr at 80°C. The filtrate of the expected product 4-((3-aminopropyl)amino)-6-methoxy-2-(pyrrolidin-1-yl)quinazolin-7-ol was collected, concentrated and directly used for next step without purification. To the solution of 4-((3-aminopropyl)amino)-6-methoxy-2-(pyrrolidin-1-yl)quinazolin-7-ol (150 mg, 0.47 mmol, calculated on the basis of the starting material **2**) and methanol (2.5 mL) were added potassium carbonate (78 mg, 0.56 mmol) and acryloyl chloride (46 μL, 0.56 mmol) successively. The resulting suspension was stirred for 2 hr at room temperature. After removal of the solvent under vacuum, the residue was redissolved in dichloromethane, and washed with brine. The organic layer was dried, concentrated and purified by ISCO CombiFlash to give compound *N*-(3-((7-hydroxy-6-methoxy-2-(pyrrolidin-1-yl)quinazolin-4-yl)amino)-propyl)acrylamide **3** (60 mg, yield 34%). $^1$H-NMR (600 MHz, CD$_3$OD) δ 7.52 (s, 1H), 6.96 (s, 1H), 6.22 (dd, $J$ = 6.0, 4.3 Hz, 2H), 5.67 (dd, $J$ = 8.5, 3.5 Hz, 1H), 3.96 (s, 3H), 3.76–3.54 (m, 6H), 3.38 (t, $J$ = 6.8 Hz, 2H), 2.07 (br.s, 4H), 1.97 (p, $J$ = 6.9 Hz, 2H). HRMS calcd for C$_{19}$H$_{25}$N$_5$O$_3$ + H, 372.2030; found, 372.2043 [M + H]$^+$.

## *N*-(3-((7-(2-aminoethoxy)-6-methoxy-2-(pyrrolidin-1-yl)quinazolin-4-yl)amino) propyl)-acrylamide (MS4138 or Inh1)

To a suspension of *N*-(3-((7-hydroxy-6-methoxy-2-(pyrrolidin-1-yl)quinazolin-4-yl)amino)propyl)-acrylamide **3** (60 mg, 0.16 mmol), KI (5 mg, 0.03 mmol), K$_2$CO$_3$ (66 mg, 0.48 mmol) and acetonitrile (10 mL) was added 2-(Boc-amino)ethyl bromide (36 mg, 0.16 mmol). The resulting suspension was stirred for 3 days at 90°C until LCMS showed that most of the starting material had disappeared. After purification by reverse phase ISCO CombiFlash, *tert*-butyl (2-((4-((3-acrylamidopropyl)amino)−6-methoxy-2-(pyrrolidin-1-yl)quinazolin-7-yl)oxy)ethyl)carbamate **4** was obtained and dissolved in dichloromethane (3.0 mL). To the solution of **4** was added trifluoroacetic acid (37%, 0.2 mL) at 0°C. The resulting solution was stirred at room temperature for 4 hr until LCMS showed that the starting material had disappeared. After removal of the solvent under vacuum, the residue was purified by HPLC to give the desired compound **MS4138** (**Inh1**) as a TFA salt, white solid (8 mg, yield 10% for two steps). $^1$H-NMR (600 MHz, CD$_3$OD): δ 7.65 (s, 1H), 7.19 (s, 1H), 6.28–6.16 (m, 2H), 5.67 (dd, $J$ = 9.0, 3.0 Hz, 1H), 4.43–4.33 (m, 2H), 3.99 (br.s, 3H), 3.74 (t, $J$ = 6.9 Hz, 4H), 3.62 (br.s, 2H), 3.52–3.44 (m, 2H), 3.39 (t, $J$ = 6.8 Hz, 2H), 2.15 (br.s, 2H), 2.05 (br.s, 2H), 1.98 (dt, $J$ = 13.8, 6.8 Hz, 2H) (*Appendix 1—figure 16*). $^{13}$C NMR (151 MHz, CD$_3$OD) δ 168.3, 160.4, 155.1, 151.6, 148.6, 136.8, 132.0, 126.8, 105.5, 104.7, 101.2, 66.8, 57.0, 47.4 (two carbons), 40.3, 40.0, 38.0, 29.7, 26.8, 25.6 (*Appendix 1—figure 17*). HRMS calcd for C$_{21}$H$_{30}$N$_6$O$_3$ + H, 415.2452; found, 415.2444 [M + H]$^+$.

## Synthesis of SGSS05NS (Inh2)
### General procedure for synthesis of SGSS05NS (Inh2)

High resolution mass spectra (HRMS) data were acquired in positive ion mode using a Waters LCT Premier XE with an electrospray ionization (ESI) source. Nuclear Magnetic Resonance (NMR) spectra were acquired on a Bruker Avance III 500 spectrometer with 600 MHz for proton ($^1$H-NMR) and Bruker Avance III 600 spectrometer with 150 MHz for carbon ($^{13}$C-NMR); chemical shifts are reported in ppm (δ).

## 2-Chloro -3-(4-methyl-1-piperazinyl) -1, 4-naphthalenedione (SGSS05N)

2,3-Dichloro-1,4-naphthalenedione **5** (100 mg, 0.44 mmol) was reacted with 1-methyl-piperazine (49 μL, 0.44 mmol) in 1,4-dioxane (5 mL) overnight at room temperature. The resulting mixture was washed with saturated sodium bicarbonate and extracted with 20 mL ethylacetate. The organic phase was further washed with water and brine, dried on sodium sulfate and concentrated by rotary evaporation. The final product was purified by normal phase silica gel flash chromatography (methanol/dichloromethane, 9:1). The desired product was obtained as red orange liquid (109 mg, yield 85%). $^1$H-NMR (500 MHz, chloroform-*d*) δ 8.12 (dd, $J$ = 7.6, 1.6 Hz, 1H), 8.01 (dd, $J$ = 7.9, 1.7 Hz, 1H), 7.72–7.65 (m, 2H), 3.65 (dd, $J$ = 6, 4.86 Hz, 4H), 2.64–2.62 (m, 4H), 2.38 (s, 3H) (*Appendix 1—figure 18*). $^{13}$C-NMR (151 MHz, chloroform-*d*) δ 182.11, 178.35, 149.95, 134.55, 133.61, 131.83, 131.65, 127.28, 127.01, 124.28, 55.52, 50.64, 45.83 (*Appendix 1—figure 19*). HRMS calcd for C$_{15}$H$_{15}$ClN$_2$O$_2$ + H, 291.0900; found, 291.0894 [M + H]$^+$.

## 2-(4-methyl-1-piperazinyl)-3-(phenylthio)-1,4-naphthalenedione (SGSS05NS)

2-chloro-3-(4-methyl-1-piperazinyl)-1,4-naphthalenedione (**SGSS05N**) (100 mg, 0.34 mmol) was reacted in methanol (5 mL) with thiophenol (70 µL, 0.68 mmol) in the presence of triethylamine (95 µL, 0.68 mmol) overnight at room temperature. The resulting mixture was washed with saturated sodium bicarbonate, and extracted with 20 mL ethylacetate. The organic phase was further washed with water and brine, dried on sodium sulfate and concentrated by vacuum. The final products were purified on silica gel flash chromatography (methanol/dichloromethane, 9:1). After removing the solvent through rotary evaporation, a red dark liquid was collected as the final product, 2-(4-methyl-1-piperazinyl)-3-(phenylthio)−1,4-naphthalenedione (**SGSS05NS**) (115 mg, yield 93%). $^{1}$H-NMR (500 MHz, Chloroform-$d$), δ 8.07 (dd, $J$ = 7.1, 1.7 Hz, 1H), 8.02 (dd, $J$ = 6.8, 1.6 Hz, 1H), 7.70–7.65 (m, 2H), 7.25–7.21 (m, 4H), 7.17–7.13 (m, 1H), 3.51 (dd, $J$ = 6.2, 3.9 Hz, 4H), 2.58–2.49 (m, 4H), 2.31 (s, 3H) (*Appendix 1—figure 20*). $^{13}$C-NMR (151 MHz, chloroform-$d$) δ 182.47, 182.11, 154.17, 136.29, 134.34, 133.29, 132.86, 132.37, 129.36, 128.14, 127.09, 126.91, 126.67, 55.68, 51.37, 46.15 (*Appendix 1—figure 21*). HRMS calcd for $C_{21}H_{20}N_2O_2S$ + H, 365.1324; found, 365.1331 $[M + H]^+$.

## Preparation of SETD8 and its mutants for biochemical assays

Human SETD8 catalytic domain (Uniprot Q9NQR1-1 positions 232–393, SRKSKAELQSEERKRIDELIE SGKEEGMKIDLIDGKGRGVIATKQFSRGDFVVEYHGDLIEITDAKKREALYAQDPSTGCYMYYFQYLSKTYC VDATRETNRLGRLINHSKCGNCQTKLHDIDGVPHLILIASRDIAAGEELLDYGDRSKASIEAHPWLKH) with an N-terminal 6 × His tag in pHIS2 vector was overexpressed in *E. coli* Rosetta 2(DE3) in LB medium in the presence of 100 µg/ml of ampicillin. Cells were grown at 37°C to an $OD_{600}$ of 0.4 ~ 0.6 and the expression of SETD8 was induced by 0.4 mM isopropyl-1-thio-*D*-galactopyranoside (IPTG) at 17° C overnight. Harvested cells were suspended in a lysis buffer (50 mM Tris-HCl, pH = 8.0, 25 mM NaCl, 10% Glycerol, 25 mM imidazole) supplemented with EASY pack protease inhibitor (one tablet/ 10 mL solution), a tip amount of lysozyme and DNAase I. The mixture was lysed by FrenchPress. SETD8 (aa 232–393) was purified by a Ni-NTA column subjected to a washing buffer (50 mM Tris-HCl, pH = 8.0, 25 mM NaCl, 10% glycerol, 25 mM imidazole) and then an eluting buffer (50 mM Tris-HCl, pH = 8.0, 25 mM NaCl, 10% glycerol, 400 mM imidazole). The protein was further purified by a Superdex-75 gel filtration column with a buffer containing 25 mM Tris-HCl (pH = 8.0), 200 mM NaCl, and 10% glycerol. The elution fractions were pooled, supplemented with 5 mM of tris(2-carboxyethyl)phosphine (TCEP), and concentrated to about 60 mg/mL for storage at −80°C. All purification was conducted at 4°C. The N-terminal 6 × His SETD8 (aa 232–393) construct was used to measure $IC_{50}$ of SETD8 inhibitors. Plasmids of SETD8 mutants were generated by QuickChange site-directed mutagesis kit (Stragaene) according to manufacturer's instructions and validated by DNA sequencing. Primer sequences for mutagesis were designed by PrimeX and listed in *Supplementary file 1p*. SETD8 mutants were expressed and purified as described above for wild-type SETD8.

## Measurement of $IC_{50}$ of SETD8 inhibitors

The $IC_{50}$ of SETD8 inhibitors were measured by a previously reported filter plate assay (*Blum et al., 2014*; *Ibanez et al., 2012*) with some modifications. DMSO stock solutions of SETD8 inhibitors with different concentrations were prepared through series dilution. The final assay mixture (a total volume of 20 µL) contains 300 nM SETD8 protein (N-terminal 6 × His taged, amino acid 232–393), 10 µM H4K20 peptide (aa 10–30, prepared by Rockefeller University Proteomics Resource Center, New York, NY), 1.5 µM [$^{3}$H-Me]-SAM (PerkinElmer Life Sciences), and various concentrations of inhibitors in a reaction buffer (50 mM HEPES, pH = 8.0, 0.005% Tween-20, 5 µg/mL BSA, 1 mM TCEP and 0.5% DMSO). Prior to each reaction, 10 µL of a reaction mixture containing 2 × concentrations of SETD8 and inhibitors was pre-incubated at ambient temperature (22°C) for 2 hr. 10 µL of another reaction mixture containing 2 × concentrations of peptide and [$^{3}$H-Me]-SAM was then added to initialize the reaction. The resulting mixture was allowed to react at ambient temperature (22°C) for 2 hr. 3 × 6 µL (total 18 µL) of this mixture were spotted onto 3 wells of MultiScreen$_{HTS}$ PH Filter plate (Millipore) to immobilize $^{3}$H-labeled peptide. After drying in ambient air overnight, each well was washed 6 times with 200 µL of 50 mM $Na_2CO_3$/$NaHCO_3$ buffer (pH = 9.2), followed by the addition of 30 µL Ultima Gold scintillation cocktail (PerkinElmer Life Sciences). The plate was sealed and the

mixture was further equilibrated for 30 min. The immobilized radioactivity of $^3$H-labeled peptide was quantified by 1450 Microbeta liquid scintillation counter. The inhibition curve was generated according to the equation: Percentage of inhibition = [("CPM of no inhibitor control" − "CPM of a reaction mixture")/("CPM of no inhibitor control" − "CPM of background")]×100%. The IC$_{50}$ values were obtained by fitting inhibition percentage versus concentrations of inhibitors using GraphPad Prism. Data presented are best fitting values ± s.e.

## Crystallography
### BC-Inh1 (6BOZ)
Human SETD8 catalytic domain (amino acids 232–393) with a C343S mutation and an *N*-terminal 6 × His tag in pHIS2 vector was overexpressed in *E. coli* BL21-CodonPlus(DE3)-RIL in Terrific Broth medium in the presence of 100 µg/ml of carbenicillin and 30 µg/ml of chloramphenicol. Cells were grown at 37°C to an OD$_{600}$ of 2.5 and SETD8 expression was induced by 0.3 mM IPTG with a supplement of 1 mM zinc sulfate at 15°C overnight. Harvested cells were suspended in a lysis buffer (50 mM sodium phosphate, pH = 7.5, 0.5 mM NaCl, 5% glycerol) and lysed by microfluidizer. The SETD8 protein (aa 232–393) was purified by a Ni-NTA column. The column was washed by a washing buffer (50 mM sodium phosphate, pH = 7.5, 0.5 mM NaCl, 5% glycerol) and the protein was eluted by an eluting buffer (50 mM Tris, pH = 8.0, 250 mM NaCl, 250 mM imidazole, 0.5 mM TCEP). *N*-terminal His tag was removed by TEV protease. The protein was further purified by a Superdex 200 (26/600) gel filtration column with a buffer containing 50 mM Tris-HCl (pH = 8.0) and 150 mM NaCl. The elution fractions were pooled and supplemented with 0.5 mM of TCEP. All purification steps were performed at 4°C and in the presence of a protease inhibitor AEBSF (Goldbio).

The purified SETD8 protein sample was mixed with **Inh1** (**MS4138**) at a molar ratio of 1:5, and incubated at 4°C overnight. The solution was then concentrated to about 20 mg/mL and crystallized with the hanging drop vapor diffusion method at 17°C by mixing equal volume of the protein solution with the reservoir solution (0.1 M HEPES, pH = 7.0, 20% (w/v) PEG 6,000, 0.2 M MgCl$_2$). SETD8-MS4138 crystals (**BC-Inh1**) were soaked in the corresponding reservoir liquor supplemented with 20% ethylene glycol as cryoprotectant before flash freezing in liquid nitrogen. X-ray diffraction data were collected at 100K at NE-CAT beamline 24-ID-E of Advanced Photon Source (APS) at Argonne National Laboratory. The data integration and reduction were performed with MOSFLM and SCALA, respectively, from the CCP4 suite (*Collaborative Computational Project, Number 4, 1994*). The structures of the SETD8-MS4138 complex were solved by molecular replacement using PHASER software (*McCoy, 2007*) using the atomic model of the SETD8 catalytic domain (PDB file 4IJ8). The locations of the bound molecules were determined from a Fo-Fc difference electron density map. REFMAC (*Murshudov et al., 1997*) and phenix.refine (*Adams et al., 2010*; *Afonine et al., 2012*) were used for structure refinement. Graphic program COOT (*Emsley and Cowtan, 2004*) was used for model building and visualization. The overall assessment of model quality was performed using MolProbity (*Chen et al., 2010*). Data reduction and refinement statistics are summarized in *Table 1*.

### BC-Inh2 (5W1Y)
Human SETD8 catalytic domain (amino acid 232–393) with a C343S mutation and an *N*-terminal 6 × His tag in pHIS2 vector was overexpressed in *E. coli* BL21 (DE3) V2R-pRARE in Terrific Broth medium in the presence of 50 µg/ml of ampicillin and 50 µg/ml of chloramphenicol (*Ma et al., 2014*). Cells were grown at 37°C to an OD$_{600}$ of 1.5 and SETD8 expression was induced by 1 mM IPTG at 15°C overnight. Harvested cells were suspended in lysis buffer (50 mM Tris-HCl, pH = 8.0, 300 mM NaCl, 20 mM imidazole, 1 mM phenylmethyl sulfonyl fluoride (PMSF)) and lysed by sonication. SETD8 (aa 232–393) was purified by Ni-NTA column. The column was washed by a washing buffer (50 mM Tris-HCl, pH = 8.0, 300 mM NaCl, 20 mM imidazole) and the protein was eluted by an eluting buffer (50 mM Tris-HCl, pH = 8.0, 300 mM NaCl, 250 mM imidazole). *N*-terminal His tag was removed by TEV protease. The protein was further purified by a Superdex-75 gel filtration column with a buffer containing 50 mM Tris-HCl (pH = 8.0), 100 mM NaCl and 5 mM 1,4-dithiothreitol (DTT). The elution fractions were pooled and concentrated to about 0.7 mg/mL.

The purified SETD8 protein sample (final concentration 1.4 mM) was mixed with **Inh2** (**SGSS05NS**, final concentration 4.2 mM) at a molar ratio of 1:3, and incubated on ice for 3 hr until SETD8 was completely covalently modified (confirmed by mass spectrometry). Crystals were initially obtained

with a sitting-drop vapor diffusion method at the condition of 0.2 M NaF, 20% w/v polyethylene glycol 3350 by mixing 0.5 uL of this solution with 0.5 uL of the SETD8-**Inh2** solution against 90 uL reservoir buffer at 18°C. Crystals grew to a mountable size in three days, and were soaked in reservoir solution with newly added glycerol (v/v 15%) as a cryoprotectant before mounting. Diffraction data were collected under cooling at beam line 19ID of the Advanced Photon Source and reduced with XDS (*Kabsch, 2010*). Intensities for a 100-degree wedge of the images were merged with POINTLESS/AIMLESS (*Evans and Murshudov, 2013*). The structure was solved by molecular replacement with PHASER software (*McCoy et al., 2007*) and coordinates from the SETD8-SAM complex (**4IJ8**, see below). Geometry restraints for the compound were calculated with PRODRG (*Schüttelkopf and van Aalten, 2004*) or, for later stages of refinement, with GRADE (*Bruno et al., 2004*), which uses MOGUL (*Langer et al., 2008*). The protein model was automatically rebuilt with ARP/wARP (*Murshudov et al., 2011*; *Perrakis et al., 1997*). REFMAC (*Bricogne et al., 2016*) and AUTOBUSTER were used for restrained refinement (*Emsley et al., 2010*). COOT and MolProbity were used for interactive rebuilding and geometry validation, respectively (*Adams et al., 2010*; *Chen et al., 2010*; *Yang et al., 2004*). Data reduction and refinement statistics are summarized in *Table 1*.

## BC-SAM (4IJ8)

The conditions for expression and purification of SETD8 (amino acid 232–393 containing a C343S mutation) for crystallography of **BC-SAM** is similar to those of **BC-Inh2** with slight modifications. Purified protein samples were concentrated to about 18 mg/mL, and then mixed with SAM at a molar ratio of 1:10 and incubated on ice for one hour. The sample was crystallized using the sitting drop vapor diffusion method at 18°C. The crystals of SETD8 in complex with SAM were grown in a condition of 1.08–1.2 M trisodium citrate and 100 mM HEPES (pH = 7.5). SETD8-SAM crystals were soaked in the corresponding reservoir liquor supplemented with 20% ethylene glycol as cryoprotectant before flash freezing in liquid nitrogen. Diffraction images were collected at beam line 08ID of the Canadian Light Source (*Grochulski et al., 2011*). Diffraction images were processed with the HKL software suite (*Otwinowski and Minor, 1997*) for early stages of structure determination. For later steps of model refinement, diffraction images were processed with XDS, and intensities further scaled with SCALA (*Evans, 2006*). A starting model was obtained from an isomorphous crystal structure, which had been solved by molecular replacement with coordinates from PDB entry 1ZKK (*Couture et al., 2005*). The model was automatically rebuilt with ARP/wARP, manually rebuilt with COOT, and refined with REFMAC. Data reduction and refinement statistics are summarized in *Table 1*.

## Apo (5V2N)

Human SETD8 catalytic domain (amino acid 231–393) with mutations of K297A, K298A, E300A and an N-terminal 6 × His tag in pET28 vector was overexpressed in Rosetta2(DE3) *E. coli* strain in LB medium in the presence of 50 mg/L kanamycin and 34 mg/L chloramphenicol. The K297A, K298A, E300A mutants were introduced to reduce entropy at the protein surface and thus enhance the ability of apo-SETD8 to crystallize. Cells were grown at 37°C to an $OD_{600}$ of 0.8 and SETD8 expression was induced by 0.4 mM IPTG at 17°C overnight. Harvested cells were suspended in lysis buffer containing 25 mM Tris (pH = 7.6), 500 mM NaCl, 0.25 mM TCEP, 0.5% Triton X-100, and protease inhibitors, and lysed by microfluidizer. SETD8 (aa 231–393) was purified by a cobalt column. The column was washed by a washing buffer containing 25 mM Tris (pH = 7.6), 500 mM NaCl, 0.25 mM TCEP. The protein was eluted by an eluting buffer containing 25 mM Tris (pH = 7.6), 500 mM NaCl, 200 mM imidazole, and 0.5 mM TCEP. N-terminal 6 × His tag was removed by TEV protease. The protein was further purified by a Superdex-75 gel filtration column with a buffer containing 20 mM TrisHCl (pH = 7.0), 100 mM NaCl and 1 mM TCEP. The elution fractions were pooled and dialyzed against 20 mM Tris-HCl (pH = 7.0), 100 mM NaCl and 1 mM TCEP. The peak fractions were pooled, concentrated to 26 mg/ml, and immediately frozen as aliquots with liquid nitrogen.

Initial crystal trials were conducted with Takeda California's automated nanovolume crystallization platform. The purified SETD8 protein sample (26 mg/ml) was crystallized with a sitting drop vapor diffusion method at 20°C with reservoirs containing 100 mM Tris (pH 8.2–8.8), 30% PEGMME 550, and 5% ethylene glycol. Crystals were soaked in the corresponding reservoir liquor supplemented

with 22% ethylene glycol as cryoprotectant before flash freezing in liquid nitrogen. Diffraction data were collected from a single cryogenically protected crystal at the Advanced Photon Source (APS) beamline 23-ID-B at Argonne National Laboratory. Data were reduced using the HKL2000 software package (*Otwinowski and Minor, 1997*). The structure was determined by molecular replacement with either MOLREP (*Vagin and Teplyakov, 1997*) of the CCP4 program suite utilizing the SETD8 catalytic domain (PDB file 4IJ8) as search model, and refined with the program REFMAC (*Murshudov et al., 1997*). Several cycles of model building with XtalView (*McRee, 1999*) and refinement were performed for improving the quality of the model. Data reduction and refinement statistics are summarized in *Table 1*.

## Preparation of SAM-free SETD8

SAM-free SETD8 was prepared as described previously (*Linscott et al., 2016*). Briefly, the concentrated *N*-terminal 6 × His tagged SETD8 protein (aa 232–393, ~60 mg/mL) was diluted by about 1:10 ratio (v/v) with a stripping buffer (25 mM Tris-HCl, pH = 8.0, 35 mM KCl, and 5% glycerol). Activated charcoal was added into the solution (1:1 w/w ratio of protein versus charcoal). The resulting mixture was incubated for 45 min. The charcoal-treated sample was then centrifuged and filtered to afford SAM-free SETD8. All these steps were performed at 4°C. SAM-free SETD8 mutants were prepared in a similar manner.

## Isothermal titration calorimetry (ITC)

Dissociation constants of SETD8 with SAM (Sigma-Aldrich) were measured using an Auto-iTC200 calorimeter (MicroCal) at 20°C. Both SAM and SAM-free SETD8 proteins were dissolved into an assay buffer containing 50 mM HEPES (pH = 8.0), 0.005% Tween-20, 5 µg/mL BSA, 0.00125% TFA, and 1 mM TCEP. 2.5 mM SAM was titrated into 125 µM SETD8 through 20 injections. Experimental data were analyzed by Origin 7.0 after correcting the heat generated upon injecting SAM into the assay buffer. Best fits were obtained with a fixed stoichiometry (N = 1). Data are shown as mean ± s.e. of at least three biological replicates.

## Stopped-flow rapid mixing experiment

The binary binding kinetics of SAM to SETD8 (wild-type and mutants) were studied using stopped flow spectrometry (SX20, Applied Photophysics). The slit widths of the entrance and exit of the monochromator were set to 2.0 mm. Equal volume of samples from two 2.5 mL syringes were driven into a 20-µL observation cell to mix at ambient temperature (22 °C), to reach the final concentration of 1 µM SAM-free SETD8 and serial concentrations of SAM (16 µM to 2000 µM) in a mixing buffer containing 50 mM HEPES-HCl (pH = 8.0), 0.005% Tween 20, and 1 mM TCEP. 6–8 shots (drives) were taken for each SAM concentration. Trp fluorescence change was recorded for 1 second upon mixing with an excitation wavelength of 295 nm and a wavelength cutoff emission filter ($\geq$ 320 nm). 10000 data points were collected with Pro-Data SX20 software for each stopped-flow experiment. Data analysis was performed using KinTek Explorer (*Johnson et al., 2009*). For the global fitting, the signal traces for all concentrations of SAM were simultaneously fitted to a two-step binding model with an initial binding step followed by the step of further conformational changes: "E + SAM", "ES" and "E′S", in which E, ES, and E′S correspond to different states of SETD8. The fluorescence signal was defined as the expression $F = a \times [E] + b \times [ES] + c \times [E'S] + bkg$, in which F is the detected total fluorescence intensity, a, b, and c are fluorescence coefficients of E, ES, and E′S, respectively, and bkg is the background fluorescence intensity. For the calculation of equilibrium constants, the equations of $K_{d1} = k_{-1}/k_1$, $K_{eq} = k_{-2}/k_2$, and $K_d = K_{d1} \times K_{eq}/(1 + K_{eq})$ were followed. For conventional fittings, the fluorescence data were fitted into *Equation. 1*, in which F is the fluorescence intensity, $A_1$ and $A_2$ are the amplitudes of the signal changes for fast and slow phases, respectively, $k_{obs}^{fast}$ and $k_{obs}^{slow}$ are the observed rate constants for two phases, and t is time. The plot of $k_{obs}^{fast}$ and $k_{obs}^{slow}$ versus SAM concentrations were fitted with *Equation. 2* and *Equation. 3*, respectively, where [S] is the concentration of SAM, $k_i$ and $k_{-i}$ are the association and dissociation rate constants for step i (i = 1 or 2), respectively. For individual rate constants, data are best fitting values ± s.e. from KinTek. Uncertainties of $K_{d1}$, $K_{eq}$, $K_d$ are shown as s.e. calculated by the propagation of s.e. from individual rate constants and dissociation constants, respectively. Meanwhile, the

data was also globally fitted into a conformational-selection model (E = E' + SAM = E'SAM) and failed to generate good fitting results (*Appendix 1—figure 22*).

$$F = A_1 \times \exp(-k_{\text{obs}}^{\text{fast}} \times t) + A_2 \times \exp(-k_{\text{obs}}^{\text{slow}} \times t) + C \tag{1}$$

$$k_{\text{obs}}^{\text{fast}} = k_1 \times [\text{S}] + k_{-1} + k_2 + k_{-2} \tag{2}$$

$$k_{\text{obs}}^{\text{slow}} \approx \frac{k_1 \times [\text{S}] + (k_{-2} + k_2) + k_{-1} \times k_{-2}}{(k_1 \times [\text{S}] + k_{-1} + k_2 + k_{-2})} \quad (\text{plateau} \approx k_{-2} + k_2) \tag{3}$$

## Stopped-flow rapid dilution experiment

25 μM SAM-free SETD8 (wild-type and mutants) was pre-mixed with serial concentrations of SAM (1000 μM to 2000 μM) in the mixing buffer and incubated for 10 min at ambient temperature (22°C). The pre-mixed samples were loaded into a 100 μL syringe, and the mixing buffer was loaded into a 2.5 mL syringe. The two syringes were then driven into the observation cell and mixed to achieve a 1:25 dilution of the pre-mixed samples. The time-dependent fluorescence signal changes were recorded up to 3 s under the same setting as described above for the binding assay. Total of 11333 points were collected with 10000 points for the first 1 s and 1333 points for 1–3 s. Conventional fitting of results was performed using KinTek Explorer following equation: $F = A_1 \times \exp(-k_{-1} \times t) + C$, in which $A_1$ is the amplitude of the signal change, $k_{-1}$ is the dissociation constant for the first step in rapid quenching experiment, and $t$ is time. Signals from different concentrations of SAM are fitted separately, and the average $k_{-1}$ is calculated accordingly. Data are best fitting values ± s.e. from KinTek.

## Methyltransferase assay of cancer-associated SETD8 mutants

The methyltransferase activities of wild-type and cancer-associated SETD8 mutations were characterized by a previously described filter paper assay (*Blum et al., 2014*; *Ibanez et al., 2012*) with some modifications. Briefly, 50 nM SETD8 protein (*N*-terminal 6 × His tag, amino acids 232–393, wild-type or mutants), 1.5 μM [³H-Me]-SAM, and 30 μM histone H4 peptide (amino acids 10–30) were incubated in a reaction buffer containing 50 mM HEPES (pH = 8.0), 0.005% Tween 20, 5 μg/mL BSA, and 1 mM TCEP at ambient temperature (22°C) for 3 hr. Each reaction mixture was split into three aliquots and quenched by spotting on phosphor cellulose (P-81) filter paper, followed by 2 hr air-dry. The dried filter paper was then washed 5 times with 50 mM $Na_2CO_3/NaHCO_3$ solution (pH = 9.2). The washed filter paper was then transferred into a scintillation vial, well mixed with 0.5 ml $ddH_2O$ and 5 ml Ultima Gold, and analyzed by a Liquid Scintillation Analyzer (Perkin Elmer Tri-Carb 2910 TR). The methyltransferase activities of SETD8 mutants relative to that of wild-type SETD8 were calculated with the following equation: Percentage of relative activity = [(CPM of mutant – CPM of background)/(CPM of wild type – CPM of background)]×100%. Data are presented as mean ± s.d. of 3 biological replicates.

## Molecular dynamics (MD) simulations of apo-SETD8

### Preparation of molecular dynamics (MD) simulations

All-atom models of the 162-residue SET-domain-containing apo-SETD8 fragment (amino acids 232–393, corresponding to the catalytic domain used in our biochemical experiments) were prepared using Ensembler 1.0.5 (*Parton et al., 2016*) with default parameters unless otherwise specified. Ensembler automatically corrects sequence variations and models in missing atoms (*Parton et al., 2016*). To prepare apo protein models with diverse conformations for simulation, the crystal structures of **BC-Inh1** (6BOZ), **BC-Inh2** (5W1Y), **BC-SAM** (4IJ8), **APO** (5V2N), and **TC** (1ZKK, 2BQZ, 3F9W, 3F9X, 3F9Y, 3F9Z) together with four structural chimeras were used as templates for MODELLER 9.16 (*Sali and Blundell, 1993*) (see *Supplementary file 1a* for details). The structural chimeras were constructed with MDTraj 1.7.2 (*McGibbon et al., 2015a*) by combining the C-flanking domain (residues 377–393) with the rest of the protein from different crystal structures with details described below. Using OpenMM 6.3.1 (*Eastman et al., 2013*), protonation states appropriate for pH = 7 were assigned with openmm.app.modeller, which uses intrinsic $pK_a$ values to determine the most likely ionization states of individual residues but ensures all models are created with the same

protonation and tautomeric state so they can be analyzed collectively. The protein was then energy-minimized for 20 steps and relaxed with 100 ps of implicit solvent dynamics using the OpenMM Langevin integrator with a 2-fs timestep and a 20 $ps^{-1}$ collision rate in the NVT ensemble (T = 300 K). All covalent bonds involving hydrogen were constrained. The protein was then solvated with water in a cubic box with at least 1 nm padding on all sides of the protein, and neutralized with a minimal amount of NaCl. All available chains in the template crystal structures were modeled separately (see SI for details), resulting in 30 simulation-ready structures (representing nine distinct conformers) solvated by an equal number of water molecules (35,200 atoms total). These structures were equilibrated for 5 ns in the NpT (p=1 atm, T = 300 K) ensemble. Pressure was controlled by a Monte Carlo molecular-scaling barostat with an update interval of 50 steps. Non-bonded interactions were treated with the Particle Mesh Ewald method (*Darden et al., 1993*) using a real-space cutoff of 0.9 nm and relative error tolerance of 0.0005, with grid spacing selected automatically. These simulations were subsequently packaged as seeds for production simulation on Folding@home (*Shirts and Pande, 2000*). For all simulations, the parameter files included in the OpenMM 6.3.1 distribution (*Eastman et al., 2013*) were used for the Amber ff99SB-ILDN force field (*Lindorff-Larsen et al., 2010*), the GBSA-OBC2 implicit solvent model (*Onufriev et al., 2004*) (for implicit refinement), the TIP3P rigid water model (*Jorgensen et al., 1983*) (for explicit equilibration and production), and the adapted Aqvist (Na$^+$) (*Aqvist, 1990*) and Smith and Dang (Cl$^-$) (*Smith and Dang, 1994*) parameters for NaCl. Default parameters were used unless noted otherwise.

## Preparation of structural chimeras as ensembler templates

Four new structural chimeras, in addition to the five crystal structures, were produced by combining the post-SET motif (residues 377–393, 'fragment 2') with the rest of the protein (residues 232–376, 'fragment 1') from two different crystal structures. The four new crystal structures (**BC-Inh1** (6BOZ), **BC-Inh2** (5W1Y), **BC-SAM** (4IJ8), and **APO** (5V2N)) were superposed to **TC** (1ZKK) in PyMOL 1.8.4 (*Schrödinger LLC, 2019*) and examined manually. The structural chimeras generated were: (1) **I1-P3** (fragment one from **TC** (1ZKK) or **BC-Inh2** (5W1Y) structures— the **BC-Inh1** (6BOZ, also **I1**) structure was not yet available when this experiment was initialized; fragment two from **APO** (5V2N) structure), (2) **I2-P3** (fragment one from **BC-SAM** (4IJ8) structure; fragment two from **APO** (5V2N) structure), (3) **I2-P4** (fragment one from **BC-SAM** (4IJ8) structure; fragment two from **BC-Inh1** (6BOZ) structure), and (4) **I3-P4** (fragment one from **APO** (5V2N) structure; fragment two from **BC-Inh1** (6BOZ) structure). The heavy-atom-only homology models derived from the corresponding crystal structures generated by Ensembler 1.0.5 were used to construct the structural chimeras so they could be directly superimposed for coordinate transfer. The homology models were superposed on all atoms to the **APO** structure, and the appropriate fragments were isolated and re-joined using MDTraj 1.8 (*McGibbon et al., 2015a*). The resulting models were injected into the Ensembler workflow as new templates using a dedicated script, as these features were not yet available in Ensembler. Steric clashes were observed for the following pairs: (1) SET-I motif of **BC-SAM** (4IJ8) (**I2**) and post-SET motif of **BC-Inh2** (5W1Y) (**P2**); (2) SET-I motif of **APO** (5V2N) (**I3**) and post-SET motif of **TC** (1ZKK) (**P1**); (3) SET-I motif of **APO** (5V2N) (**I3**) and post-SET motif of **BC-Inh2** (5W1Y) (**P2**). The three combinations were then excluded from subsequent procedures.

## Ensembler homology modeling

For the generation of structural chimeras, only 'A' chains from the appropriate crystal structures were used, as this part of the workflow was not automated. For all other seed conformations, if multiple protein chains were present in the crystal structures, these were treated as separate templates by Ensembler 1.0.5 in order to increase the conformational heterogeneity of simulation starting points. When multiple crystal structures of the same conformation were available (*e.g.* the **TC** conformation), all chains of all crystal structures were modeled separately. Herein one chain was present in the **APO** crystal structure (5V2N), two in the **BC-SAM** structure (4IJ8), two in the **BC-Inh1** structure (6BOZ), two in the **BC-Inh2** (5W1Y) structure, and twenty in the **TC** structures (1ZKK, 2BQZ, 3F9W, 3F9X, 3F9Y, 3F9Z). In total, 30 final simulation models were generated. The overall set of seed structures is summarized in *Supplementary file 1a*.

## Preservation of stereochemistry

During quality checks following the Ensembler automated modeling procedure, it was discovered that some of the final models showed incorrect Cα stereochemistry on some residues and/or cis-peptide bonds were present (using VMD 1.9.2; *Kruskal, 1964*), inspired by a previous study on a 15-amino-acid α-helix (*Schreiner et al., 2011*). This was determined to stem from homology modeling errors or flips during the energy minimization/implicit solvent refinement due to initial strain. This was solved by repetition of the whole Ensembler workflow for those models a number of times until no more chirality and/or cis-peptide conformational error was detected. This was not successful within a reasonable number of repeats for chain 'C' of the 1ZKK crystal structure and chain 'A' of the 3F9Z crystal structure for unknown reasons. These were replaced with another copy of chain 'A' of 1ZKK and chain 'B' of 3F9Z.

## Diversity of histidine tautomers

Both tautomers of His351 were present in the simulations - the π tautomer (*McNaught and Wilkinson, 1997*) for seed structures **TC_APO**, **BC-Inh2_APO**, and **BC-SAM_APO** (structures 26–28 in *Supplementary file 1a*), and the τ tautomer for all others. This was because, by default, for all models created in a given run Ensembler enforces the use of the same tautomer of His351, which is chosen by OpenMM mainly upon consideration of its optimal hydrogen bonding. The models used here were prepared in three separate Ensembler 1.0.5 runs, as the crystal structures became available. All data analysis was performed after removing the hydrogen atoms from the trajectories, so that all trajectories had the same topology. To ensure that the mixing of different tautomers did not have an effect on the overall model estimation, the estimates of kinetics of escape from a selection of macrostates were compared using subsets of the dataset containing the different tautomers of His351 (*Appendix 1—figure 23*). The discrete microstate trajectories were transformed into discrete macrostate trajectories, by changing the label of each microstate to the label of a macrostate to which it had the largest membership probability in the apo-SETD8 HMM. The count matrices for both trajectory subsets were obtained from PyEMMA 2.5.2 (*Scherer et al., 2015*) MSM objects at the Markovian lag time τ = 50 ns. Macrostates were ranked by the sums of counts out of (out-of-state-transitions) or remaining (self-transitions) in each macrostate and the ranks obtained from both count matrices were added. The three macrostates with the highest consensus ranks were chosen for comparison (macrostates A9, A1, and A4 in *Figure 4*). Count matrices were then estimated at lag times between 50–400 ns, at 50 ns intervals. The probability of remaining in each of the three macrostates at a given lag time $f$ was estimated by dividing the number of self-transition counts $M$ by the sum of self- and out-of-state-transitions $N$: $f = M/N$. The errors were estimated using the Beta distribution and assuming $Neff = N/\tau$ uncorrelated counts as $p(f)=Beta(Neff*f, Neff(1–f))$. This procedure was bootstrapped (assuming independent trajectories) 40 times at each lag time, the estimates were averaged, and the 95% confidence intervals of the mean were determined as 2.5[th] and 97.5[th] percentiles of the 95% confidence intervals of $p(f(t))$ traces.

## Folding@home simulations

The simulation seeds representing nine distinct conformers (30 distinct structures derived from multiple chains in each PDB structure, see SI for details) were used to initiate parallel distributed MD simulations on Folding@home (*Shirts and Pande, 2000*). Production simulations used the same Langevin integrator as the NpT equilibration described above, except that the Langevin collision rate was set to 1 ps$^{-1}$ to provide realistic heat exchange with a thermal bath while minimally perturbing dynamics. In total, 5,020 independent MD simulations were generated on Folding@home (*Shirts and Pande, 2000*): 600 simulations were produced from each seed conformation prepared from the five crystal structures, and 500 or 510 simulations for each seed conformation prepared from the four structural chimeras. At least 500 MD trajectories were produced for each seed conformation. 99.1% of the generated trajectories (4,976 trajectories) successfully reached 1 μs each (see *Appendix 1—figure 1* for length distribution histogram), resulting in 5.058 ms of aggregate simulation time and 10,115,617 frames. This amount of simulation time corresponds to ~231 GPU-years on an NVIDIA GeForce GTX 980 processor. Conformational snapshots (frames) were stored at an interval of 0.5 ns/frame for subsequent analysis. Prior to data analysis, the first 50 frames (25 ns) of each trajectory were discarded to allow the trajectories to relax away from their initial seed

conformations. On initial analysis of the RMSDs of the trajectories to their starting frames, one trajectory showed the protein unfolding and was removed from the dataset. The resulting final dataset contained 5,019 trajectories, 4.931 ms of aggregate simulation time, and 9,862,657 frames. This trajectory dataset without solvent is available via the Open Science Framework at https://osf.io/2h6p4/ . The code used for the generation and analysis of the molecular dynamics data is available via a Github repository at https://github.com/choderalab/SETD8-materials.

## Optimal hyperparameter selection for featurization and tICA

To select the optimal featurization of the data for subsequent Markov state model (MSM) analysis, we used variational scoring (*McGibbon and Pande, 2015b*; *Noé and Nüske, 2013*; *Nüske et al., 2014*; *Wu and Noe, 2017*) combined with cross-validation (*Husic et al., 2016*) to evaluate model quality, consistent with modern MSM construction practice (*Husic et al., 2016*). To evaluate a large set of hyperparameters to achieve optimal featurization, a reduced dataset subsampled to 5 ns/frame intervals (986,464 frames, 10% of the dataset) was used for computational feasibility. The following trajectory featurization choices were assessed: *a*) all residue–residue distances (calculated as the closest distance between the heavy atoms of two residues separated in sequence by at least two neighboring residues) that cross a 0.4 nm contact threshold in either direction at least once (yielding 6,567 of 12,720 total residue-residue distances); *b*) a transformed version of (a) used by MSMBuilder (*Harrigan et al., 2017*) to emphasize short-range distances in the proximity of residue-residue contact via *Equation. 4*, with *steepness* = 5 nm$^{-1}$ and *center* = 0.5 nm; *c*) backbone (phi, psi) and side-chain (chi1) dihedral angles, with each angle featurized as its sine and cosine (yielding 920 total features).

$$logistic\ transformed\ distance = 1/(1 + \exp(steepness \times (distance - center)))\tag{4}$$

To identify the optimal featurization, we used a 50:50 shuffle-split cross-validation scheme to evaluate various model hyperparameters while avoiding overfitting. In this scheme, subsets of the groups of trajectories initiated from the same conformation (RUNs − see *Supplementary file 1a* for further explanation) are randomly split into training and test sets of 2,509 and 2,510 trajectories respectively, using scikit-learn 0.9.1 (*Pedregosa et al., 2011*). All further steps until scoring were conducted by fitting the model to the training set only, then transforming the test set according to this model. Scoring was based on the sum of the top 10 squared-eigenvalues of the transition matrix (rank-10 VAMP-2 [*Wu and Noe, 2017*]). Model scores are reported below as means with standard deviations over five shuffle-splits.

To evaluate each featurization choice, the data were projected into a kinetically relevant space using tICA (*Pérez-Hernández et al., 2013*), retaining all tICs, at lag times of either 5 or 50 ns, with either kinetic (*Noé and Clementi, 2015*) or commute mapping (*Noé et al., 2016*). Each of the tICA outputs was discretized using k-means clustering into 50, 100, 500, or 1000 microstate clusters (see *Supplementary file 1b* for the summary of options assessed). Featurization was performed using MDTraj 1.8 (*McGibbon et al., 2015a*) and PyEMMA 2.4 (*Scherer et al., 2015*), tICA was performed with PyEMMA 2.4 (*Scherer et al., 2015*), and clustering was performed with PyEMMA 2.5.1 (*Scherer et al., 2015*). MSMs at a lag time of 50 ns were constructed with PyEMMA 2.5.1 (*Scherer et al., 2015*) using discrete microstate trajectories from the training set and scored on the test set trajectories. To obtain standard deviations indicative of out-of-sample model performance, this shuffle-split model evaluation procedure was repeated 5 times with different random divisions of the dataset into training and test sets. The data showed (*Appendix 1—figures 2*, *3*) that the four individual models with highest average scores were featurized with dihedral angles (featurization *c*; scores: 9.68 (SD = 0.05), 9.68 (SD = 0.05), 9.63 (SD = 0.03), 9.62 (SD = 0.02)), while the highest median score over all models was the residue-residue distance featurization (the median score of 8.20 (mean = 7.98, SD = 1.11) for featurization *a*; 8.06 (mean = 7.49, SD = 2.07) for featurization *c*; 6.99 (mean = 6.47, SD = 2.13) for featurization *b*). Therefore, both featurizations *a* and *c* were further evaluated on the full dataset to determine the optimal model. For both featurizations, commute mapping resulted in significantly higher scores (*Appendix 1—figures 4*, *5*) than kinetic mapping, hence commute mapping was used for the full dataset. The shorter tICA lag time (5 ns) was used for the full dataset, as there was no significant difference in scores between 5 and 50 ns (*Appendix 1—figures 4*, *5*).

## Final featurization and microstate number selection

To determine the optimal number of microstates, we again used variational scoring (*McGibbon and Pande, 2015b*; *Noé and Nüske, 2013*; *Nüske et al., 2014*; *Wu and Noe, 2017*) combined with cross-validation (*Husic et al., 2016*) to evaluate model quality. The full dataset (4.931 ms, 0.5 ns/frame, 9,862,657 frames) was separately featurized with the top-scoring feature sets: 6,567 distances (featurization *a* above) and 920 dihedral angles (featurization *c* above). As for the featurization selection, we used the 50:50 shuffle-split cross-validation scheme, using the same five data splits. All further steps until scoring were conducted by fitting the model to the training set only, subsampled to 5 ns/frame intervals for computational feasibility, then transforming the full training set and the test set according to this model. Data were projected into the tICA space using a lag time of 5 ns. The tICs were scaled by commute mapping, with subsequent clustering operations using a sufficient number of tICs necessary to explain 95% of the total kinetic content. The tICA outputs were discretized using k-means clustering into 100, 500, 1000, 2000, 3000, 4000, or 5000 microstate clusters (see *Supplementary file 1c* for the summary of options assessed). Featurization was performed with MDTraj 1.8 (*McGibbon et al., 2015a*) and PyEMMA 2.4 (*Scherer et al., 2015*), tICA was performed with PyEMMA 2.4 (*Scherer et al., 2015*), and clustering was performed with PyEMMA 2.5.1 (*Scherer et al., 2015*). MSMs were constructed with PyEMMA 2.5.1 (*Scherer et al., 2015*) from the discrete trajectories of the training set using a lag time of 50 ns and subsequently scored on the test set, using the rank-10 VAMP-2 (*Wu and Noe, 2017*) score. The highest scoring model (*Appendix 1—figure 6*) had dihedral features (featurization *c* above) and 100 microstates (VAMP-2 = 9.25 (SD = 0.32)). tICA and k-means clustering were refitted to the full dataset subsampled to 5 ns/frame intervals for computational feasibility. Keeping the number of tICs necessary to explain 95% of the total kinetic content resulted in 466 tICs used for k-means clustering. The full dataset was then transformed to give the final discretized trajectories at 0.5 ns/frame intervals. Checking the convergence of the implied timescales validated the choice of the MSM lag time (*Appendix 1—figure 7*). The Chapman-Kolmogorov test (*Prinz et al., 2011*) was then conducted on the MSM to examine the self-consistency of the model (*Appendix 1—figure 8*). To aid structural interpretation, the 10 frames closest to each of the 100 microstate cluster centers were extracted from the dataset. The RMSDs of the 10 frames in each microstate were calculated with C-alpha atoms only, after first superposing each frame onto the reference structure using only the C-alpha atoms of the conformationally homogenous SET motifs (residues 257–290 and 327–376). To quantify the structural diversity of each microstate, a sample of 100 frames was randomly drawn from each. The C-alpha RMSD (after superposition of the SET motifs only) of each frame versus all other 99 frames was then calculated, and the minimum average RMSD over all 100 reference frames was reported (*Supplementary file 1d*).

## Coarse-graining to kinetically metastable macrostates

To coarse-grain the MSM into a small number of kinetically metastable macrostates, a Hidden Markov Model (HMM) was constructed from the discrete trajectories of the optimal model above using PyEMMA 2.5.1 (*Scherer et al., 2015*). Increasing numbers of macrostates were explored and interpreted structurally by assigning the 10 frames closest to each of the 100 microstate cluster centers to the macrostate to which they had the largest fractional membership. We chose the minimal number of macrostates that achieved increasing structural separation of the distinct SET-I and post-SET motif configurations and hence constructed a 24-macrostate HMM. The resulting HMM provides both a macrostate-to-macrostate transition matrix and a fractional membership of each microstate to each kinetically metastable macrostate. Checking the convergence of the HMM implied timescales further validated the choice of the MSM/HMM lag time (*Appendix 1—figure 9*). To preserve kinetic relationships between macrostates in a two-dimensional representation, the log-inverse fluxes between all pairs of macrostates (calculated using the third power of the transition matrix to eliminate sparsity) were embedded in two dimensions using iterative multidimensional scaling (MDS) (*Borg and Groenen, 2005*; *Kruskal, 1964*; *Kruskal, 1979*) with scikit-learn 0.9.1 (*Pedregosa et al., 2011*). MDS was repeated at least 50 times with random initializations and the projection that leads to a figure with the fewest crossings of inter-state flux arrows was selected. To aid structural interpretation, the 10 frames closest to each of the 100 microstate cluster centers were assigned to the macrostate to which they had the largest fractional membership. The RMSDs of the macrostates to the homology models derived from all five crystal structures generated by Ensembler were

calculated by averaging the RMSDs of all 10 frames in each microstate as described above, then for each macrostate taking the mean over all microstates weighted by the HMM observation probabilities. To quantify the structural diversity of each macrostate, a sample of 100 frames was drawn from each macrostate with probabilities for each frame given by the observation probability of the frame's microstate from the given macrostate divided by the total number of frames in the frame's microstate. The C-alpha RMSD (after superpose of the SET motifs only) of each frame versus all other 99 frames was then calculated, and the minimum average RMSD over all 100 reference frames was reported (*Supplementary file 1e*).

## Molecular dynamics (MD) simulations of SAM-bound SETD8

### Preparation of molecular dynamics (MD) simulations

All-atom models of the same 162-residue SETD8 fragment in complex with SAM were prepared in a similar manner as apo-SETD8 except that a manual pipeline was used instead of Ensembler. Briefly, two available cofactor-bound crystal structures were used to generate two seed structures for simulation: 4IJ8 (the crystal structure of the binary complex of SETD8 with SAM) and 2BQZ (the crystal structure of the tertiary complex of SETD8 with SAH and a methylated H4K20 peptide). Among the available tertiary complex structures (1ZKK, 2BQZ, 3F9W, 3F9X, 3F9Y, 3F9Z), 2BQZ was selected for MD simulations because of the following conditions met simultaneously: no mutations present, minimum number of missing residues requiring modeling (1), methylated lysine resolved on the histone peptide (for future simulations of the tertiary complex). Protein chains 'A' of both structures were used. Mutations in 4IJ8 were corrected to the reference sequence, and missing protein residues and atoms were added using PDBFixer 1.3 (*Eastman, 2013*; *Eastman et al., 2013*). To replace SAH with SAM in the 2BQZ model, the coordinates of SAM were copied from 4IJ8, where all SAM atoms were resolved, after aligning the common atoms in SAM and SAH with MDTraj 1.7.2 (*McGibbon et al., 2015a*). The peptide and SAH were then removed from 2BQZ. Using OpenMM 7.0.1 (*Eastman et al., 2017*), protonation states appropriate for pH 7 were assigned with openmm.app. modeller. SAM was modeled in the +1 cationic form at its sulfonium center and the zwitterionic form at its α-amino acid moiety. GAFF force field parameters (*Wang et al., 2004*) and AM1-BCC (*Jakalian et al., 2002*) charges were assigned using Antechamber (*Wang et al., 2006*) from Amber-Tools 14 (*Case et al., 2014*) with missing parameters assigned using Antechamber's ParmChk2. The SAM parameter files were then converted from the Amber format to the OpenMM XML format using the conversion script distributed with the openmm-forcefields package (*Chodera, 2018*). The systems were solvated in cubic water boxes with at least 1 nm padding and neutralized with a minimal amount of NaCl. This resulted in the final systems containing 34,556 atoms (system prepared from 4IJ8) and 35,588 atoms (system prepared from 2BQZ). These were energy-minimized with 10 kJ/mol tolerance and relaxed for 1 fs in the NVT (T = 10 K) ensemble using the OpenMM Langevin integrator with a 0.01 fs timestep, and 91 ps$^{-1}$ collision rate. Nonbonded interactions were treated with the reaction field method only during minimization (due to its increased stability over PME when steric clashes needed to be resolved following introduction of mutations) (*Barker and Watts, 1973*) at a cutoff of 0.9 nm. The systems were then equilibrated for 10 ns in the NpT (p = 1 atm, T = 300 K) ensemble using the OpenMM Langevin integrator, the PME nonbonded method, a Monte Carlo molecular-scaling barostat with an update interval of 25 steps, and packaged with OpenMM 6.3.1 (*Eastman et al., 2013*) as seeds for production simulation on Folding@home (*Shirts and Pande, 2000*). All other force field parameters and simulation settings were as previously described for apo-SETD8.

### Preservation of native configuration

The N-terminal residue Ser232 and the Ser232–Arg233 amide bond were modeled with *D*-configuration and *cis*-configuration, respectively, upon preparing SAM-bound SETD8 models by PDBFixer. No further correction was conducted, because this residue does not participate in functionally relevant conformational dynamics and makes minimal interactions with the rest of the protein. The rest of the sequence in the models adopts native configuration.

## Folding@home simulations

In total, 1000 independent MD simulations were generated on Folding@home: 500 each for the two seed structures prepared above. Simulations employed the same settings as for NpT production of apo-SETD8. 99.7% of the generated trajectories (997 trajectories) successfully reached 1 µs each (see *Appendix 1—figure 10* for length distribution), resulting in 1.003 ms of aggregate simulation time and 2,005,945 frames. This amount of simulation time corresponds to ~46 GPU years on an NVIDIA GeForce GTX 980 processor. Prior to data analysis, the first 25 ns of each trajectory were discarded to allow the trajectories to relax away from the initial equilibrated configurations. One trajectory was shorter than the length being discarded and was removed from the dataset. The resulting final dataset contained 999 trajectories, 0.978 ms of aggregate simulation time, and 1,955,965 frames. This trajectory dataset without solvent is available via the Open Science Framework at https://osf.io/2h6p4. The code used for the generation and analysis of the molecular dynamics data is available via a Github repository at https://github.com/choderalab/SETD8-materials.

## Coarse-graining to kinetically metastable macrostates

To construct a Hidden Markov model of SAM-bound SETD8, the full dataset (0.978 ms, 0.5 ns/frame, 1,955,965 frames) was featurized using the final model generated from apo-SETD8 (featurization *c*, backbone and sidechain dihedral features). The data were projected into the tICA space derived from the apo-SETD8 simulations and assigned to the 100 k-means microstates of apo-SETD8. The SAM-bound SETD8 trajectories populated 67 of the 100 microstates of apo-SETD8. An MSM with a 50 ns lag time was constructed, and the Chapman-Kolmogorov test (*Prinz et al., 2011*) was conducted to examine the self-consistency of the model (*Appendix 1—figure 12*). Finally, a Hidden Markov model (HMM) was constructed for a 50-ns lag time using 10 macrostates (the minimal number of macrostates to achieve increasing structural separation between distinct SET-I and post-SET configurations was chosen in the same way as for apo-SETD8). As for apo-SETD8, log-inverse fluxes between macrostates were used to construct a two-dimensional representation, and the 10 frames closest to the microstate cluster centers were assigned to the macrostate to which they had the highest fractional membership to aid structural interpretation. Prior to visualization, frames were re-imaged with MDTraj 1.8 (*McGibbon et al., 2015a*) to ensure SETD8 was centered and the SAM ligand was in the same unit cell. As for apo-SETD8, microstate C-alpha RMSDs to the homology models derived from all five crystal structures were calculated by averaging the RMSDs of all 10 frames in each microstate, and structural diversity was quantified by the reference frame with the minimum average RMSD of each microstate (*Supplementary file 1f*). Further, as for apo-SETD8, macrostate C-alpha RMSDs were calculated by weighted mean over microstate average RMSDs, and structural diversity was quantified by the reference frame with the minimum average RMSD of each macrostate (*Supplementary file 1g*).

## Cancer-associated mutant apo-SETD8 simulations

### Selection of SETD8 mutants

The MSKCC-internal cBioPortal Cancer Genomics Database was searched in August of 2017 to map cancer-associated SETD8 mutations. The resulting mutations except the R365* nonsense mutation in the region of residues 232–393 (191–352 in the database), which corresponds to the catalytic domain of SETD8 used in our biochemical experiments, were selected for MD simulations (see *Supplementary file 1n* for the list of all mutants).

### Preparation of molecular dynamics (MD) simulations

All-atom models of the same 162-residue SETD8 fragment with each of 24 cancer-associated single mutations (including one deletion giving a 161-residue fragment) identified from the cBioPortal for Cancer Genomics (*Cerami et al., 2012*; *Cheng et al., 2015*; *Gao et al., 2013*) were prepared in an analogical way to apo-SETD8 using Ensembler 1.0.5. The mutants prepared are summarized in *Supplementary file 1n* (#1–21, 23–25). As we aimed to gain a direct interpretation of the influence of these mutations on the enzymatic activity of SETD8 and the cost of direct simulations of all mutants mapped onto all crystal structures is computationally prohibitive, only a single chain of the TC structure was used as the template. To choose the particular chain out of the 18 TC chains used in the apo-SETD8 simulations (*Supplementary file 1a*), the homology models of all the TC chains

generated by Ensembler were projected into the apo-SETD8 tICA space using PyEMMA 2.4 (*Scherer et al., 2015*). The distances between the points in the tICA space were then calculated with SciPy 1.0 (*McKinney, 2010*) and chain 'A' of the 1ZKK structure, which had the smallest average distance to all others, was selected. The modeling procedure was the same as for apo-SETD8, except the appropriately mutated sequences were passed to Ensembler (*Parton et al., 2016*). Briefly, homology models were created with MODELLER 9.16 (*Sali and Blundell, 1993*), protonation states appropriate for pH 7 were assigned with OpenMM 6.3.1 (*Eastman et al., 2013*), the models were then energy-minimized for 20 steps and relaxed with 100 ps of implicit solvent dynamics. The proteins were then solvated in cubic boxes with at least 1 nm padding and neutralized with a minimal amount of NaCl. This resulted in the final systems containing between 35,185 and 35,208 atoms. These were equilibrated for 5 ns and packaged as seeds for production simulation on Folding@home. All force field parameters and simulation settings were as previously described for wild-type apo-SETD8.

## Preservation of stereoselectivity

As for the wild-type apo-SETD8 models, during quality checks following the Ensembler automated modeling procedure, it was discovered that some of the final models showed incorrect Cα chirality on some residues and/or cis-peptide bonds were present (using VMD 1.9.2; *Kruskal, 1964*), inspired by a previous study on a 15-amino-acid α-helix (*Schreiner et al., 2011*). This was determined to be due to homology modeling errors or flips because of initial strain during the energy minimization/implicit solvent refinement. This was solved by repetition of the whole Ensembler 1.0.5 workflow for those models a number of times until no more chirality and/or cis-peptide issues were detected. This was not successful within a reasonable number of repeats for mutant I247L, for which the error was introduced by the MODELLER homology modeling and was finally solved by replacing the *allhmodel* class (explicit hydrogen modeling, default in Ensembler 1.0.5) with the *automodel* class (implicit hydrogen modeling, default in Ensembler 1.0.6, which was used).

## Folding@home simulations

In total, 960 simulations were generated on Folding@home: 40 for each of the mutants. Simulations employed the same settings as for NpT production of wild-type apo-SETD8. 99.7% of the generated trajectories (957 trajectories) successfully reached 1 μs each (see *Appendix 1—figure 24* for length distribution), resulting in the final aggregate dataset of 0.966 ms and 1,931,849 frames. This amount of simulation time corresponds to ~44 GPU years on an NVIDIA GeForce GTX 980 processor. This trajectory dataset without solvent is available via the Open Science Framework at https://osf.io/2h6p4. The code used for the generation and analysis of the molecular dynamics data is available via a Github repository at https://github.com/choderalab/SETD8-materials.

## Contact map analysis

Prior to data analysis, the first 750 ns of each trajectory were discarded to allow for successful metastable transitions out of the wild-type kinetic basin. This number was chosen arbitrarily for ensuring a reasonable balance between eliminating bias from the initial configurations in mutant trajectories while keeping a reasonable amount of data for analysis. For the remaining frames of each mutant, all residue-residue distances (calculated as the closest distance between the heavy atoms of two residues) for which the two residues are separated in sequence by at least two other residues (yielding 12,720 residue-residue distances) were calculated with MDTraj 1.8 (*McGibbon et al., 2015a*). These were converted into binary contact maps by replacing all distances below the 0.4 nm contact threshold with 1's and all other distances with 0's, and casting into a square-form matrix. These were then averaged over all frames of each mutant, yielding one contact map for each mutant. In the same way, the wild-type contact map was calculated using the 60 wild-type apo-SETD8 trajectories started from chain 'A' of the 1ZKK structure (*Supplementary file 1a*). The wild-type contact map was then subtracted from the mutant contact maps to generate one absolute contact change map for each mutant. In the one case of amino acid deletion, all contact changes corresponding to that residue position were set to zero. Relative contact changes were also calculated by dividing the absolute contact change value by the wild-type contact value and taking the modulus of the result. The result

was set to zero where the wild-type value was zero. Contacts with absolute changes of more than 0.2 and relative changes of more than three were selected for further structural annotations.

## Extraction of hypothetical new conformations

For each mutant, of the contacts that showed an absolute fractional change of more than 0.2 or more negative than −0.2, up to five contacts with the most positive changes ('positive contacts') and up to five contacts with the most negative changes ('negative contacts') were noted. For each trajectory (after discarding the first 750 ns) of a given mutant, every 10th (every 20th if more than 7,500 such frames were present) of the frames that had all 'positive contacts' present or none of the 'negative contacts' present was extracted. These frames were manually inspected using PyMOL 1.8.4 (Schrödinger, LLC) against the representative conformations of the wild-type apo-SETD8 microstates (the 10 frames closest to each of the microstate cluster centers) and frames that displayed similar SET-I and post-SET motif configurations to any of the wild-type conformations were discarded. For the remaining frames, the C-alpha RMSDs to all frames of the wild-type apo-SETD8 dataset sub-sampled to 5 ns/frame were calculated using MDTraj 1.8 (*McGibbon et al., 2015a*) and the wild-type frames with the lowest RMSD to each mutant frame were extracted. The mutant frames were manually inspected using PyMOL 1.8.4 (Schrödinger, LLC) against the extracted wild-type frames and further mutant frames similar to the wild-type frames were discarded. The remaining frames for each mutant were clustered based on the manual inspection and their C-alpha RMSDs to all frames of the given mutant (without discarding the first 750 ns of each trajectory) were calculated using MDTraj 1.8 (*McGibbon et al., 2015a*). For each cluster of the hypothetical new conformations, every 10th of all mutant frames with RMSDs below the 0.3 nm, 0.35 nm, and 0.4 nm thresholds to any of the cluster frames were extracted and manually inspected in PyMOL 1.8.4 (*Schrödinger LLC, 2019*). The 0.3 nm threshold gave good structural similarities and only a small number of false positives (frames that were not sufficiently similar to the originally chosen hypothetical new conformations) were discarded, while the other two thresholds introduced too many false positives. Hence the remaining frames extracted at the 0.3 nm threshold were taken as the final clusters of hypothetical new conformations. To further confirm that the discovered conformations were relevant and not simply an artifact of additional sampling, the rate of new microstate discovery was compared between equivalent cumulative aggregate simulation lengths (corresponding to a uniform initial fraction of all trajectories in the dataset). PyEMMA 2.5.1 (*Scherer et al., 2015*) was used for all of the following steps. The wild-type and mutant datasets were featurized with sine/cosine of the same set of backbone and side chain dihedral angles (accounting for the angles not present after mutations). The wild-type + mutant data combined were then projected into the tICA space, using a lag time of 5 ns with commute mapping with 468 tICs sufficient to explain 95% of the kinetic content. These were then jointly clustered into 2,000 microstates using k-means. This number of microstates was chosen by examining increasing numbers of microstates, until the number of microstates populated by mutants but not wild-type was larger than the number of mutants in the dataset (we found 79 such microstates for 24 mutants). The number of new microstates discovered for equal amounts of data (~1 ms aggregate simulation time) from the final portion of the WT trajectories and from mutant trajectories were plotted (*Appendix 1—figure 25*), showing that the mutant dataset rapidly discovers 79 new microstates at a rate that far outstrips the discovery rate of new wild-type conformations.

## Calculation of microstate coverage

To quantify how the diversity of starting conformations influences the number of microstates observed out of the total of 100, the apo-SETD8 discrete trajectories were split into nine sets corresponding to their starting conformations. All logical relations between the sets were generated and the numbers of microstates explored in each intersection were counted in order to produce Venn diagrams of microstate coverage. Analogically, the SAM-bound SETD8 discrete trajectories were split into two sets and the microstate coverage was evaluated. Further, to quantify how the number and the length of trajectories influence the number of microstates observed in addition to the diversity of starting conformations, all combinations of all possible lengths of the five apo-SETD8 sets started from crystal structures were enumerated. Appropriate sets out of the four originating from structural chimeras were added to those combinations which contained the appropriate SET-I and post-SET motif configurations for the formation of those chimeras. Also, if a combination contained

either of the BC-Inh1 or BC-Inh2 sets (SET-I configuration I1), and the BC-SAM set (post-SET configuration P1), the TC set (configurations I1-P1) was added as it could be generated as a structural chimera. For all combinations that resulted, the microstate coverage was assessed at all trajectory numbers between 0 to all trajectories in the combination at intervals of 50 trajectories, and simultaneously at all maximum trajectory lengths between 0 to the length of the longest trajectory in the combination at intervals of 50 ns. The desired number of trajectories was randomly drawn from all trajectories in the combination without replacement and the trajectories were trimmed to the desired maximum length. The number of microstates observed was then calculated. This was repeated five times with different draws of trajectories and the results of the five draws were averaged. Analogically, the microstate coverage at increasing trajectory numbers and trajectory lengths was evaluated for the two SAM-bound SETD8 sets.

### List of used software

The Anaconda Python (*Millman and Aivazis, 2011*; *Oliphant, 2007*) distribution with Python 2.7, 3.5, or 3.6 was used for all programming. Conda was used for package management. The IPython (*Perez and Granger, 2007*) shell and the Jupyter Notebook (*Kluyver et al., 2016*) environment were used for interactive scripting and data analysis. Data were managed with the NumPy (multiple versions) (*Oliphant, 2006*) and pandas (multiple versions) (*McKinney, 2010*) libraries. Mathematical operations were performed with the NumPy library or Python built-in functions. Figures were made with PyMOL 1.8.4 (*Schrödinger LLC, 2019*), matplotlib 2.2.2 (*Hunter, 2007*), seaborn 0.8.1 (*Waskom et al., 2017*), MSMExplorer 1.1 (*Harrigan et al., 2017*), and PyEMMA 2.5.1 (*Scherer et al., 2015*).

### Code and data availability

The molecular dynamics datasets generated and analyzed in this study are available via the Open Science Framework at https://osf.io/2h6p4. The code used for the generation and analysis of the molecular dynamics data is available via a Github repository at https://github.com/choderalab/SETD8-materials (*Wiewiora, 2019*; a copy archived at https://github.com/elifesciences-publications/SETD8-materials).

## Acknowledgements

The authors thank for National Institute of General Medical Sciences (ML: R01GM096056, R01GM120570, 1R35GM131858; JDC: R01GM121505; JJ: R01GM122749; YGZ, R01GM126154), National Cancer Institute (ML, JDC: 5P30 CA008748), Starr Cancer Consortium (ML, JDC), MSKCC Functional Genomics Initiative (ML, JDC), the Sloan Kettering Institute (ML, JDC, KAB), Mr. William H Goodwin and Mrs. Alice Goodwin Commonwealth Foundation for Cancer Research, and the Experimental Therapeutics Center of Memorial Sloan Kettering Cancer Center (ML), Tri-Institutional Therapeutics Discovery Institute (ML), and Louis VGerstner Young Investigator Award (JDC), National Natural Science Foundation of China (CL: 91853205, 81625022; KC: 81430084; HJ: 21820102008), K. C. Wong Education Foundation (CL), Science and Technology Commission of Shanghai Municipality (HJ: 18431907100; CL: 19XD1404700), National Science & Technology Major Project of China (HJ: 2018ZX09711002), Chinese Academy of Sciences (CL: XDA12020353), the Tri-Institutional PhD Program in Chemical Biology (RPW and SC), Peer Reviewed Cancer Research Program of the Department of Defense (RPW: W81XWH-17-1-0412) for research supports; the Marie-Josée and Henry R Kravis Center for Molecular Oncology, and the Molecular Diagnostics Service in the Department of Pathology for the access to tumor mutation data via cBioPortal; Carolina Adura at High Throughput and Spectroscopy Resource Center at The Rockefeller University for the assistance to ITC experiments; Henry Zebroski III and Susan Powell at Proteomics Resource Center at The Rockefeller University for peptide synthesis; the Folding@home project for computational resources; Kanishk Kapilashrami, Josh Fass, Sonya Hanson, Frank Noé, Simon Olsson, and Martin Scherer for insightful discussions or software support; the peer reviewers—Gregory Bowman, Erik Lindahl, and an anonymous reviewer for comprehensive advice. The Structural Genomics Consortium is a registered charity (no. 1097737) that receives funds from AbbVie; Bayer Pharma AG; Boehringer Ingelheim; Canada Foundation for Innovation; Eshelman Institute for Innovation; Genome Canada; Innovative Medicines Initiative (EU/EFPIA) (ULTRA-DD grant no. 115766); Janssen; Merck and Co.;

Novartis Pharma AG; Ontario Ministry of Economic Development and Innovation; Pfizer; São Paulo Research Foundation-FAPESP; Takeda; and the Wellcome Trust. Diffraction data for the **BC-Inh2** complex were collected using a Structural Biology Center (SBC) beam line at the Advanced Photon Source, Argonne National Laboratory. SBC-CAT is operated by UChicago Argonne, LLC, for the U.S. Department of Energy, Office of Biological and Environmental Research under contract DE-AC02-06CH11357. The Canadian Light Source is supported by the Canada Foundation for Innovation, Natural Sciences and Engineering Research Council of Canada, the University of Saskatchewan, the Government of Saskatchewan, Western Economic Diversification Canada, the National Research Council Canada, and the Canadian Institutes of Health Research. The X-ray experiment of **BC-Inh1** was conducted with NE-CAT beam line 24-ID-E (GM103403) and an Eiger detector (OD021527) at the APS (DE-AC02-06CH11357).

## Additional information

### Competing interests

John D Chodera: declares that no competing interests exist. In the interests of transparency they wish to make the following disclosures: John D Chodera was a member of the Scientific Advisory Board for Schrodinger, LLC during part of this study; is a member of the Scientific Advisory Board of OpenEye Scientific Software The Chodera laboratory receives or has received funding from multiple sources, including the National Institutes of Health, the National Science Foundation, the Parker Institute for Cancer Immunotherapy, Relay Therapeutics, Entasis Therapeutics, Silicon Therapeutics, EMD Serono (Merck KGaA), AstraZeneca, XtalPi, the Molecular Sciences Software Institute, the Starr Cancer Consortium, the Open Force Field Consortium, Cycle for Survival, a Louis V. Gerstner Young Investigator Award, and the Sloan Kettering Institute. A complete funding history for the Chodera lab can be found at http://choderalab.org/funding. The other authors declare that no competing interests exist.

### Funding

| Funder | Grant reference number | Author |
| --- | --- | --- |
| UNC Eshelman Institute for Innovation | | Cheng Luo |
| Science and Technology Commission of Shanghai Municipality | 19XD1404700 | Cheng Luo |
| Science and Technology Commission of Shanghai Municipality | 18431907100 | Hualiang Jiang |
| Ontario Ministry of Economic Development and Innovation | | Peter J Brown |
| Novartis Pharma | | Peter J Brown |
| National Institute of General Medical Sciences | R35GM131858 | Minkui Luo |
| Merck | | Peter J Brown |
| National Institute of General Medical Sciences | R01GM120570 | Minkui Luo |
| Janssen Pharmaceuticals | | Peter J Brown |
| National Natural Science Foundation of China | 81625022 | Cheng Luo |
| National Institute of General Medical Sciences | R01GM126154 | Yujun George Zheng |
| National Science & Technology Major Project of China | 2018ZX09711002 | Hualiang Jiang |

| National Natural Science Foundation of China | 21820202008 | Hualiang Jiang |
|---|---|---|
| National Natural Science Foundation of China | 81430084 | Kaixian Chen |
| National Institute of General Medical Sciences | R01GM121505 | John D Chodera |
| National Natural Science Foundation of China | 91853205 | Cheng Luo |
| National Institute of General Medical Sciences | R01GM122749 | Jian Jin |
| Mr. William H. Goodwin and Mrs. Alice Goodwin Commonwealth Foundation for Cancer Research and the Experimental Therapeutics Center of Memorial Sloan Kettering Cancer Center | | Minkui Luo |
| Tri-Institutional Therapeutics Discovery Institute | | Minkui Luo |
| National Cancer Institute | 5P30 CA008748 | John D Chodera Minkui Luo |
| National Institute of General Medical Sciences | R01GM096056 | Minkui Luo |
| Starr Cancer Consortium | | John D Chodera Minkui Luo |
| Memorial Sloan-Kettering Cancer Center | Functional Genomics Initiative | John D Chodera Minkui Luo |
| The Sloan Kettering Institute | | Kyle A Beauchamp John D Chodera Minkui Luo |
| Memorial Sloan-Kettering Cancer Center | Louis V. Gerstner Young Investigator Award | John D Chodera |
| K. C. Wong Education Foundation | | Cheng Luo |
| Chinese Academy of Sciences | XDA12020353 | Cheng Luo |
| The Tri-Institutional PhD Program in Chemical Biology | | Shi Chen Rafal P Wiewiora |
| U.S. Department of Defense | Peer Reviewed Cancer Research Program W81XWH-17-1-0412 | Rafal P Wiewiora |
| AbbVie | | Peter J Brown |
| Bayer Pharma AG | | Peter J Brown |
| Boehringer Ingelheim | | Peter J Brown |
| Genome Canada | | Peter J Brown |
| Innovative Medicines Initiative | | Peter J Brown |
| Canada Foundation for Innovation | | Peter J Brown |
| Pfizer | | Peter J Brown |
| São Paulo Research Foundation | | Peter J Brown |
| Takeda Pharmaceutical Company | | Hua Zou Robert J Skene Peter J Brown |
| Wellcome Trust | | Peter J Brown |

The funders had no role in study design, data collection and interpretation, or the decision to submit the work for publication.

## Author contributions

Shi Chen, Conceptualization, Resources, Data curation, Formal analysis, Validation, Investigation, Visualization, Methodology, Writing—original draft, Project administration, Writing—review and editing; Rafal P Wiewiora, Conceptualization, Resources, Data curation, Software, Formal analysis, Funding acquisition, Validation, Investigation, Visualization, Methodology, Writing—original draft, Writing—review and editing; Fanwang Meng, Conceptualization, Investigation; Nicolas Babault, Investigation, Methodology; Anqi Ma, Wenyu Yu, Kun Qian, Hao Hu, Hua Zou, Gil Blum, Fabio Pittella-Silva, Wolfram Tempel, Resources, Investigation, Methodology; Junyi Wang, Resources, Investigation, Characterized inh1/2 as covalent SETD8 inhibitors, measured their affinities to SETD8 and conducted the quality control for the compounds used for X-ray crystallography; Shijie Fan, Conceptualization, Investigation, Generated key preliminary modeling data, which allowed formulating this project, Participated in the conceptual development of this project; Kyle A Beauchamp, Resources, Software, Investigation, Methodology; Hualiang Jiang, Resources, Supervision, Funding acquisition, Project administration; Kaixian Chen, Conceptualization, Resources, Supervision, Funding acquisition, Project administration; Robert J Skene, Resources, Supervision, Funding acquisition, Methodology, Writing—review and editing; Yujun George Zheng, Resources, Formal analysis, Supervision, Funding acquisition, Methodology, Writing—review and editing; Peter J Brown, Resources, Formal analysis, Supervision, Funding acquisition, Methodology, Project administration, Writing—review and editing; Jian Jin, Resources, Supervision, Funding acquisition, Methodology, Project administration, Writing—review and editing; Cheng Luo, Conceptualization, Formal analysis, Supervision, Funding acquisition, Methodology, Project administration, Writing—review and editing; John D Chodera, Conceptualization, Resources, Data curation, Software, Formal analysis, Supervision, Funding acquisition, Validation, Visualization, Methodology, Writing—original draft, Project administration, Writing—review and editing; Minkui Luo, Conceptualization, Resources, Data curation, Formal analysis, Supervision, Funding acquisition, Validation, Visualization, Methodology, Writing—original draft, Project administration, Writing—review and editing

## Author ORCIDs

Shi Chen https://orcid.org/0000-0002-5860-2616
Rafal P Wiewiora https://orcid.org/0000-0002-8961-7183
Fanwang Meng https://orcid.org/0000-0003-2886-7012
Kun Qian https://orcid.org/0000-0003-1132-2374
Fabio Pittella-Silva https://orcid.org/0000-0002-9644-7098
Robert J Skene http://orcid.org/0000-0002-1482-6546
Peter J Brown https://orcid.org/0000-0002-8454-0367
Jian Jin http://orcid.org/0000-0002-2387-3862
John D Chodera https://orcid.org/0000-0003-0542-119X
Minkui Luo https://orcid.org/0000-0001-7409-7034

## Decision letter and Author response

Decision letter https://doi.org/10.7554/eLife.45403.074
Author response https://doi.org/10.7554/eLife.45403.075

## Additional files

### Supplementary files

• Supplementary file 1. The table files associated with computational modeling and biochemical characterization of SETD8. (**a**) All models used in the simulation, their origin and numbers of trajectories generated (apo simulations). *RUN is a collection of CLONEs, all started from the same initial equilibrated homology model. Many RUNs can be generated from the same initial model to meet total

trajectory number criteria, depending on the CLONEs/RUN settings of a particular project. CLONE is an individual trajectory, all CLONEs in a RUN are given different, randomized initial velocities. (**b**) All of the options assessed combinatorially for featurization and tICA optimal hyperparameter selection. *Definitions are described in Materials and methods. (**c**). All of the options assessed combinatorially for final featurization and microstate number selection. *Definitions are described in Materials and methods. (**d**) Summary of 100 microstates in the conformational landscape of apo-SETD8. *Structural features of microstates are assigned based on the conformations of SET-I and post-SET motifs of the 10 conformers that are closest to the cluster center (as 'representative conformations'). The distinct conformational states of SET-I and post-SET motifs described in *Figure 1d* are used as references. Ix (x = 1,2,3) or Py (y = 1,2,3,4) indicate that the representative conformations are very similar to the Ix or Py conformational state observed in crystal structures, respectively. Iab (a,b = 1,2,3, a < b) or Pcd (c,d = 1,2,3,4, c < d) indicate that the representative conformations are positioned between Ia and Ib states or Pc and Pd states, respectively. (**e**) Summary of macrostates in the conformational landscape of apo-SETD8. #Structural features of macrostates are assigned based on the structural features of most populated microstate(s) (>70%). *A11 is composed of two microstates with distinct structural features and comparable populations. (**f**) Summary of 67 microstates in the conformational landscape of SAM-bound SETD8. *Structural features of microstates are assigned based on the conformations of SET-I and post-SET motifs of the 10 conformers that are closest to the cluster center (as 'representative conformations'). The distinct conformational states of SET-I and post-SET motifs described in *Figure 1d* are used as references. Ix (x = 1,2) or Py (y = 1,2,3,4) indicate that the representative conformations are very similar to the Ix or Py conformational state observed in crystal structures, respectively. Iab (a,b = 1,2,3, a < b) or Pcd (c,d = 1,2,3,4, c < d) indicate that the representative conformations are positioned between Ia and Ib states or Pc and Pd states, respectively. (**g**) Summary of macrostates in the conformational landscape of SAM-bound SETD8. *Structural features of macrostates are assigned based on the structural features of most populated microstate(s) (>70%). (**h**) Summary of analysis of rapid-mixing stopped-flow experiments. *Estimated from the average of three data points at highest SAM concentration. Data are best fitting values ± s.e. from KinTek. (**i**) Discovery of microstates by different seed combinations in the conformational landscape of apo-SETD8. Each row presents the condition and results of one test. Seed conformations included in the test are marked as $\sqrt{}$. *Numbering of microstates covered in *Supplementary file 1d*. (**j**) Discovery of microstates by different motif states in the conformational landscape of apo-SETD8. * For #1 ~ 7, combination of seed conformations with the noted SET-I motif conformational states and all possible post-SET motif states, as annotated withthe SET-I states. For #8 ~ 16, combination of seed conformations with the noted post-SET motif conformational states and all possible SET-I motif states, as annotated with the post-SET states. Conformers that display steric clashes and were thus excluded are described in *Figure 3a*. (**k**) Discovery of microstates by different seed combinations in the conformational landscape of SAM-bound SETD8. *Numbering of microstates covered in *Supplementary file 1f*. #Covered by both simulations from TC and BC-SAM. (**l**) Completeness and efficiency of constructing the conformational landscapes of apo-SETD8. *For conditions with a '(⋃TC)", the TC conformer could be either derived directly from crystal structure or generated from the chimeric operations of crystallographically-derived conformers outside the parentheses. The corresponding number of crystallographically-derived conformers as seeds are shown in the next column. The number of covered microstates contributed by seed conformations derived from chimeric operations (including both structural chimeras and TC) are shown outside the parentheses, and the number of covered microstates contributed by only structural chimeras (with TC excluded) are shown in the parentheses. +The minimum simulation time of a seed combination to reach corresponding microstate coverage is listed in the first row. The box is left empty if the maximum coverage of a seed combination is smaller than the corresponding number in the first row. (**m**) Completeness and efficiency of constructing the conformational landscapes of SAM-bound SETD8. +The minimum simulation time of a seed combination to reach corresponding microstate coverage listed in the first row. The box is left empty if the maximum coverage of a seed combination is smaller than the corresponding number in the first row. (**n**) Summary of cancer-associated mutations in the C-terminal region of SETD8 from cBioPortal Cancer Genomics Database. #1 ~ 25: reported before 8/30/2017. #26 ~ 34: reported after 8/30/2017, before 5/1/2018. (**o**) Summary of cancer-associated mutations in the SET-I motif of PKMTs from cBioPortal Cancer Genomics Database (by 5/1/2018). (**p**) Primer sequences for site-directed mutagenesis. Only forward primer

sequences are displayed here. Reverse complementary primers were also ordered. Both forward and reverse primers were used for the experiments.

DOI: https://doi.org/10.7554/eLife.45403.033

• Transparent reporting form

DOI: https://doi.org/10.7554/eLife.45403.034

## Data availability

The molecular dynamics datasets generated and analyzed in this study are available via the Open Science Framework at https://osf.io/2h6p4. The code used for the generation and analysis of the molecular dynamics data is available via a Github repository at https://github.com/choderalab/SETD8-materials (copy archived at https://github.com/elifesciences-publications/SETD8-materials). PDB files: 6BOZ for BC-Inh1, 5W1Y for BC-Inh2, 4IJ8 for BC-SAM, and 5V2N for APO.

The following datasets were generated:

| Author(s) | Year | Dataset title | Dataset URL | Database and Identifier |
|---|---|---|---|---|
| Yu W, Tempel W, Li Y, El Bakkouri M, Shapira M, Bountra C, Arrowsmith CH, Edwards AM, Peter J Brown, Structrual Genomics Consortium (SGC) | 2013 | Crystal structure of the complex of SETD8 with SAM | https://www.rcsb.org/structure/4IJ8 | Protein Data Bank, 4IJ8 |
| Wiewiora R, Chodera J | 2019 | SETD8 wild-type apo and cofactor-bound, and mutant apo Folding@home simulations | https://osf.io/2h6p4 | Open Science Framework, osf.io/2h6p4 |
| Babault N, Anqi M, Jin J | 2019 | Structure of human SETD8 in complex with covalent inhibitor MS4138 | https://www.rcsb.org/structure/6BOZ | Protein Data Bank, 6BOZ |
| Skene RJ | 2018 | Crystal Structure of APO Human SETD8 | https://www.rcsb.org/structure/5V2N | Protein Data Bank, 5V2N |
| Tempel W, Yu W, Li Y, Blum G, Luo M, Pittella-Silva F, Bountra C, Arrowsmith CH, Edwards AM, Brown PJ, Structural Genomics Consortium (SGC) | 2017 | SETD8 in complex with a covalent inhibitor | https://www.rcsb.org/structure/5W1Y | Protein Data Bank, 5W1Y |

The following previously published dataset was used:

| Author(s) | Year | Dataset title | Dataset URL | Database and Identifier |
|---|---|---|---|---|
| Cheng DT, Mitchell TN, Zehir A, Shah RH, Benayed R, Syed A, Chandramohan R, Liu ZY, Won HH, Scott SN, Brannon AR, O'Reilly C, Sadowska J, Casanova J, Yannes A, Hechtman JF, Yao J, Song W, Ross DS, Oultache A, Dogan S, Borsu L, Hameed M, Nafa K, Arcila ME, Ladanyi M, Berger MF | 2015 | MSK-IMPACT | http://www.cbioportal.org/public-portal/ | CBioPortal, MSK-IMPACT |

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

# Appendix 1

DOI: https://doi.org/10.7554/eLife.45403.035

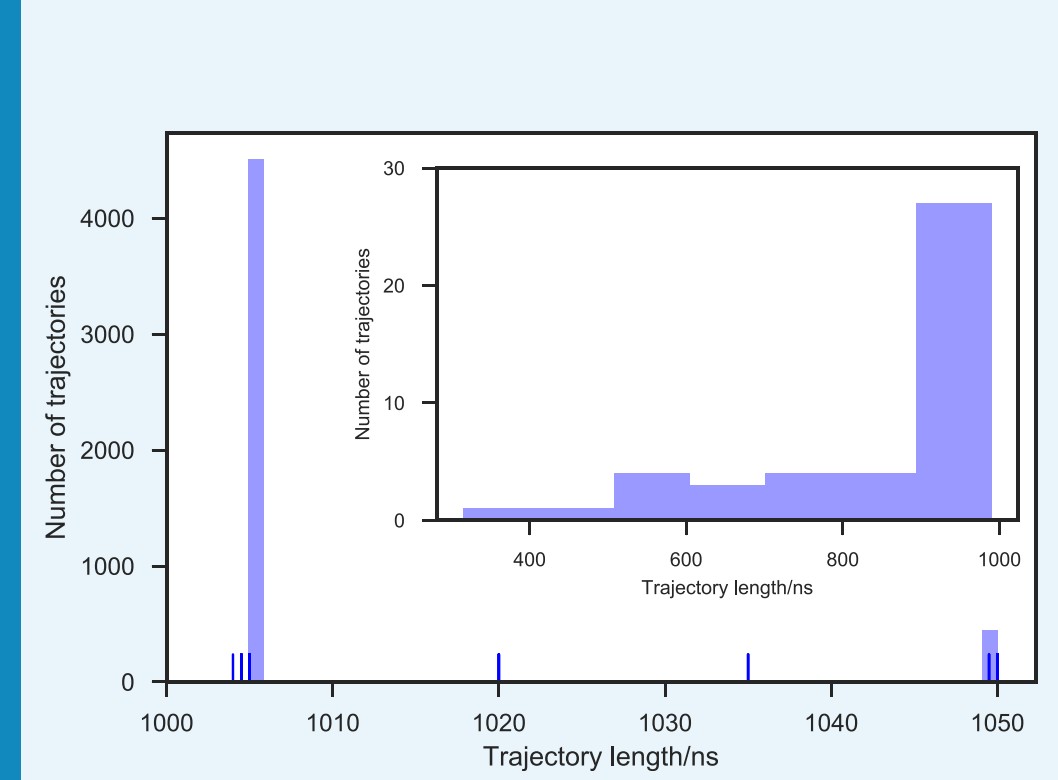

**Appendix 1—figure 1.** Distribution of trajectory lengths of apo-SETD8 simulation. 5,020 Trajectories were collected in total. 99.1% of these trajectories (4,976 trajectories) reached at least 1 µs of simulation time, resulting in 5.058 ms of aggregate simulation time.
DOI: https://doi.org/10.7554/eLife.45403.036

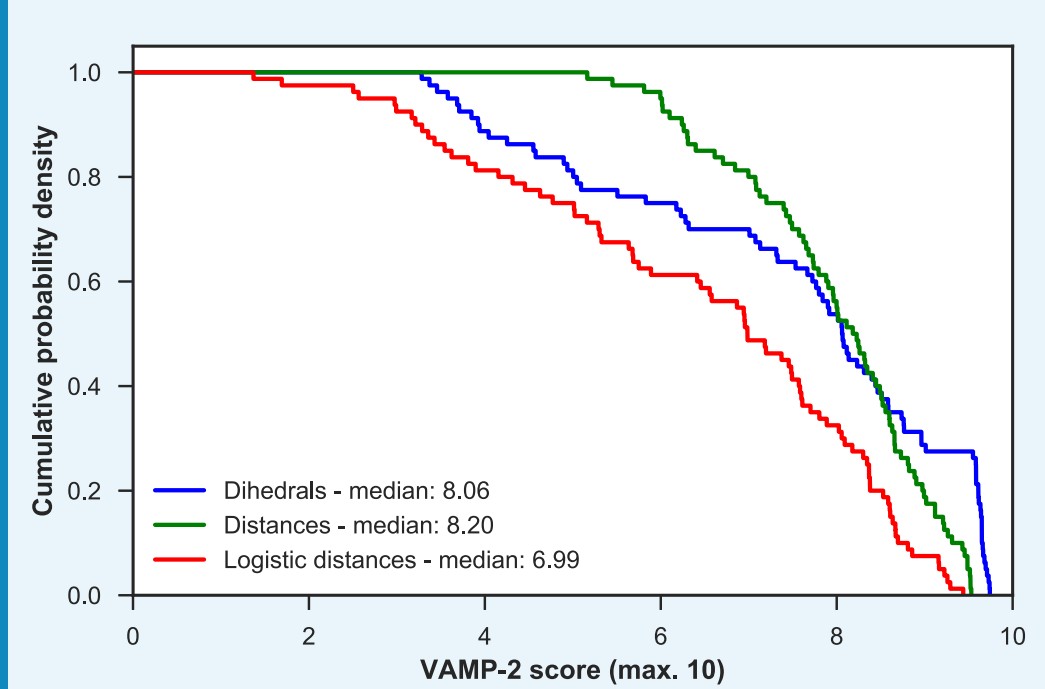

**Appendix 1—figure 2.** VAMP-2 scoring results for optimal hyperparameter choice for featurization and tICA. For each featurization, the empirical cumulative distribution functions of VAMP-2 scores (from highest to lowest) for all other hyperparameter choices combined are shown. On the basis of the medians of all scores, the distance features perform the best.

DOI: https://doi.org/10.7554/eLife.45403.037

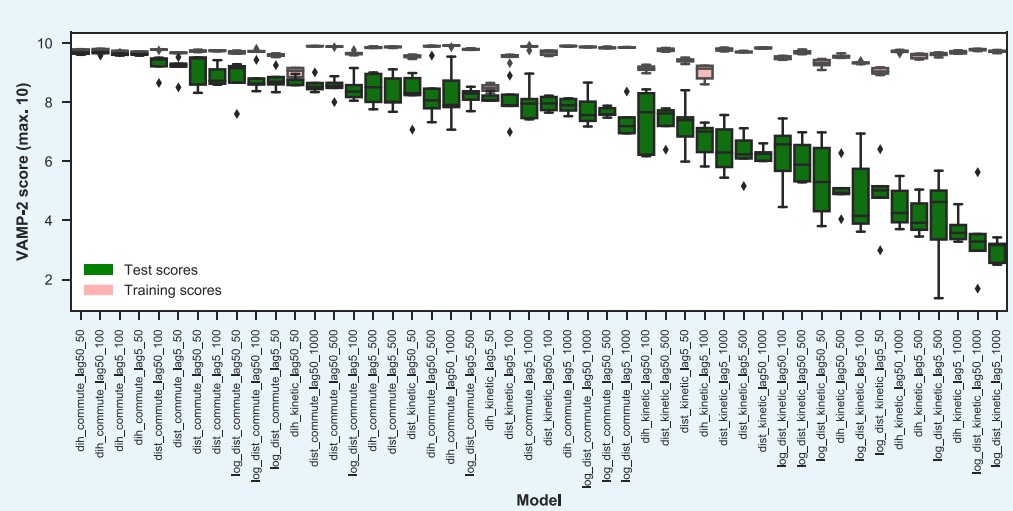

**Appendix 1—figure 3.** VAMP-2 scoring results for optimal hyperparameter choice for featurization and tICA. The distributions of VAMP-2 scores of five shuffle-splits of the data for each individual set of hyperparameters (model) are shown as box-and-whisker plots. Bands of boxes show the first, second, and third quartiles, while whisker ends represent the lowest and highest scores still within 1.5 of the interquartile range from the first and third quartiles respectively. Scores lying outside of that range are shown as diamonds. The models are denoted as {featurization}_{tICA mapping}_{tICA lag time (in ns)}_{number of microstates}, where the featurization is one of 'dih' for dihedrals, 'dist' for distances, or 'log_dist' for logistic distances. Test scores are shown in green and training scores in red. On the basis of the highest scoring individual model, the dihedral features, which were used for the four top scoring models, perform the best.

DOI: https://doi.org/10.7554/eLife.45403.038

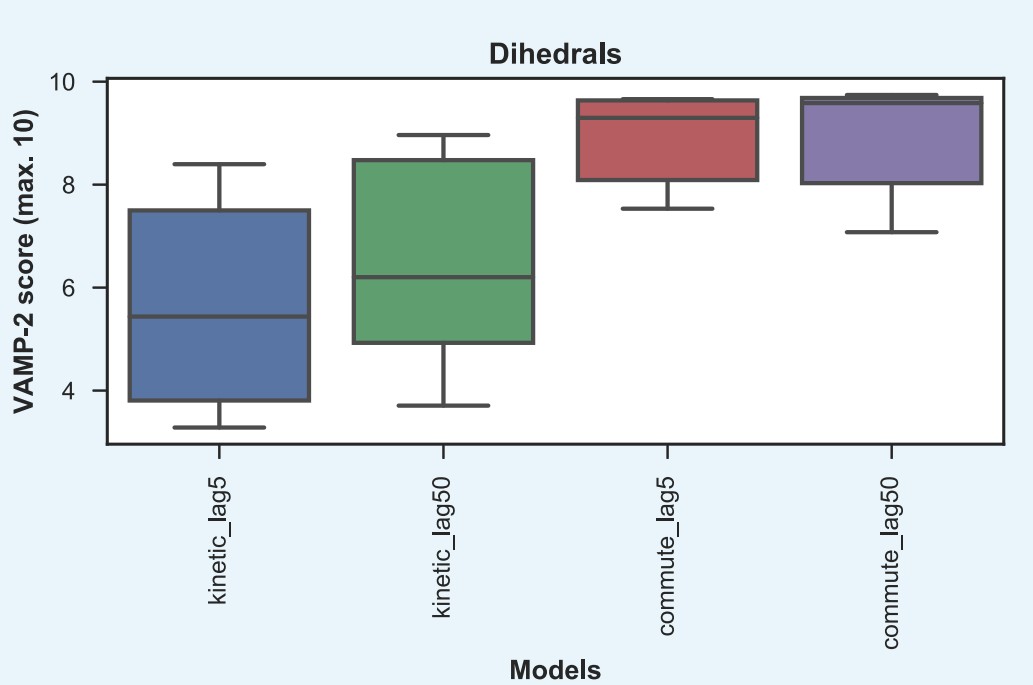

**Appendix 1—figure 4.** VAMP-2 scoring results for optimal hyperparameter choice for tICA with dihedral features. The distributions of test scores of all microstate number choices for all combinations of tICA mapping and lag times are shown as box-and-whisker plots. Bands of boxes show the first, second, and third quartiles, while whisker ends represent the lowest and highest scores still within 1.5 of the interquartile range from the first and third quartiles respectively. The models are described as {featurization}_{tICA lag time (in ns)}. Commute mapping performs significantly better than kinetic mapping and there is no significant difference in performance between the two lag times when commute mapping is used.
DOI: https://doi.org/10.7554/eLife.45403.039

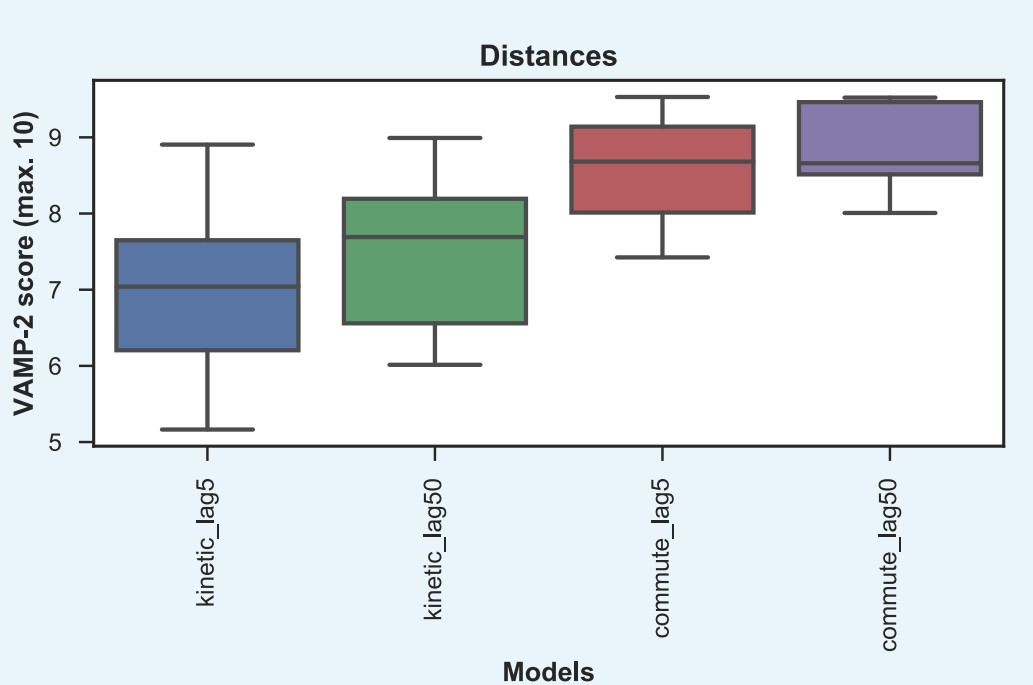

**Appendix 1—figure 5.** VAMP-2 scoring results for optimal hyperparameter choice for tICA with distance features. The distributions of test scores of all microstate number choices for all combinations of tICA mapping and lag times are shown as box-and-whisker plots. Bands of boxes show the first, second, and third quartiles, while whisker ends represent the lowest and highest scores still within 1.5 of the interquartile range from the first and third quartiles respectively. The models are described as follows: {tICA mapping}_{tICA lag time (in ns)}. Commute mapping performs significantly better than kinetic mapping and there is no significant difference in performance between the two lag times when commute mapping is used.

DOI: https://doi.org/10.7554/eLife.45403.040

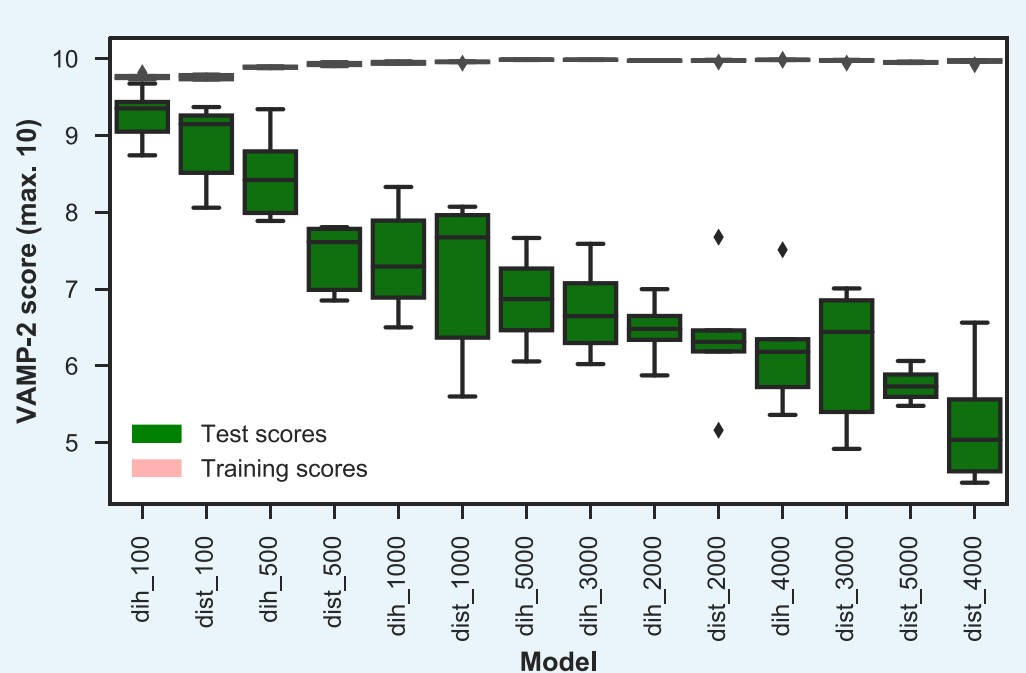

**Appendix 1—figure 6.** VAMP-2 scoring results for final featurization and microstate number choice. The distributions of scores of five shuffle-splits of the data for all combinations of dihedrals or distances featurization and the numbers of microstates are shown as box-and-whisker plots. Bands of boxes show the first, second, and third quartiles, while whisker ends represent the lowest and highest scores still within 1.5 of the interquartile range from the first and third quartiles respectively. Scores lying outside of that range are shown as diamonds. The models are described as follows: {featurization}_{number of microstates}. Featurizations are denoted by 'dih' for dihedrals or 'dist' for distances. Test scores are shown in green and training scores in red (due to the narrowness of the boxes for the training scores the color is not visible). The highest scoring model has dihedral features and 100 microstates.
DOI: https://doi.org/10.7554/eLife.45403.041

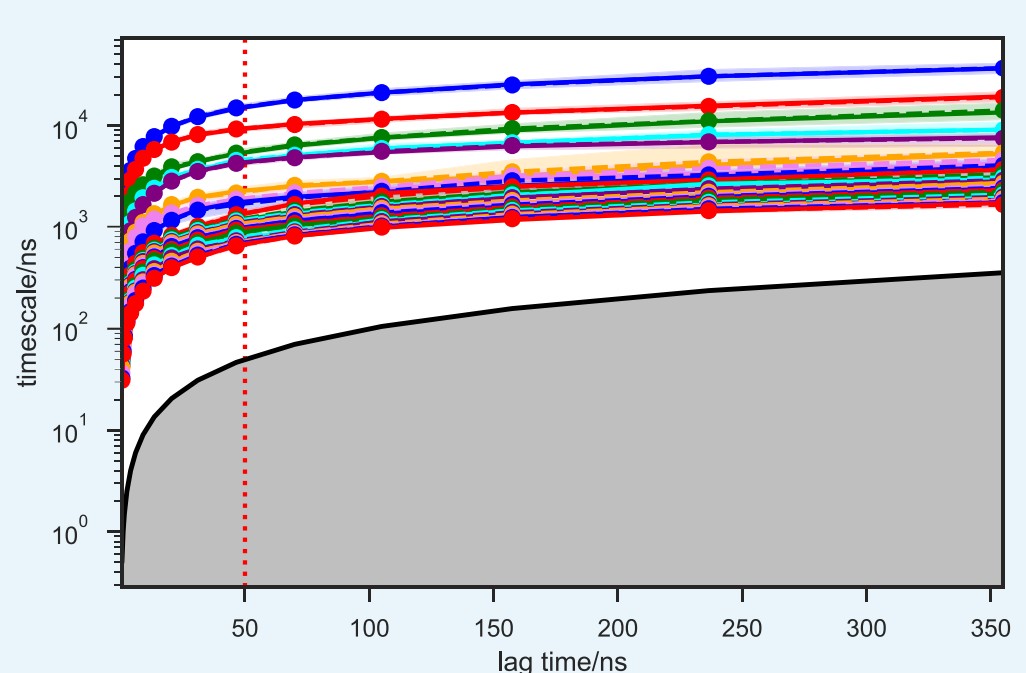

**Appendix 1—figure 7.** Implied timescales of the apo-SETD8 Bayesian Markov state models (BMSMs). The top 23 implied timescales (corresponding to 24 macrostates) of the BMSMs calculated at a range of lag times are shown: the maximum likelihood estimates (MLEs) as solid lines, the means as dashed lines, and the 95% confidence intervals of the means as shaded regions. The gray area signifies the region where timescales become equal to or smaller than the lag time and can no longer be resolved. The lag time of 50 ns (marked by the dashed red vertical line) is chosen for our models, as the timescales have approximately leveled off at that point.

DOI: https://doi.org/10.7554/eLife.45403.042

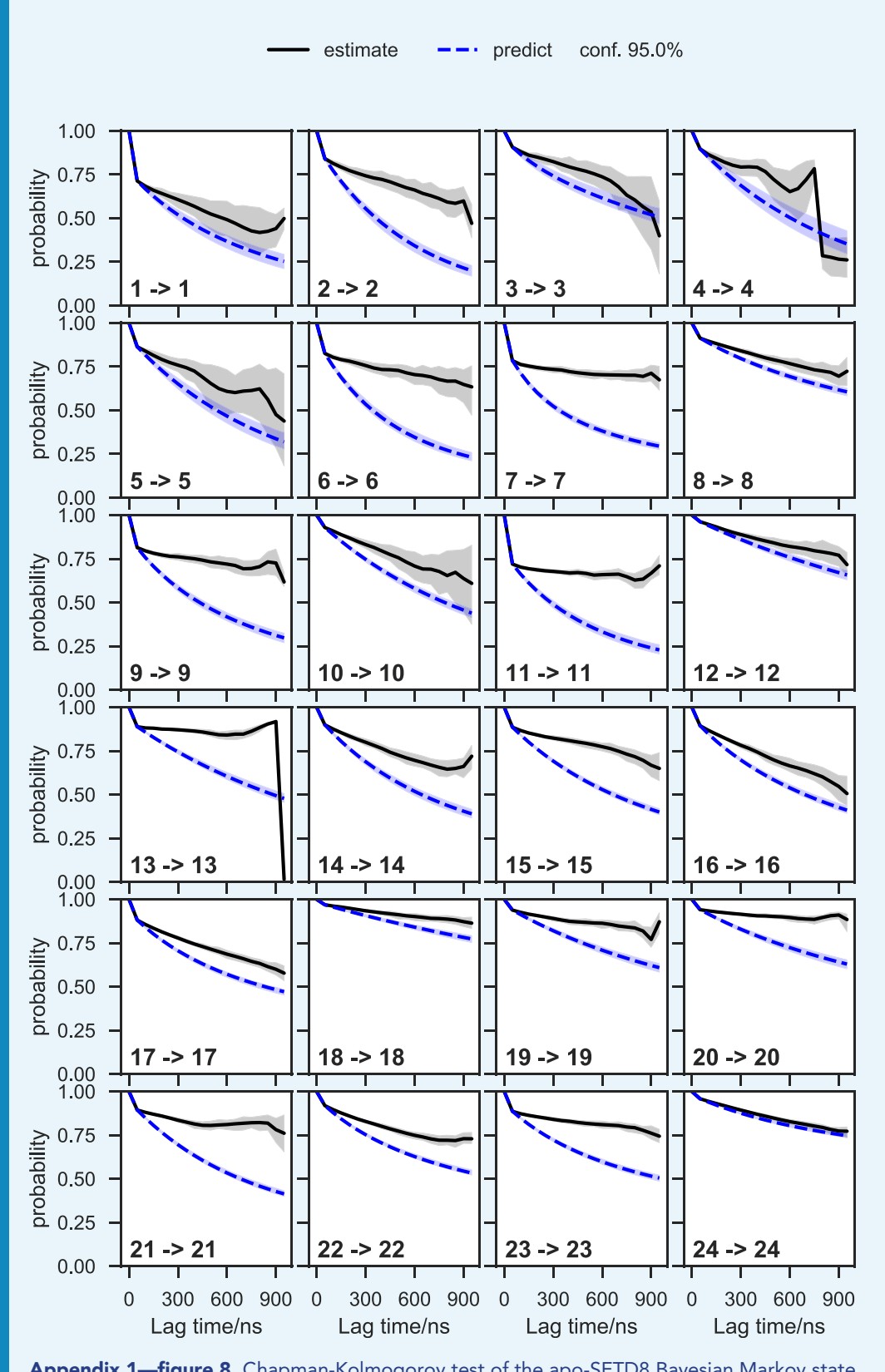

**Appendix 1—figure 8.** Chapman-Kolmogorov test of the apo-SETD8 Bayesian Markov state model (BMSM) using the metastable memberships from the Hidden Markov state model (HMM). The objective of the Chapman-Kolmogorov test is to assess the kinetic self-consistency of the MSM, i.e., whether the predictions of longer time behavior made from the BMSM being

tested match the estimates made from BMSMs generated at longer lag times. For each HMM macrostate, probability density is assigned to the BMSM microstates according to their metastable memberships to the given macrostate and evolution of the probability in time in the tested BMSM is plotted in blue. At those same longer lag times new BMSMs are estimated and their probability densities of being in the given macrostate are plotted in black. The shaded regions correspond to the 95% confidence intervals of the mean of the predictions and estimates. In this case, our model does not faithfully reproduce the empirically-observed slow escape times for many of the macrostates, meaning that insufficient data is available for a quantitative reproduction of the inter-state kinetics; despite this, the equilibrium populations of the macrostates and qualitative resolution of low and high interstate fluxes can still be estimated with good fidelity.

DOI: https://doi.org/10.7554/eLife.45403.043

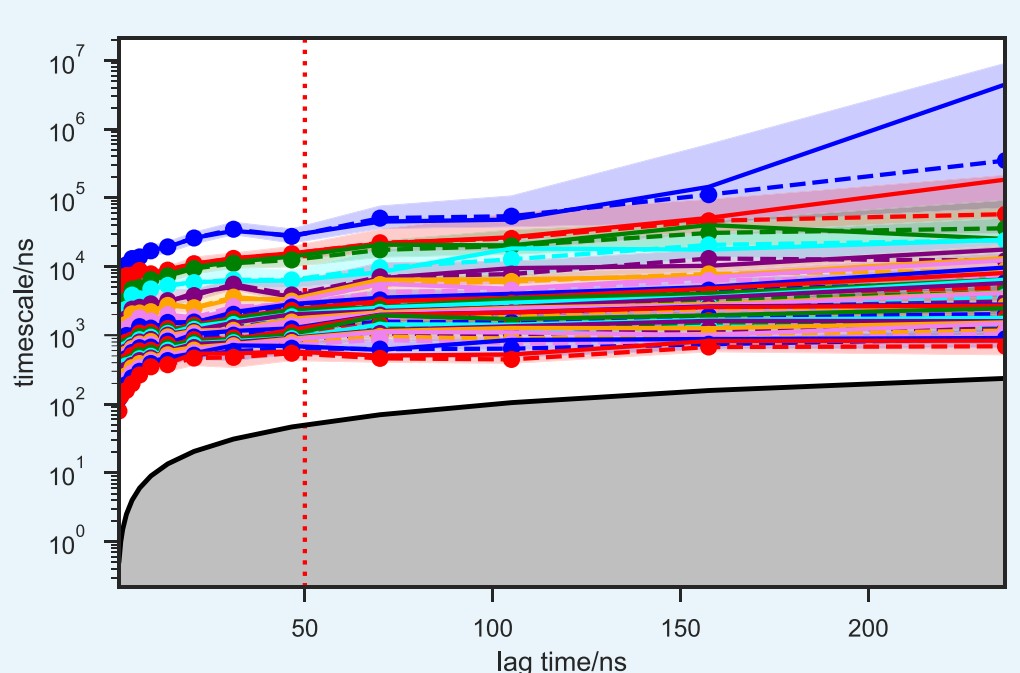

**Appendix 1—figure 9.** Implied timescales of the apo-SETD8 Bayesian Hidden Markov state models (BHMSMs). The top 23 implied timescales (corresponding to 24 macrostates) of the BHMSMs calculated at a range of lag times are shown: the maximum likelihood estimates (MLEs) as solid lines, the means as dashed lines, and the 95% confidence intervals of the means as shaded regions. The gray area signifies the region where timescales become equal to or smaller than the lag time and can no longer be resolved. The lag time of 50 ns (marked by the dashed red vertical line) is chosen for our models, as the timescales have approximately leveled off at that point.

DOI: https://doi.org/10.7554/eLife.45403.044

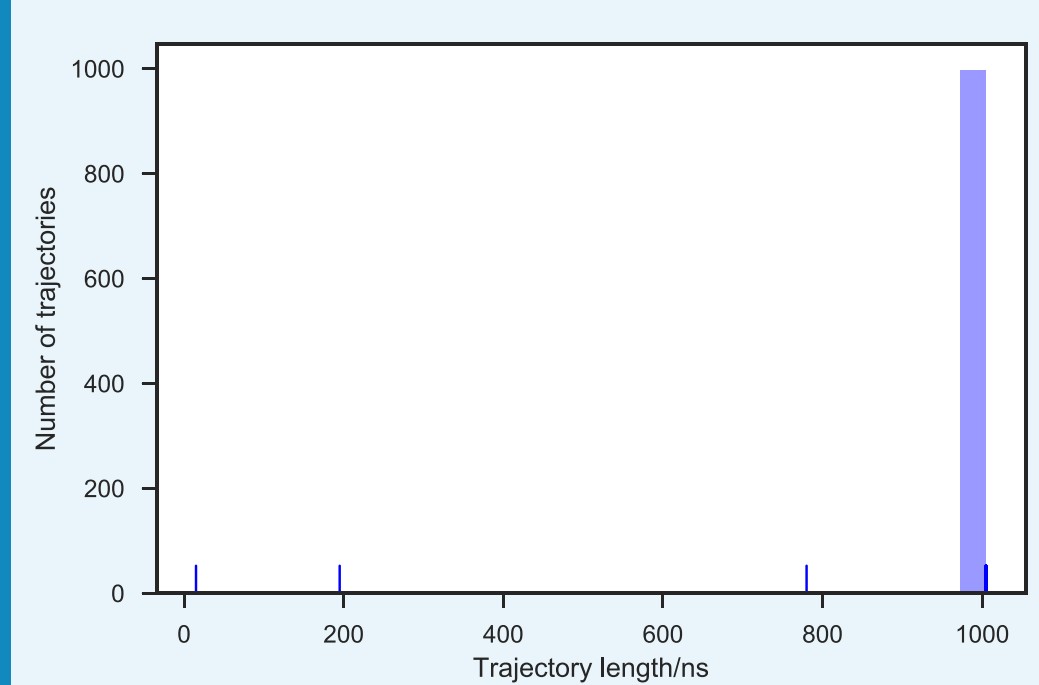

**Appendix 1—figure 10.** Distribution of the SAM-bound SETD8 simulation trajectory lengths. 1,000 trajectories were collected in total. 99.7% of the 1,000 trajectories (997 trajectories) reached at least 1 μs of simulation time, resulting in 1.003 ms of aggregate simulation time.

DOI: https://doi.org/10.7554/eLife.45403.045

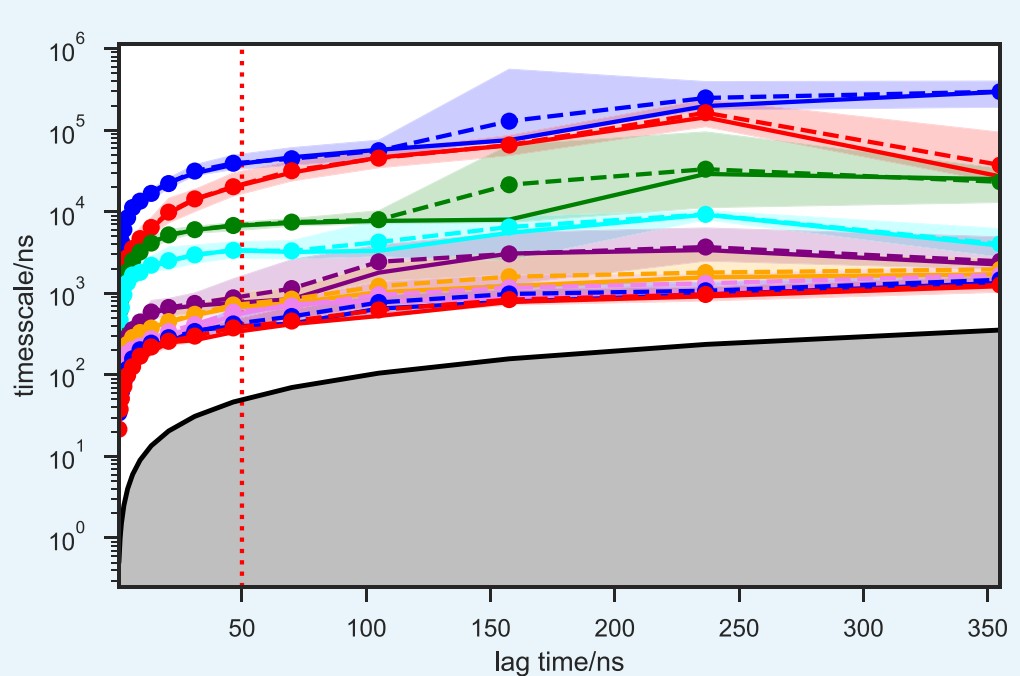

**Appendix 1—figure 11.** Implied timescales of the SAM-bound SETD8 Bayesian Markov state models (BMSMs). The top 9 implied timescales (corresponding to 10 macrostates) of the BMSMs calculated at a range of lag times are shown: the maximum likelihood estimates (MLEs) as solid lines, the means as dashed lines, and the 95% confidence intervals of the means as shaded regions. The gray area signifies the region where timescales become equal to or smaller than the lag time and can no longer be resolved. The lag time of 50 ns (marked by the dashed red vertical line) is chosen for our models, as the timescales have approximately leveled off at that point.

DOI: https://doi.org/10.7554/eLife.45403.046

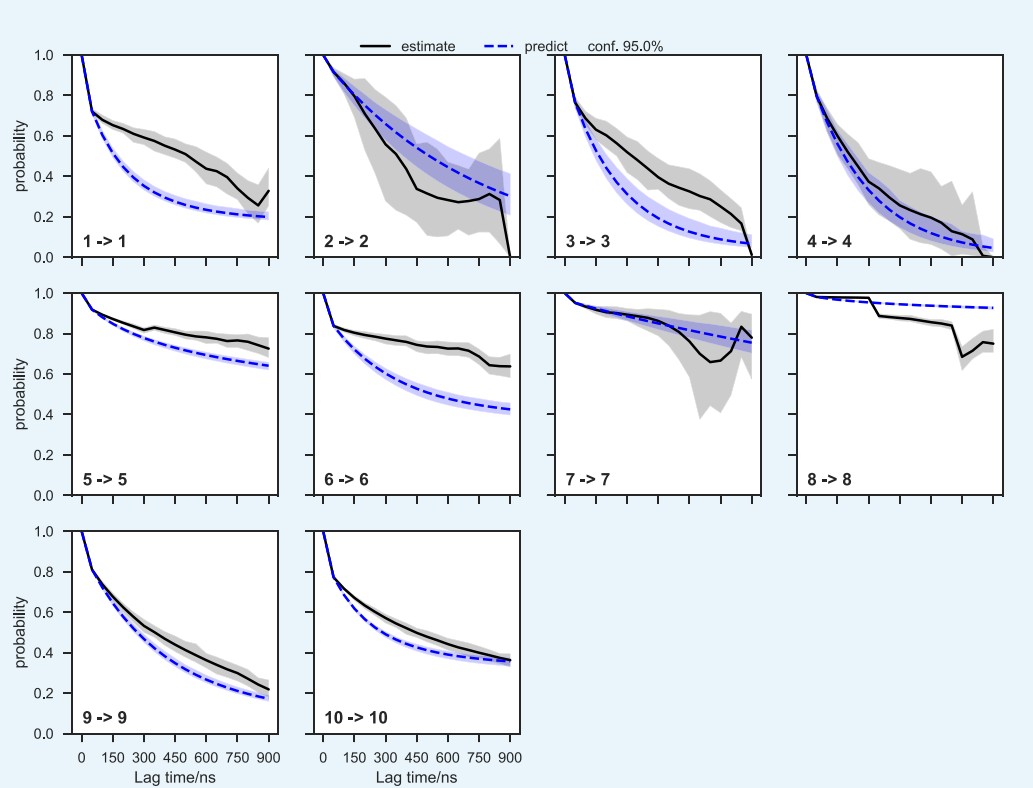

**Appendix 1—figure 12.** Chapman-Kolmogorov test of the SAM-bound SETD8 Bayesian Markov state model (BMSM) using the metastable memberships from the Hidden Markov state model (HMM). The objective of the Chapman-Kolmogorov test is to assess the kinetic self-consistency of the MSM, i.e., whether the predictions of longer time behavior made from the BMSM being tested match the estimates made from BMSMs generated at longer lag times. For each HMM macrostate, probability density is assigned to the BMSM microstates according to their metastable memberships to the given macrostate and evolution of the probability in time in the tested BMSM is plotted in blue. At those same longer lag times new BMSMs are estimated and their probability densities of being in the given macrostate are plotted in black. The shaded regions correspond to the 95% confidence intervals of the mean of the predictions and estimates. In this case, our model does not faithfully reproduce the empirically-observed slow escape times for many of the macrostates, meaning that insufficient data is available for a quantitative reproduction of the inter-state kinetics; despite this, the equilibrium populations of the macrostates and qualitative resolution of low and high interstate fluxes can still be estimated with good fidelity.

DOI: https://doi.org/10.7554/eLife.45403.047

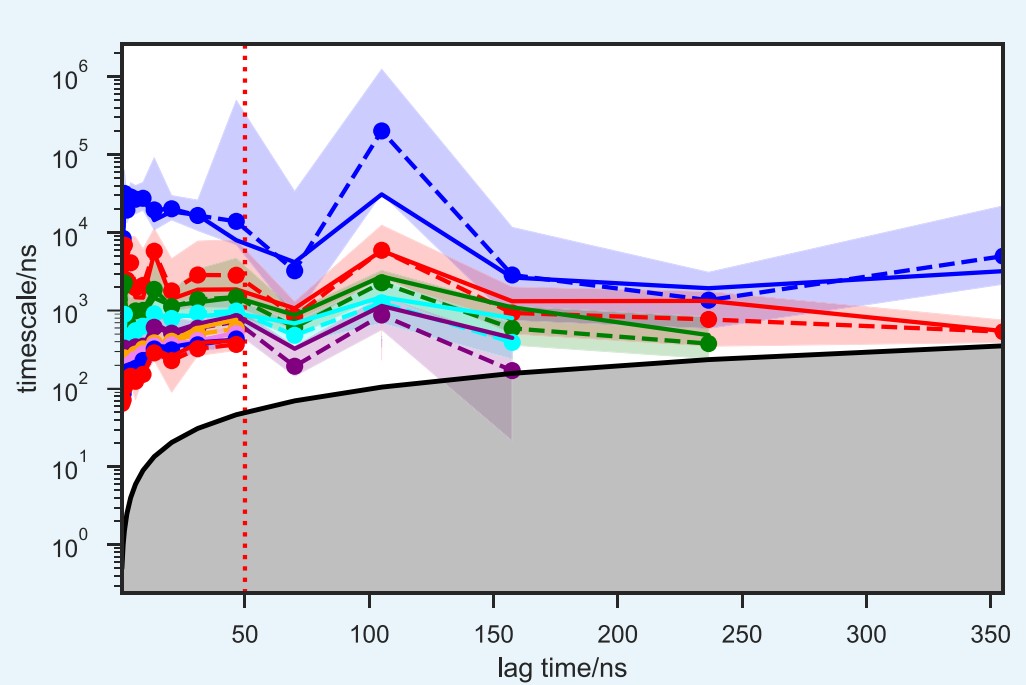

**Appendix 1—figure 13.** Implied timescales of the SAM-bound SETD8 Bayesian Hidden Markov state models (BHMSMs). The top 9 implied timescales (corresponding to 10 macrostates) of the BHMSMs calculated at a range of lag times are shown: the maximum likelihood estimates (MLEs) as solid lines, the means as dashed lines, and the 95% confidence intervals of the means as the shaded regions. The gray area signifies the region where timescales become equal to or smaller than the lag time and can no longer be resolved. The lag time of 50 ns (marked by the dashed red vertical line) is chosen for our models, as the timescales have approximately leveled off at that point.

DOI: https://doi.org/10.7554/eLife.45403.048

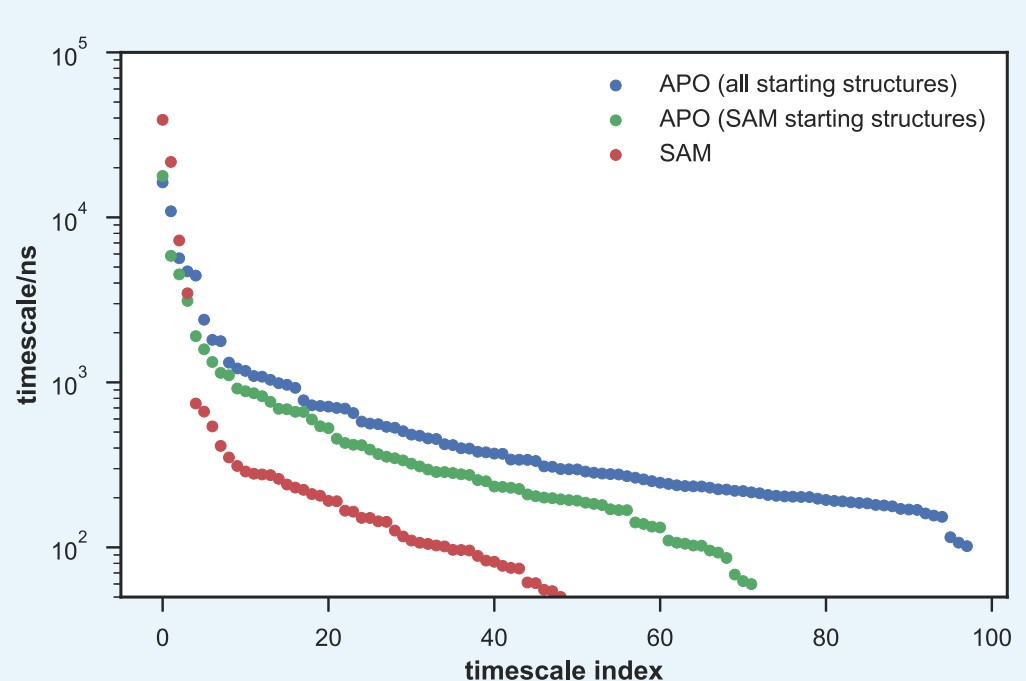

**Appendix 1—figure 14.** Comparison of the kinetic complexity of apo- and SAM-bound SETD8. All timescales larger than the Markovian lag time (50 ns) are shown for Markov state models built using: all apo trajectories (5,019 trajectories), all SAM-bound trajectories (1,000 trajectories), and the subset of apo trajectories starting from the same conformations as SAM-bound trajectories (1,200 trajectories). There is a large decrease in the number of slow processes seen in the SAM-bound model compared to the other two (respectively for the apo, SAM-bound, and subset of apo MSMs there are 14, 4, and 9 processes slower than 1 μs).
DOI: https://doi.org/10.7554/eLife.45403.049

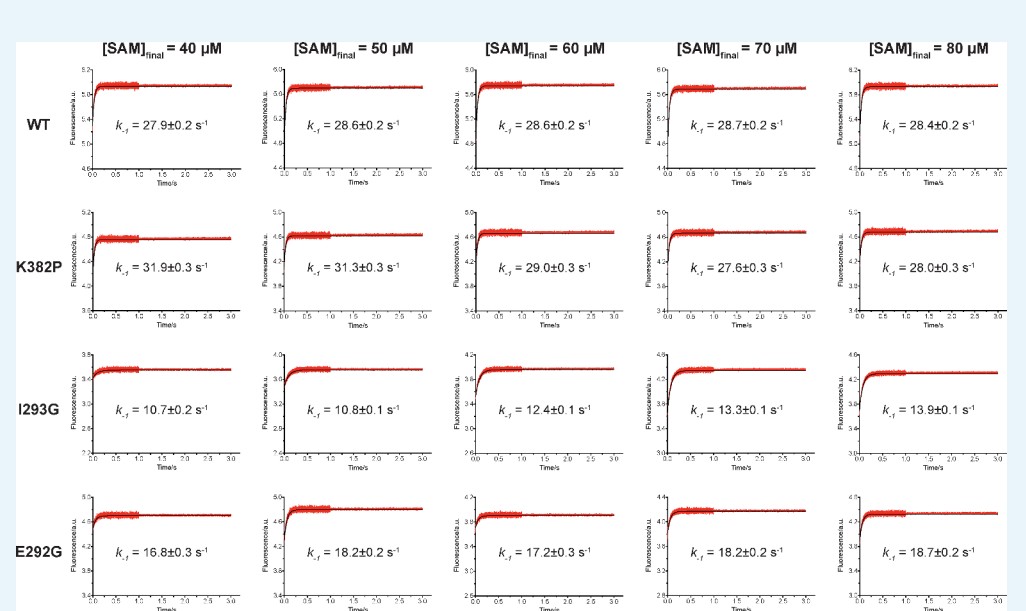

**Appendix 1—figure 15.** Rapid-mixing stopped-flow dilution of SAM-bound SETD8. Fluorescence increase of pre-incubated SETD8-SAM binary complex was determined by rapid-mixing stopped-flow dilution experiments with various final SAM concentrations and analyzed by one-exponential conventional fitting. Given the small fraction of second step in the SAM-binding process, the $k_{obs}$ mainly reflect $k_{-1}$. Data are best fitting values ± s.e. from KinTek.
DOI: https://doi.org/10.7554/eLife.45403.050

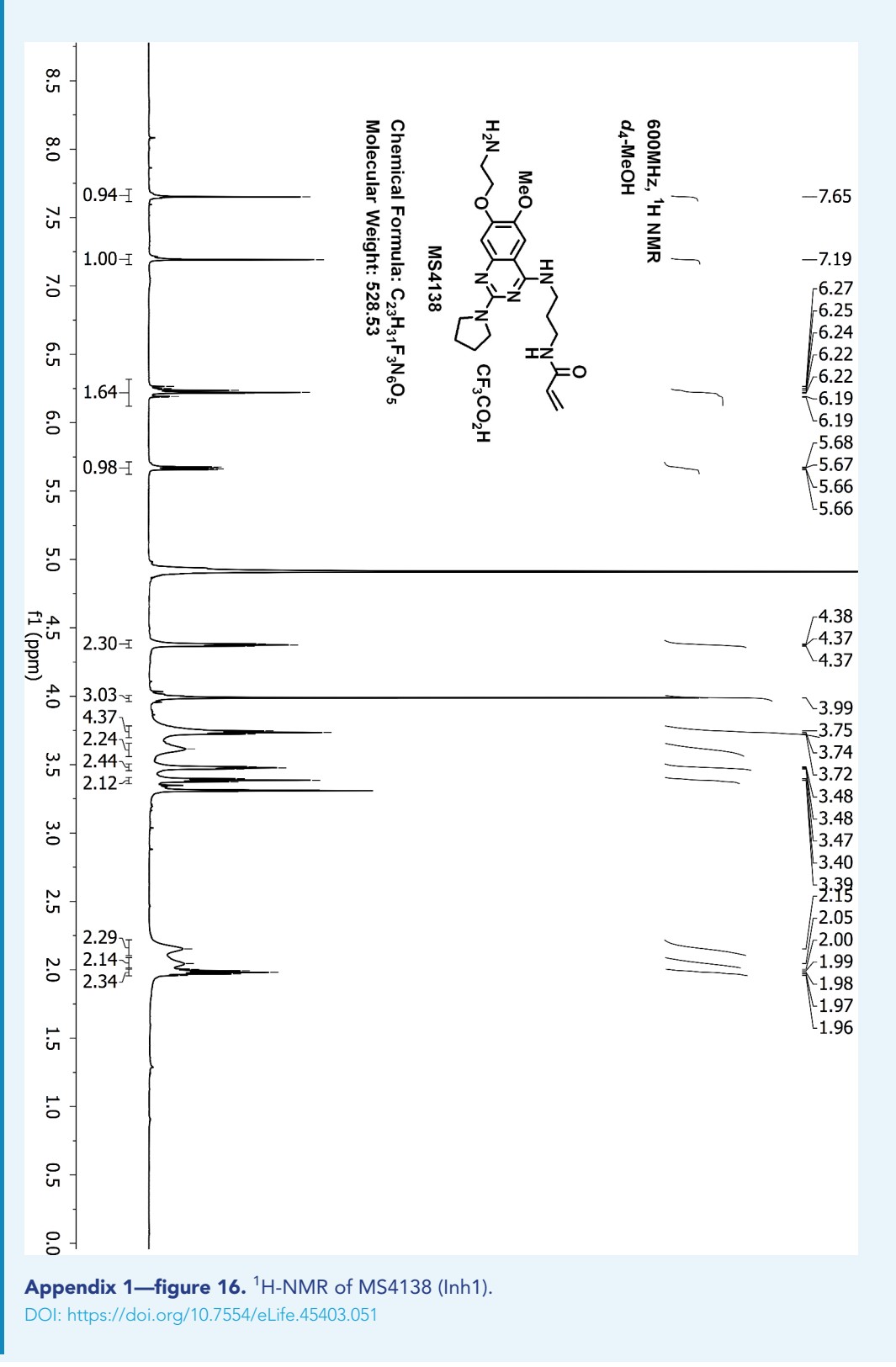

**Appendix 1—figure 16.** [1]H-NMR of MS4138 (Inh1).
DOI: https://doi.org/10.7554/eLife.45403.051

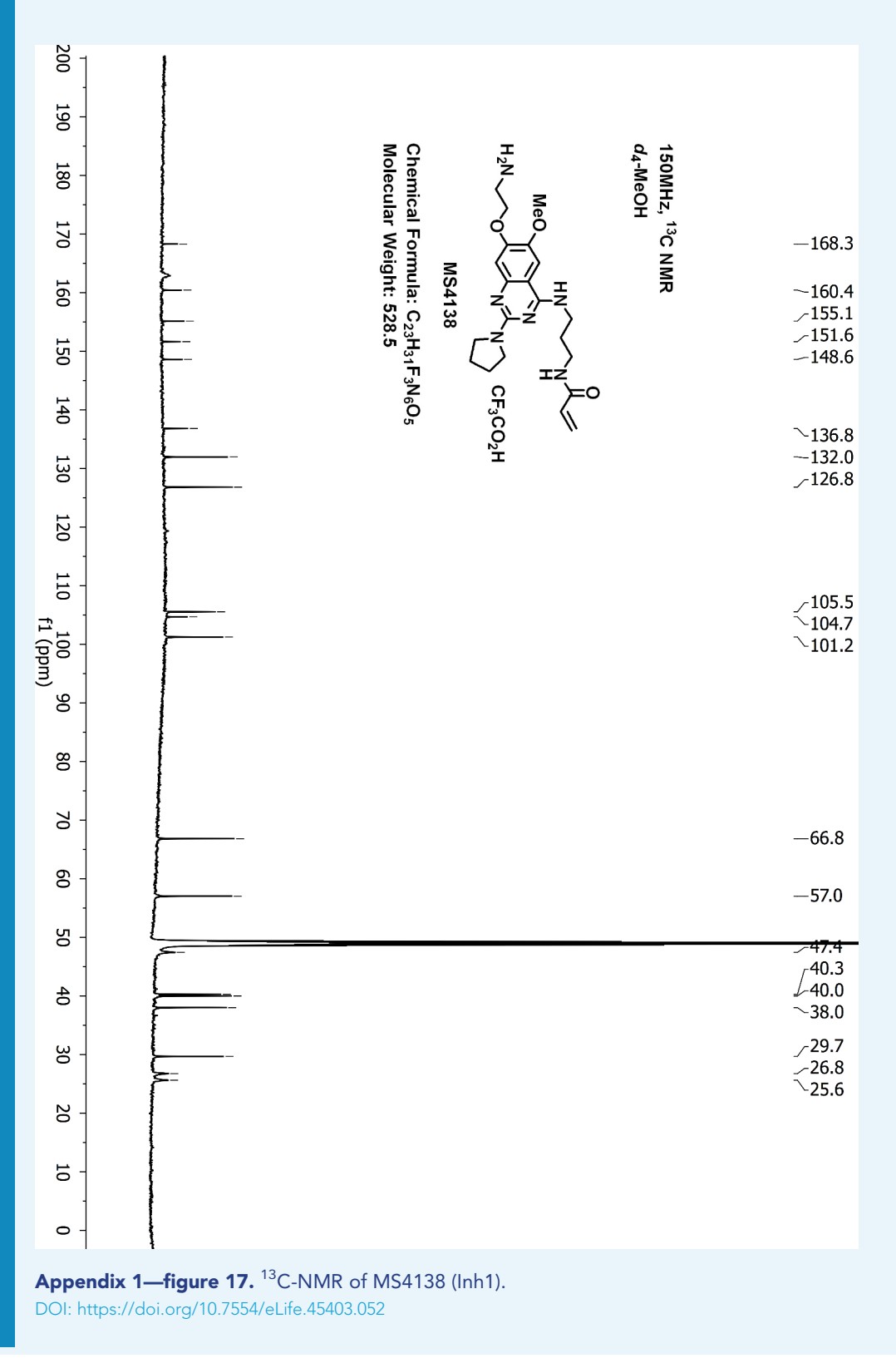

**Appendix 1—figure 17.** $^{13}$C-NMR of MS4138 (Inh1).
DOI: https://doi.org/10.7554/eLife.45403.052

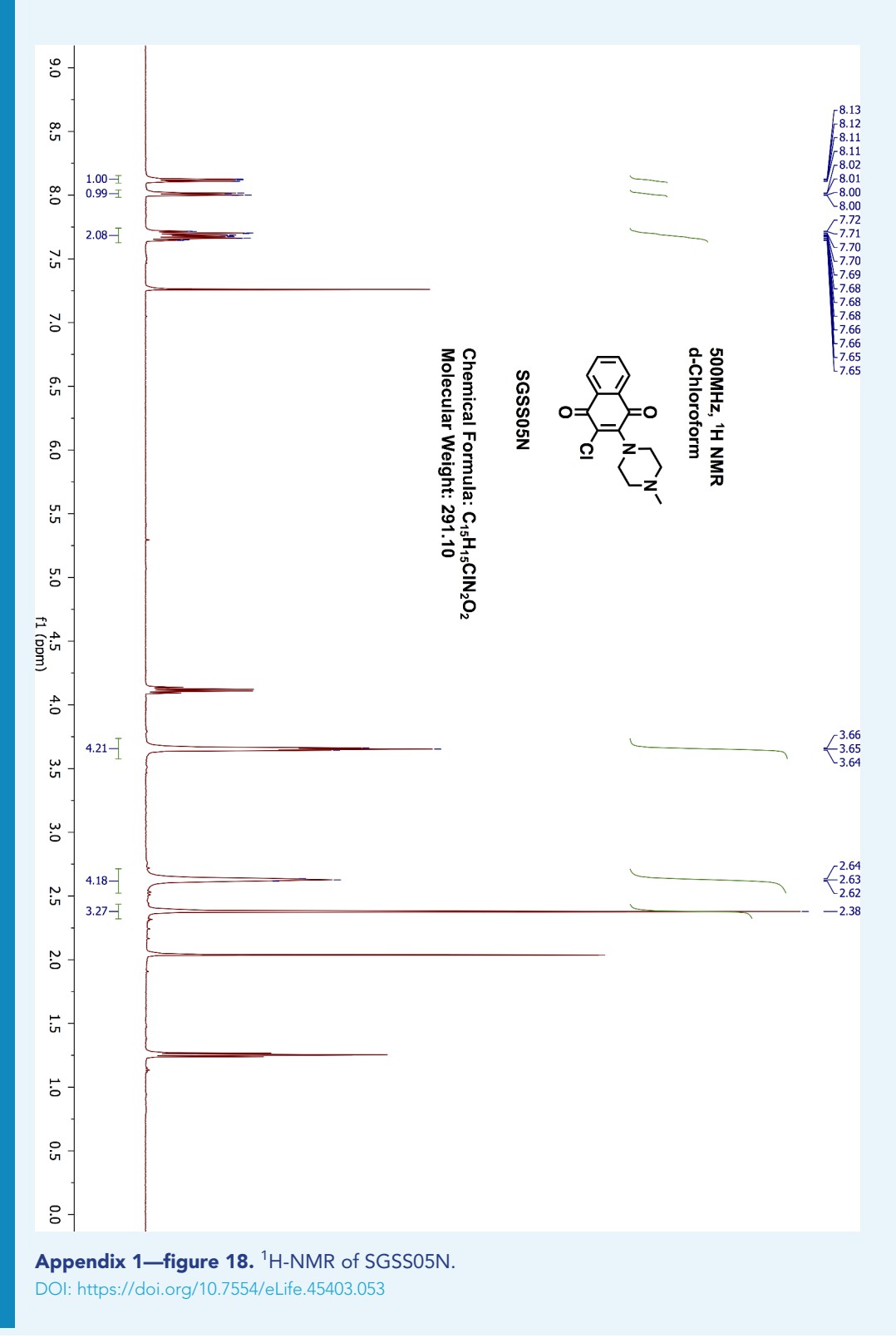

**Appendix 1—figure 18.** [1]H-NMR of SGSS05N.
DOI: https://doi.org/10.7554/eLife.45403.053

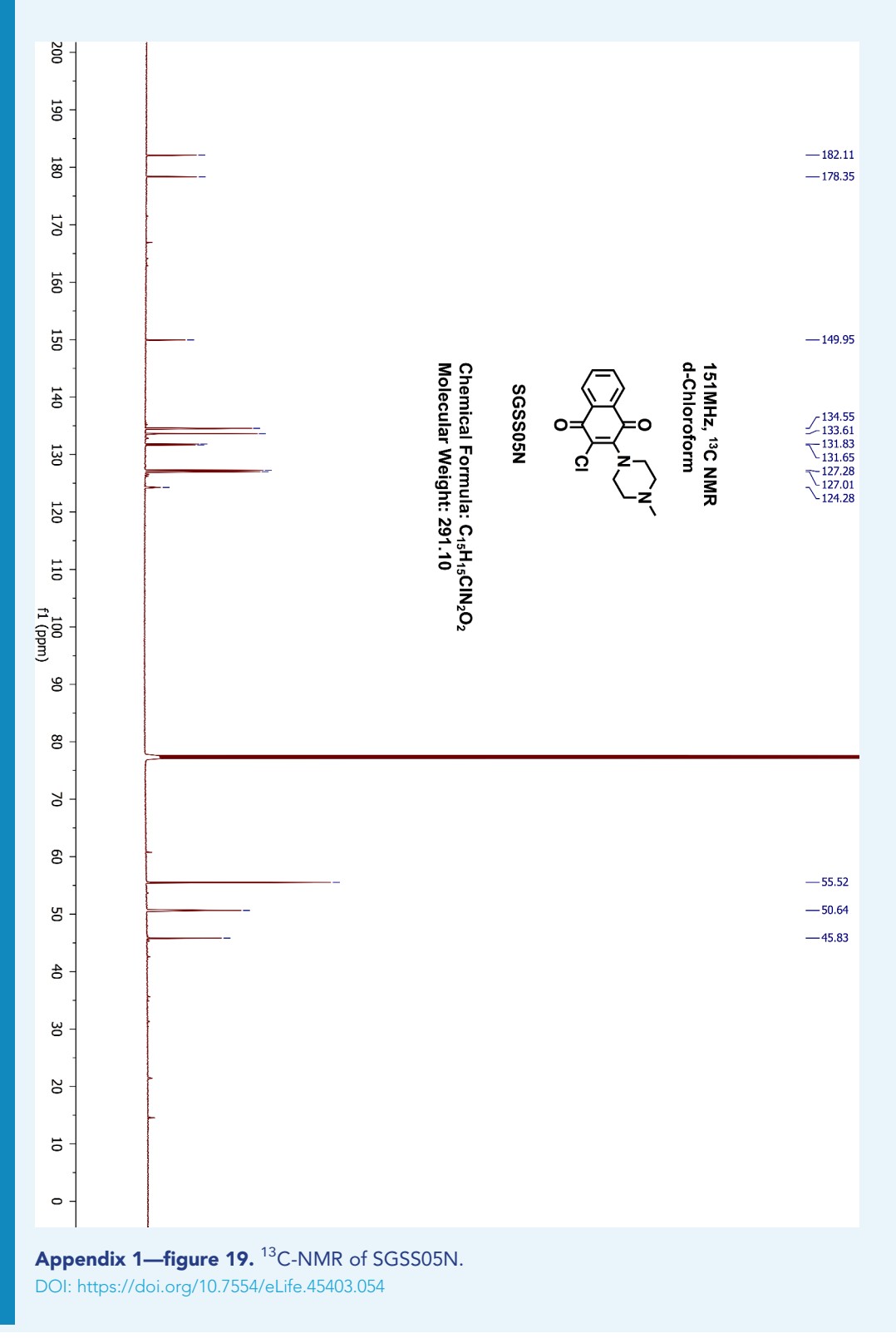

**Appendix 1—figure 19.** $^{13}$C-NMR of SGSS05N.

DOI: https://doi.org/10.7554/eLife.45403.054

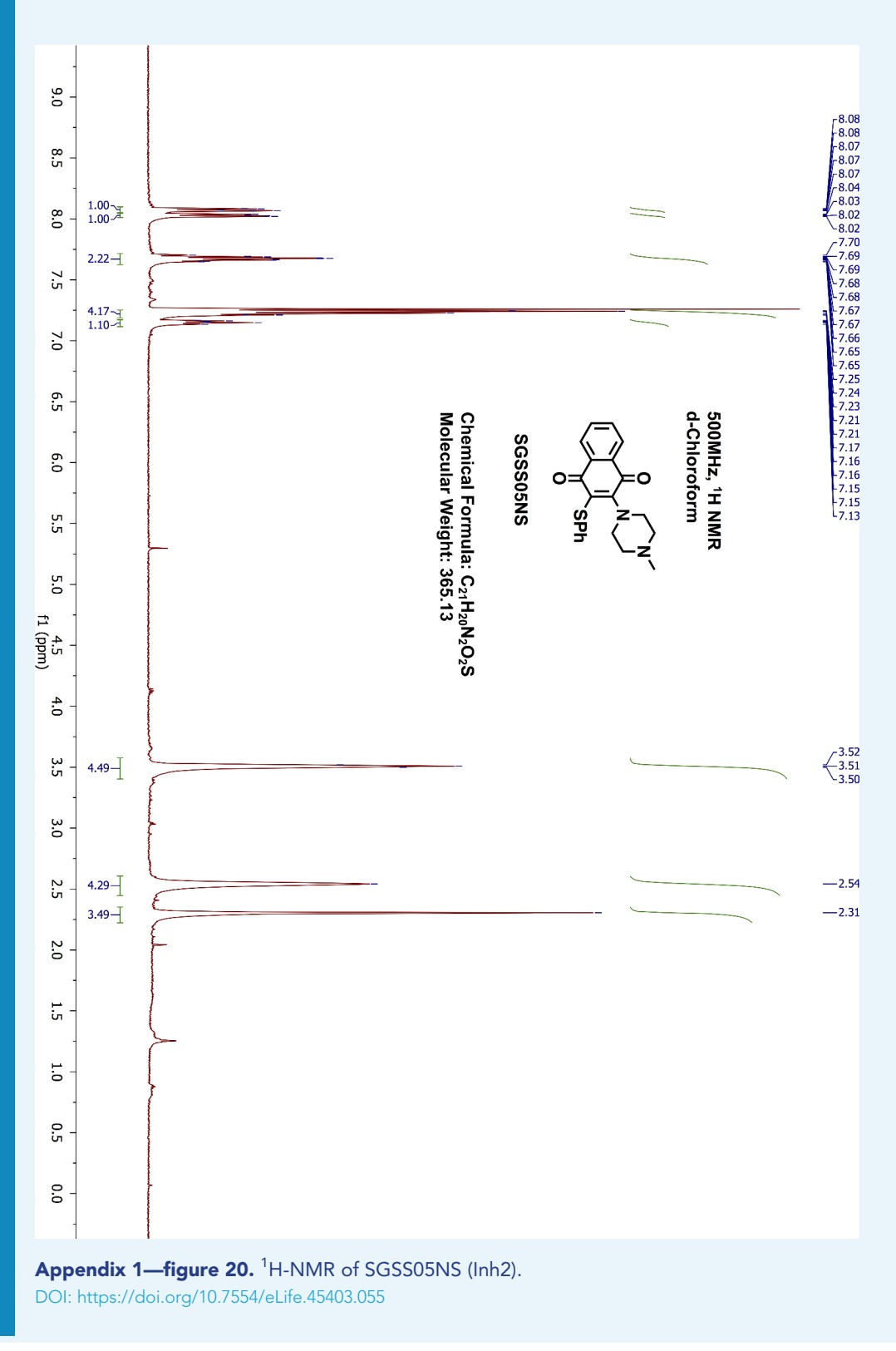

**Appendix 1—figure 20.** [1]H-NMR of SGSS05NS (Inh2).
DOI: https://doi.org/10.7554/eLife.45403.055

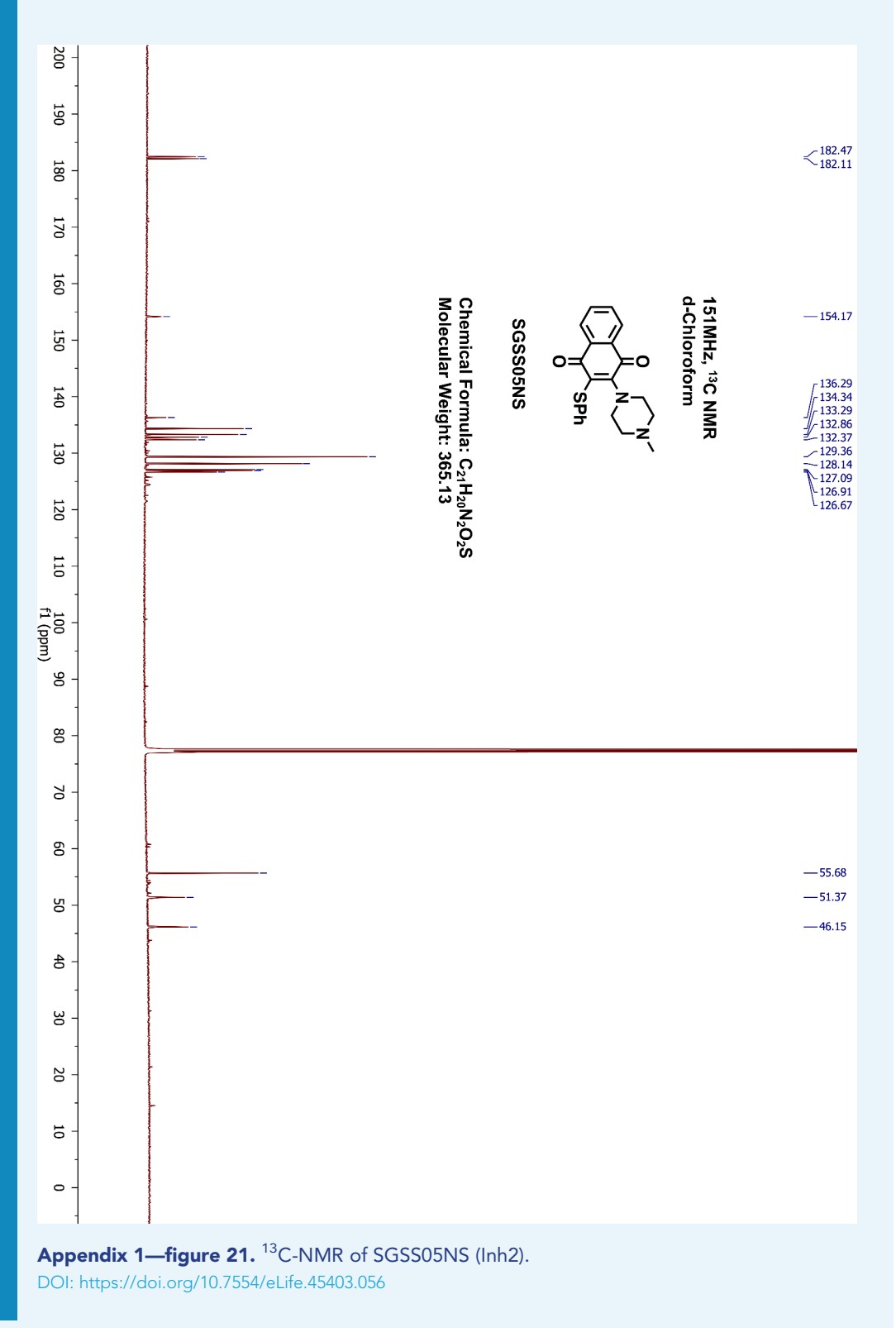

**Appendix 1—figure 21.** $^{13}$C-NMR of SGSS05NS (Inh2).

DOI: https://doi.org/10.7554/eLife.45403.056

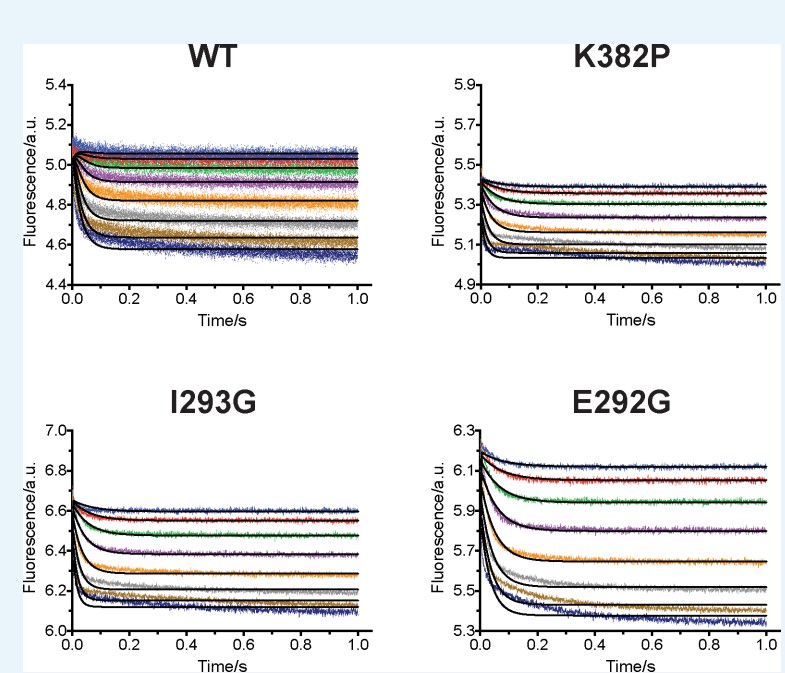

**Appendix 1—figure 22.** Global fitting analysis of stopped-flow binding experiment into a conformational selection model. In contrast to the model we proposed, global fitting analysis of fluorescence decreases from stopped-flow binding experiments into a conformational-selection model (E = E'+SAM = E'SAM) failed to generate good fitting results.
DOI: https://doi.org/10.7554/eLife.45403.057

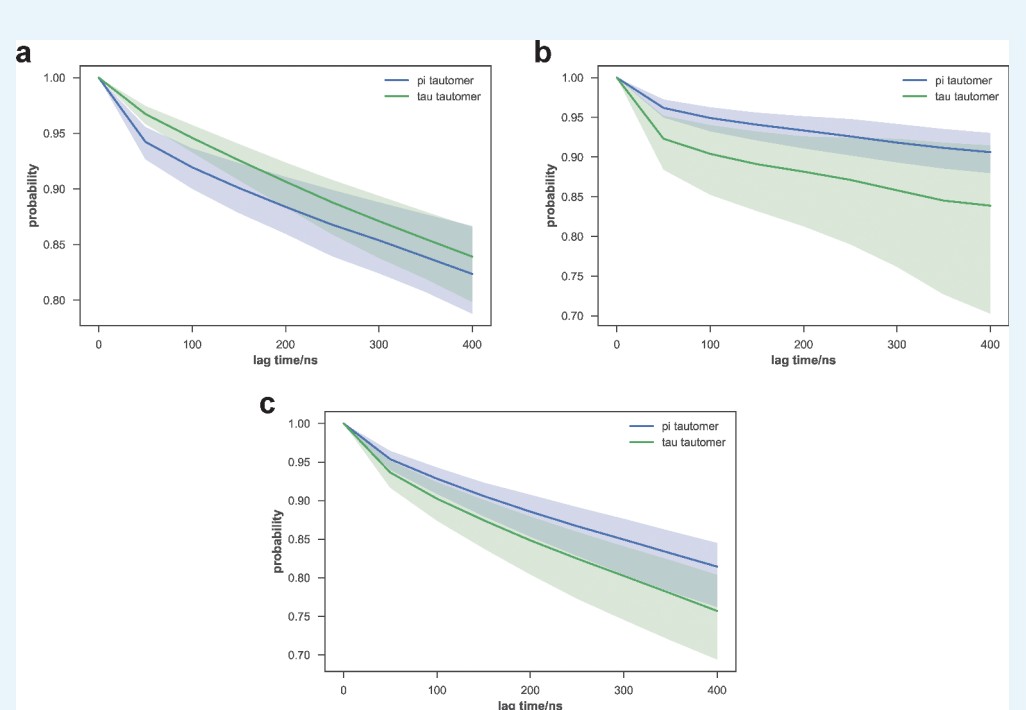

**Appendix 1—figure 23.** Comparison of macrostate escape kinetics for different His351 tautomers of apo-SETD8. As subsets of initial models used for apo-SETD8 simulations used different His351 tautomers, we examined the kinetics of escape from several macrostates that were highly populated by both sets of trajectories. The probabilities of remaining in macrostates A9 (**top**), A1 (**middle**), and A4 (**bottom**) after a given lag time are shown. Means of 40 bootstraps are depicted as solid lines, with 95% confidence intervals shown as shaded regions.
DOI: https://doi.org/10.7554/eLife.45403.058

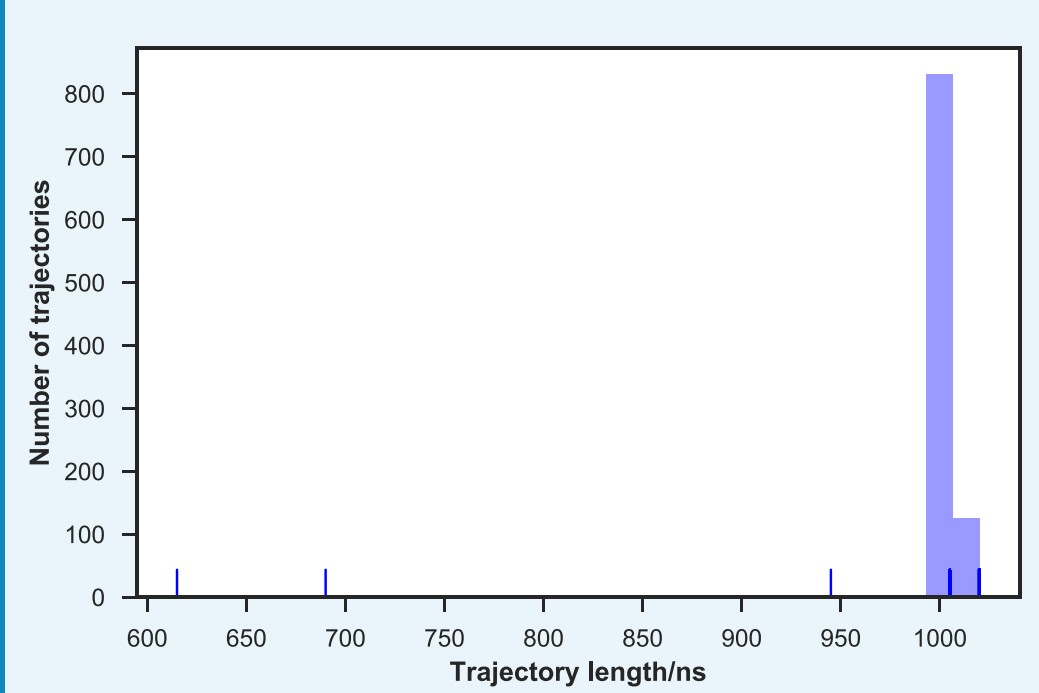

**Appendix 1—figure 24.** Distribution of the 24 mutant apo-SETD8 simulation trajectory lengths. 960 trajectories were collected in total. 99.7% of them (957 trajectories) reached at least 1 µs in length, resulting in 0.966 ms of aggregate simulation time.

DOI: https://doi.org/10.7554/eLife.45403.059

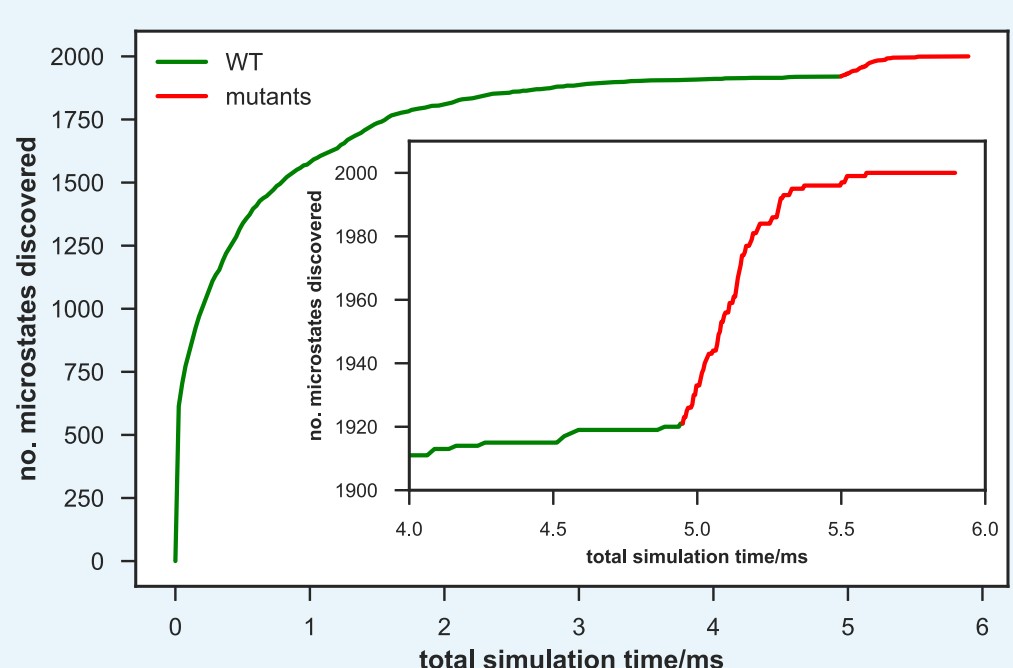

**Appendix 1—figure 25.** Rate of discovery of new microstates in wild-type (WT) and mutant apo-SETD8 datasets. The number of microstates discovered as a function of cumulative aggregate simulation time (corresponding to a uniform initial fraction of all trajectories in the dataset) are shown for a 2,000 joint microstate clustering of the combined WT + mutants apo-SETD8 dataset. The WT data is shown in green, and the mutant data in red, appended to the WT curve for easy comparison. The inset plot shows the number of new microstates discovered for equal amounts of data (~1 ms aggregate simulation time) from the final portion of the WT trajectories and from mutant trajectories. The mutant dataset rapidly discovers 79 new microstates at a rate that far outstrips the discovery rate of new wild-type conformations.
DOI: https://doi.org/10.7554/eLife.45403.060

