## [Decision Letter]

Thank you for submitting your article "The dynamic conformational landscapes of the protein methyltransferase SETD8" for consideration by *eLife*. Your article has been reviewed favorably by three peer reviewers, one of whom is a member of our Board of Reviewing Editors, and the evaluation has been overseen by John Kuriyan as the Senior Editor. The following individuals involved in review of your submission have agreed to reveal their identity: Gregory R Bowman (Reviewer #2); Erik Lindahl (Reviewer #3).

The reviewers have discussed the reviews with one another and the Reviewing Editor has drafted this decision to help you prepare a revised submission.

We expect that the essential points below will be addressed satisfactorily in a revision. You will also find the detailed reviewer reports in an attachment and you should use your own judgment in deciding how to respond to the individual points in the detailed reviews. In any case, please provide a point-by-point response to all of the issues raised by the reviewers.

Summary:

The authors employ a systematic combination of experiments and massively parallel simulations to study the conformational landscape of the methyl transferase SETD8. This is an important protein, and notably, a number of mutations in this protein are associated with cancer. However, the field's understanding of the mechanism of this protein is currently insufficient to understand the effects of these mutations or predict the impact of new mutations.

The results are a tour de force both in terms of the quantity and quality of both the experimental and computational work. The authors show how crystallising a protein in complex with multiple binding partners helps capture different regions of conformational space and how subsequent simulations can provide a fuller picture of the landscape. They explain the conformational/functional impact of binding different partners and of mutations. Importantly, they experimentally verify some of their findings with multiple techniques.

Essential Revisions:

The reviewers agreed that the work described in the manuscript is exciting, comprehensive and rigorous and that the target system is well chosen. The main points to consider during a revision are as follows:

1) Reviewer #2 points out important prior publications that were not cited in the submitted manuscript. These papers and their relevance to the current work should be described in a revision.

2) The reviewers agreed that the main contribution of this paper is to the understanding of the mechanism of action of the SETD8 enzyme itself, and that the language which concerns the general importance of energy landscapes should be limited in all sections of the paper and put into its proper context; that is, that the work contributes to a growing appreciation of energy landscapes to understanding enzyme mechanism. Broader claims about the relevance of the work to other systems should be limited to the discussion and qualified as speculative.

3) The manuscript relies on a large number of figures and supplemental material. Some of the figures are overloaded and should be broken up and the panels enlarged. Also note *eLife*'s rules for supplemental figures and tables.

4) Reviewer #3 asks whether a quantitative analysis of the agreement between observation and simulation of kinetic parameters is possible, and reviewer #1 asks for a clearer conclusion in the Main Text regarding the agreement between the prediction and experimental analysis of mutations in the final part of the paper. Obviously, the agreement between simulation or prediction and experiment is not perfect and the readers would appreciate a quantitative analysis.

5) Reviewer #2 and #3 noted several places where importance or novelty were unnecessarily exaggerated and recommended toning down.

*Reviewer #1:*

Chen et al., describe a comprehensive and interdisciplinary analysis of a human methyltransferase (SETD8). They employ a wide range of techniques, using covalent inhibitors to trap the enzyme in different conformational states, crystallography to provide atomic data on these states, very long MD simulations seeded with the experimental structures to map conformational flexibility, machine learning and Monte Carlo based strategies to define macro conformational states both bound and unbound to the native substrate (SAM), and rational design of gain-of-function mutations based on the analysis of macrostates to verify the model structures. The authors also simulate mutations involved in human disease, revealing a change in the conformational landscape that could not be anticipated from crystal structures alone and suggest that their underlying mechanism is due to changes in conformational dynamics. This is a heroic effort that adds very nicely to a growing body of studies on the importance of analysing the conformational landscape of enzymes in understanding the role of mutations in changing catalytic activity. Although no new methods were developed in this study and the study has not generated conclusions that can generalise to other enzymes, the integration of all the above methods, the specific conclusions drawn for this system and the clarity of the writing are all very impressive.

1) For the benefit of other enzymologists, it would be useful to have a statement about the prerequisites for carrying out this sort of analysis. It clearly requires a free Cys for targeting (right?). How large can the protein system be? Could the authors point to at least a few other disease-related enzymes for which this sort of analysis may be applied?

2) In subsection “Characterization of cancer-associated SETD8 mutants”, the authors state that a number of mutations yielded neoconformations, even though they showed activities similar to the wild type. It would be good to provide some explanation for this observation (how would we know when a neoconformation would be associated with changes in function?) and also a summary sentence that says what percentage of the mutations that exhibited altered activity profiles could be explained through neoconformations as well as what percentage of those with putative neoconformations exhibited altered activity profiles.

*Reviewer #2:*

The authors employ a systematic combination of experiments and massively parallel simulations to study the conformational landscape of the methyl transferase SETD8. This is an important protein, and notably, a number of mutations are associated with cancer. However, the field's understanding of the mechanism of this protein is currently insufficient to understand the effects of these mutations or predict the impact of new mutations.

The results are a tour de force both in terms of the quantity and quality of both the experimental and computational work. The authors show how crystalizing a protein in complex with multiple binding partners helps capture different regions of conformational space and how subsequent simulations can provide a fuller picture of the landscape. They explain the conformational/functional impact of binding different partners and of mutations. Importantly, they experimentally verify some of their findings with multiple techniques.

Taken together, the paper is well aligned with *eLife* and merits publication. However, I have numerous comments that should be addressed prior to publication in order to maximize the impact of this impressive work. Despite the large number of comments, they are easily addressable and I expect the authors can respond quickly.

I suggest increasing the emphasis on the specific system studied in the abstract and introduction and reducing the emphasis on the general conclusion that landscapes are important. The specific system is of great importance, and the importance of landscapes in this sort of setting is a growing theme in the field that is already the subject of reviews (e.g. Knoverek et al., 2018).

The citations disregard the most relevant work in the literature. Hart et al. used MSMs to learn the differences between the landscapes of different enzyme variants, designed new variants to modulate activity by controlling the relative populations of different states, and experimentally verified their designs using a combination of activity assays and structural methods, like mass spectrometry based foot-printing. Latallo et al., 2017 have done something analogous, including solving crystal structures of variants to confirm their structural insights. And Zimmerman et al., 2017 have used MSMs to understand the connection between conformational landscapes and stability, again experimentally testing new designs with stability measurements and crystallography.

Many of the figures have a large number of small panels with tiny text (e.g. Figure 2). I suggest breaking them into multiple figures.

A number of points are overstated. The work is strong enough that exaggeration is unnecessary.

- I don't think it's remarkable the authors see new conformations when they simulate variants. Even 5 ms of data is far from exhaustive. They would probably see new conformations if they simulated wild-type for an additional microsecond. This is also consistent with the fact that some of the variants occupy "new" conformations but show no effect on activity.

- In a number of places, the wording suggests this is unprecedented mechanistic insight into any system, not just SETD8. I strongly disagree with that claim given all the work on allostery and past integration of MSMs with experiments, particularly the work from Hart, Cortina, and Zimmerman listed above. Credit for the related work should be given, and the claims of novelty should be more clearly focused on the specific system.

- Likewise, the claim that this is "the first time that these diverse approaches are consolidated explicitly with the goal of illuminating conformational dynamics of an enzyme in a comprehensive and feasible manner" is false. Again, Hart, Cortina, and Zimmerman provide precedence.

- The lack of correlation between structural similarity and kinetics is not remarkable. It was one of the motivating factors for the development of MSMs.

Mixing data from different tautomers is concerning as the transition rates between states could depend heavily on the tautomeric state. How do the rates compare if the authors estimate them based on 1) only data with τ protonation vs 2) only data with π protonation?

I was more convinced by the comparison between the apo and SAM-bound models after I learned from deep within the methods that they used the same degrees of freedom. I suggest mentioning this point in the main text. Still, I'm curious if the SAM-bound landscape is really simpler, or if the authors just have less sampling. How do the bound/unbound networks compare if they just use 1 ms of unbound data started from conformations analogous to those used for the SAM-bound simulations?

*Reviewer #3:*

This is an interesting quite large multi-group project where new X-ray structures of intermediate states have been engineered (mainly by using ligands) to solve the problem that computational methods aiming at sampling an entire conformational landscape are limited by slow transitions.

Combining multiple experimental structures and simulations to improve sampling might not be quite as novel as the authors indicate; already simple methods such as string with swarms of simulation rely on multiple states, and it is also common for experimental studies to try and capture intermediate states that have then been used in modeling. However, I still very much like this study, in particular the combination with Markov State Models, very careful (automated) selection of features/parameters, and the way it shows how these concepts can be applied to a new system and gain very important biological and medical insight.

The Markov State Models and conclusions are reported with quite a bit of detail, but I found it more difficult to assess exactly HOW well they match the experimental data, rather than just matching gain-vs-loss of function, and to argue the model has been "validated" I would expect to see such a quantitative analysis where the correlation between experiment/model is clearly visible, in particular in terms of kinetics (since "dynamics" is a main argument in the work). Related to this, the simulations appear to have been performed in settings with e.g. "a minimal amount of NaCl", which likely does not correspond to the experiments, not to mention the limited accuracy of force fields.

Second, while dynamics is no doubt critical to understand the transitions in the MSM used to understand the energy landscape, it is less obvious to me how important dynamics itself really is for the mutations studied in the final part of the work. Given that it appears to work quite well to analyse the data changes in terms of residue contacts, most of the functional changes appears to be explained by conformations (and their relative free energy) rather than necessarily the transition rates between them?

---

## [Author Response]

[…] Essential Revisions:The reviewers agreed that the work described in the manuscript is exciting, comprehensive and rigorous and that the target system is well chosen. The main points to consider during a revision are as follows:1) Reviewer #2 points out important prior publications that were not cited in the submitted manuscript. These papers and their relevance to the current work should be described in a revision.

We have corrected this oversight. See our responses to Dr. Bowman below for details.

2) The reviewers agreed that the main contribution of this paper is to the understanding of the mechanism of action of the SETD8 enzyme itself, and that the language which concerns the general importance of energy landscapes should be limited in all sections of the paper and put into its proper context; that is, that the work contributes to a growing appreciation of energy landscapes to understanding enzyme mechanism. Broader claims about the relevance of the work to other systems should be limited to the discussion and qualified as speculative.

We have corrected this oversight. See our responses to Dr. Bowman below for details.

3) The manuscript relies on a large number of figures and supplemental material. Some of the figures are overloaded and should be broken up and the panels enlarged. Also note eLife's rules for supplemental figures and tables.

We have broken up the original figures into 12 smaller, more digestible figures, and associated them with their most essential figure supplements.

4) Reviewer #3 asks whether a quantitative analysis of the agreement between observation and simulation of kinetic parameters is possible, and reviewer #1 asks for a clearer conclusion in the Main Text regarding the agreement between the prediction and experimental analysis of mutations in the final part of the paper. Obviously, the agreement between simulation or prediction and experiment is not perfect and the readers would appreciate a quantitative analysis.

See our answers to Q2 of reviewer #1 and Q1 of Dr. Lindahl below.

5) Reviewer #2 and #3 noted several places where importance or novelty were unnecessarily exaggerated and recommended toning down.

We have corrected this. See our responses to Q4 of Dr. Bowman and Q4 of Dr. Lindahl below.

Reviewer #1:Chen et al. describe a comprehensive and interdisciplinary analysis of a human methyltransferase (SETD8). […] Although no new methods were developed in this study and the study has not generated conclusions that can generalise to other enzymes, the integration of all the above methods, the specific conclusions drawn for this system and the clarity of the writing are all very impressive.1) For the benefit of other enzymologists, it would be useful to have a statement about the prerequisites for carrying out this sort of analysis. It clearly requires a free Cys for targeting (right?). How large can the protein system be? Could the authors point to at least a few other disease-related enzymes for which this sort of analysis may be applied?

While we only examined SETD8 here, with sufficiently diverse seed structures and sufficient computational power, our method is expected to be sufficiently general that it could be readily applied to other enzymes such as H3K36 PKMTs, regardless of their sizes. Our Cys-targeting strategy for accessing seed structures is specific for SETD8 given the general difficulty in developing SETD8 inhibitors – other approaches could be used to capture diverse structures for other systems. We have included these points in the end of Discussion section.

2) In subsection “Characterization of cancer-associated SETD8 mutants”, the authors state that a number of mutations yielded neoconformations, even though they showed activities similar to the wild type. It would be good to provide some explanation for this observation (how would we know when a neoconformation would be associated with changes in function?) and also a summary sentence that says what percentage of the mutations that exhibited altered activity profiles could be explained through neoconformations as well as what percentage of those with putative neoconformations exhibited altered activity profiles.

We reason that certain neo-conformations could still be catalytically active and plan (in future work) to test this hypothesis computationally and experimentally. Our current approach is limited to a qualitative analysis, but as we now mention in the manuscript, future work can develop a more quantitative approach to assessing the impact on individual state populations for various stages of the catalytic cycle using relative alchemical free energy calculations to introduce mutations into mapped metastable states. Our current approach with reference to the conformational landscape shows ~70% accuracy in predicting the outcomes of remote SETD8 mutations (14 out of 20). We have included these points in subsection “Characterization of cancer-associated SETD8 mutants”; Discussion section.

Reviewer #2:[…] Taken together, the paper is well aligned with eLife and merits publication. However, I have numerous comments that should be addressed prior to publication in order to maximize the impact of this impressive work. Despite the large number of comments, they are easily addressable and I expect the authors can respond quickly.I suggest increasing the emphasis on the specific system studied in the abstract and introduction and reducing the emphasis on the general conclusion that landscapes are important. The specific system is of great importance, and the importance of landscapes in this sort of setting is a growing theme in the field that is already the subject of reviews (e.g. Knoverek et al., 2018).

The corresponding changes have been made in the text by putting our results in the context of SETD8 and occasionally PKMTs.

The citations disregard the most relevant work in the literature. Hart et al. used MSMs to learn the differences between the landscapes of different enzyme variants, designed new variants to modulate activity by controlling the relative populations of different states, and experimentally verified their designs using a combination of activity assays and structural methods, like mass spectrometry based foot-printing. Latallo et al., 2017 have done something analogous, including solving crystal structures of variants to confirm their structural insights. And Zimmerman et al., 2017 have used MSMs to understand the connection between conformational landscapes and stability, again experimentally testing new designs with stability measurements and crystallography.

The three references are now included in Introduction and Discussion section.

Many of the figures have a large number of small panels with tiny text (e.g. Figure 2). I suggest breaking them into multiple figures.

We broke up the original figures into 12 smaller, more digestible figures and associated them with the most essential figure supplements.

A number of points are overstated. The work is strong enough that exaggeration is unnecessary.- I don't think it's remarkable the authors see new conformations when they simulate variants. Even 5 ms of data is far from exhaustive. They would probably see new conformations if they simulated wild-type for an additional microsecond. This is also consistent with the fact that some of the variants occupy "new" conformations but show no effect on activity.

We compare the discovery rate of new conformations in the mutant simulations of apo-SETD8 to the discovery rate in an equivalent amount of simulation data of wild-type apo-SETD8 in a newly added Appendix Figure (Appendix 1–figure 25). This figure clearly demonstrates that the mutant simulations indeed discover new conformations at a rate that far outstrips the discovery rate of new wild-type conformations.

- In a number of places, the wording suggests this is unprecedented mechanistic insight into any system, not just SETD8. I strongly disagree with that claim given all the work on allostery and past integration of MSMs with experiments, particularly the work from Hart, Cortina, and Zimmerman listed above. Credit for the related work should be given, and the claims of novelty should be more clearly focused on the specific system.- Likewise, the claim that this is "the first time that these diverse approaches are consolidated explicitly with the goal of illuminating conformational dynamics of an enzyme in a comprehensive and feasible manner" is false. Again, Hart, Cortina, and Zimmerman provide precedence.

We removed the uses of “remarkable”and “unprecedented” and kept a few uses of “the first time” only for SETD8 and PKMTs. We also included the references of Hart, Cortina and Zimmerman.

- The lack of correlation between structural similarity and kinetics is not remarkable. It was one of the motivating factors for the development of MSMs.

We have removed this claim and added an appropriate citation.

Mixing data from different tautomers is concerning as the transition rates between states could depend heavily on the tautomeric state. How do the rates compare if the authors estimate them based on 1) only data with τ protonation vs 2) only data with π protonation?

To compare whether differences in tautomeric state for His351 have a significant impact on kinetics, we have added a new Appendix Figure (Appendix 1–figure 23) that assesses the escape kinetics from macrostates that were highly visited by ensembles of trajectories from both sets of initial conditions (to permit a statistical comparison). No statistically significant difference is evident.

I was more convinced by the comparison between the apo and SAM-bound models after I learned from deep within the methods that they used the same degrees of freedom. I suggest mentioning this point in the main text. Still, I'm curious if the SAM-bound landscape is really simpler, or if the authors just have less sampling. How do the bound/unbound networks compare if they just use 1 ms of unbound data started from conformations analogous to those used for the SAM-bound simulations?

As shown in Supplementary file 1l, 1m, with the same of computational power, SAM-bound SETD8 always samples fewer microstates than apo-SETD8. We have added a discussion of this to the manuscript. We have also added a new Appendix Figure (Appendix 1–figure 14) comparing the number of kinetically metastable states with lifetimes longer than 1 μs for each species. There is indeed a large decrease in the number of the slow kinetically metastable states when SAM is bound.

Reviewer #3:[…] Combining multiple experimental structures and simulations to improve sampling might not be quite as novel as the authors indicate; already simple methods such as string with swarms of simulation rely on multiple states, and it is also common for experimental studies to try and capture intermediate states that have then been used in modeling. However, I still very much like this study, in particular the combination with Markov State Models, very careful (automated) selection of features/parameters, and the way it shows how these concepts can be applied to a new system and gain very important biological and medical insight.

While we agree that such a quantitative analysis would be exciting, it is not feasible for this work, as we do not have quantitative computational models of the SAM binding process – only of the two end states (apo and SAM-bound). Current experiments, using a single fluorescent reporter site, also do not provide sufficient structural information to reconcile their results with computation. We have included a discussion of potential future work on building kinetic models of the whole catalytic cycle of SETD8, and how a powerful framework of “dynamical fingerprints” can be used to reconcile the experimentally observed relaxation processes with structure, as well as design further site-specific labeling experiments (Discussion section).

The Markov State Models and conclusions are reported with quite a bit of detail, but I found it more difficult to assess exactly HOW well they match the experimental data, rather than just matching gain-vs-loss of function, and to argue the model has been "validated" I would expect to see such a quantitative analysis where the correlation between experiment/model is clearly visible, in particular in terms of kinetics (since "dynamics" is a main argument in the work). Related to this, the simulations appear to have been performed in settings with e.g. "a minimal amount of NaCl", which likely does not correspond to the experiments, not to mention the limited accuracy of force fields.

While our simulation conditions are commonly used in the literature, we agree that a study of the potential effects of conditions such as salt concentration, molecular crowding, or choice of forcefield would be exciting and informative, though beyond the scope of the current work. Our models can be used to seed adaptive simulations or extract coordinates for enhanced sampling to efficiently investigate these in the future.

Second, while dynamics is no doubt critical to understand the transitions in the MSM used to understand the energy landscape, it is less obvious to me how important dynamics itself really is for the mutations studied in the final part of the work. Given that it appears to work quite well to analyse the data changes in terms of residue contacts, most of the functional changes appears to be explained by conformations (and their relative free energy) rather than necessarily the transition rates between them?

We agree with Dr. Lindahl on this point and have removed the references to “dynamics” in the discussion of mutants.